# Diamond Blackfan anemia is mediated by hyperactive Nemo-like kinase

M. C. Wilkes [1], K. Siva [2], J. Chen [2], G. Varetti[3,4,5], M. Y. Youn [1], H. Chae[1], F. Ek [6], R. Olsson [6], T. Lundbäck[7], D. P. Dever [8], T. Nishimura [9], A. Narla[1], B. Glader [1], H. Nakauchi [9,10], M. H. Porteus [8], C. E. Repellin [11], H. T. Gazda[12,13], S. Lin[14], M. Serrano [3,4,5], J. Flygare [2] & K. M. Sakamoto[1✉]

Diamond Blackfan Anemia (DBA) is a congenital bone marrow failure syndrome associated with ribosomal gene mutations that lead to ribosomal insufficiency. DBA is characterized by anemia, congenital anomalies, and cancer predisposition. Treatment for DBA is associated with significant morbidity. Here, we report the identification of Nemo-like kinase (NLK) as a potential target for DBA therapy. To identify new DBA targets, we screen for small molecules that increase erythroid expansion in mouse models of DBA. This screen identified a compound that inhibits NLK. Chemical and genetic inhibition of NLK increases erythroid expansion in mouse and human progenitors, including bone marrow cells from DBA patients. In DBA models and patient samples, aberrant NLK activation is initiated at the Megakaryocyte/ Erythroid Progenitor (MEP) stage of differentiation and is not observed in non-erythroid hematopoietic lineages or healthy erythroblasts. We propose that NLK mediates aberrant erythropoiesis in DBA and is a potential target for therapy.

[1] Division of Hematology/Oncology, Department of Pediatrics, Stanford University, Stanford, CA 94305, USA. [2] Department of Molecular Medicine and Gene Therapy, Lund Stem Cell Center, Lund University, Lund 22184, Sweden. [3] Institute for Research in Biomedicine (IRB Barcelona), Barcelona 08028, Spain. [4] Barcelona Institute of Science and Technology (BIST), Barcelona 08028, Spain. [5] Catalan Institution for Research and Advanced Studies (ICREA), Barcelona 08028, Spain. [6] Chemical Biology and Therapeutics Group, Department of Medical Science, Lund University, Lund 22184, Sweden. [7] Chemical Biology Consortium Sweden (CBCS), Science for Life Laboratory, Department for Medical Biochemistry and Biophysics, Karolinska Institutet, 17177 Stockholm, Sweden. [8] Department of Pediatrics, Stanford University, Stanford, CA 94305, USA. [9] Department of Genetics, Institute for Stem Cell Biology and Regenerative Medicine, Stanford University School of Medicine, Stanford, CA 94305, USA. [10] Division of Stem Cell Therapy, Center for Stem Cell Biology and Regenerative Medicine, Institute of Medical Science, University of Tokyo, Tokyo 108-8639, Japan. [11] Biosciences Division, SRI International, Menlo Park, CA 94025, USA. [12] Broad Institute of MIT and Harvard, Cambridge, MA 02142, USA. [13] Division of Genetics and Genomics, Manton Center for Orphan Disease Research, Boston Children's Hospital, Harvard Medical School, Boston, MA 02115, USA. [14] Department of Molecular, Cell and Development Biology, University of California, Los Angeles, CA 90095, USA. ✉email: kmsakamo@stanford.edu

Diamond Blackfan Anemia (DBA) is a congenital bone marrow failure syndrome usually diagnosed within the first year of life[1]. Treatment for DBA, including chronic transfusions, steroid therapy, and stem cell transplantation, are associated with significant morbidity. Therefore, new therapies are needed to treat DBA patients.

Approximately 70% of DBA patients possess a mutation in one of 19 genes that encode ribosomal proteins, with mutations in Ribosomal Protein S19 (RPS19) accounting for over 25% and RPL11 comprising ~5% of cases. Complete loss of ribosomal components is embryonic lethal, while mutations resulting in haploinsufficiency cause erythropoiesis failure due to a block in differentiation of early erythroid progenitors[2].

The master hematopoietic transcription factor c-Myb is expressed in hematopoietic stem cells and early progenitors and is upregulated during early erythropoiesis. Myb serves a number of cellular roles in erythropoiesis, including transcriptional regulation of the master regulators KLF1 and LMO2[3,4] and has been reported to be downregulated in RPS19-insufficiency[5].

NLK is an atypical member of the MAP kinase family and contributes to morphological changes during early embryogenesis, nervous system development and has also been implicated in the pathogenesis of several cancers[6–12]. NLK regulates a diverse array of signaling pathways[6]. In Wnt-stimulated HEK293T cells, NLK phosphorylates c-Myb, priming it for ubiquitination by the E3-ubiquitin kinase Fbxw7 and subsequent proteasome degradation[13–15]. Raptor is another substrate of NLK preventing mTOR-associated Raptor from localizing to the lysosomal membrane for activation[16]. Additional NLK substrates regulated by NLK phosphorylation include STAT3[17], ATF5[18], FoxO1[19], Lef1[20], and HDAC[21].

NLK expression is tissue specific and is regulated by microRNAs (miRNAs), including miR208[22] and miR181[6,23,24]. Family members of miR181 are reported to control NLK expression in a variety of tissues[6].

Here we report NLK activity is increased in erythroid progenitors from DBA models, and suppression of NLK activity increases erythroid expansion. Our results suggest that NLK is critical to the pathogenesis of DBA and is a potential target for therapy.

## Results

**Identification of small molecules that rescue erythropoiesis.** Approximately 12,000 compounds from the Chemical Biology Consortium, Sweden (CBCS), Stockholm and Chemical Biology and Therapeutics (CBT), Lund were screened for the ability to rescue c-Kit$^+$ erythroid expansion in RPS19-insufficient Lin$^-$Kit$^+$ progenitors from mouse embryonic (E14.5–E15.5) fetal livers over 4 days[25] (Fig. 1a). Compound SB431542, a known inhibitor of the TGFβ pathway was among the strongest hits. This compound was also identified as a hit in a similar screen for rescued proliferation of reprogrammed hematopoietic progenitors from DBA patients[26]. We sought to determine if SB431542 rescued proliferation in our screen through inhibition of TGFβ or an off-target effect. Eight TGFβ inhibitors in the murine RPS19-insufficient model were tested at 10 μM and only SB431542 and SD208 significantly rescued c-Kit$^+$ erythroid expansion ($p = 0.0014$, paired Student's $t$ test), while six other TGFβ inhibitors displayed no significant effect (Fig. 1b). Erythroid growth in murine RPS19-insufficient cells improved with SB431542 and SD208 with EC$_{50s}$ of 5 μM and 0.7 μM respectively (Supplementary Fig. 1a). All of the compounds inhibited TGFβ in these cells as each rescued the growth suppression of TGFβ–treated c-Kit$^+$ cells (Fig. 1c).

To determine if SD208 and SB431542 increases erythropoiesis in human DBA models, human cord blood (CB)-derived CD34$^+$ hematopoietic stem and progenitor cells (HSPCs) were transduced with GFP co-expressed with shRNA against control (luciferase) or RPS19 and treated with a more extensive panel of TGFβR1 inhibitors, including SD208 and SB431542. In vitro differentiation of GFP$^+$RPS19-insufficient HSPCs demonstrated impaired hematopoiesis. As in DBA, erythroid differentiation was most dramatically impacted (80–90%) while CD41$^+$ megakaryocyte and CD11b$^+$ myeloid lineages display 20–35% reductions[27–30]. The two compounds (SD208 and SB431542) that increased erythroid expansion in murine RPS19-insufficient progenitors also increased human erythroid progenitor expansion. Galunisertib was not tested in the murine model but also significantly increased erythropoiesis in human DBA models. Compared with control treated cells, SD208, SB431542, and Galunisertib increased erythropoiesis in RPS19-insufficiency by 6.3-fold, 3.2-fold and 3.2-fold, respectively (Fig. 1d). No significant effect was observed in CD41$^+$ megakaryocyte or CD11b$^+$ myeloid populations treated with these compounds (Supplementary Fig. 1b).

Our results demonstrate that the effect of SD208 and SB431542 was not through TGFβ inhibition in mouse progenitors (Fig. 1b, c), however, differences between humans and mice have been reported[31–33]. Since TGFβ inhibits human erythroid and myeloid development[34,35], we compared the ability of SD208 (improved erythropoiesis) and SB525334 (no impact on erythropoiesis) to rescue TGFβ-mediated growth suppression. The presence of either SD208 or SB525334 restored ribosome-competent erythroid (Fig. 1e.i) and myeloid expansion (Fig.1e.iii) of TGFβ-treated cells comparable to untreated controls.

In RPS19-insufficient cells, SD208 increased erythroid expansion by 428.6% of vehicle treated control cells. This increase was not due to TGFβ inhibition, as it was observed in cells treated with or without TGFβ (Fig. 1e.iii). SB525334 restored erythropoiesis to 128.6% of controls in TGFβ-treated cells but did not influence cells that were not treated with TGFβ (Fig. 1e.iii). Both inhibitors equally rescued expansion of TGFβ-treated RPS19-insufficient myeloid cells (Fig. 1e.iv). This indicates that improved erythropoiesis observed by SD208 treatment is not due to TGFβ inhibition in human models of DBA.

To determine if SD208, SB431542 and Galunisertib directly inhibit NLK, all 12 TGFβ inhibitors were added to activated NLK and TGFβR1 and in vitro kinase activity was assessed (see Supplementary Fig. 1c). Similar to functional assays (Fig. 1c, e), all the TGFβR1 inhibitors tested suppressed the phosphorylation of TGFβR1 substrates, Smad2 and Smad3 (Supplementary Fig. 1c) at 1 μM, ranging from ~50 to 90% efficiency. In contrast, the three compounds that increased erythroid expansion in RPS19-insufficiency (SD208, SB431542, and Galunisertib) inhibited NLK-mediated phosphorylation of NLK substrates NLK, c-Myb and Raptor by ~80%, 55% and 40–45%, respectively (Supplementary Fig. 1c).

Immunopurified NLK from drug-treated cells demonstrated the same activity profile as active NLK that had been treated with the drug panel in vitro (Supplementary Fig. 1d). Analysis of SD208, SB431542, and Galunisertib revealed IC$_{50}$ values of 50.6, 87.5, and 62.3 nM against TGFβR1 and IC$_{50}$ values of 435 nM, 1.07 μM and 1.34 μM against NLK (Supplementary Fig. 1e). SD208, SB431542, and Galunisertib are reported to inhibit TGFβR1 with IC$_{50}$ values of 49[35], 94[36], and 69 nM[37] and Eli Lilly has reported NLK inhibition with an IC$_{50}$ value of 0.91 μM for Galunisertib[38].

**NLK is target of SD208 in ribosomal insufficiency.** Kinase Profiling of SD208 indicated a number of kinases are strongly inhibited by this compound (Supplementary Fig. 2a). Many of the

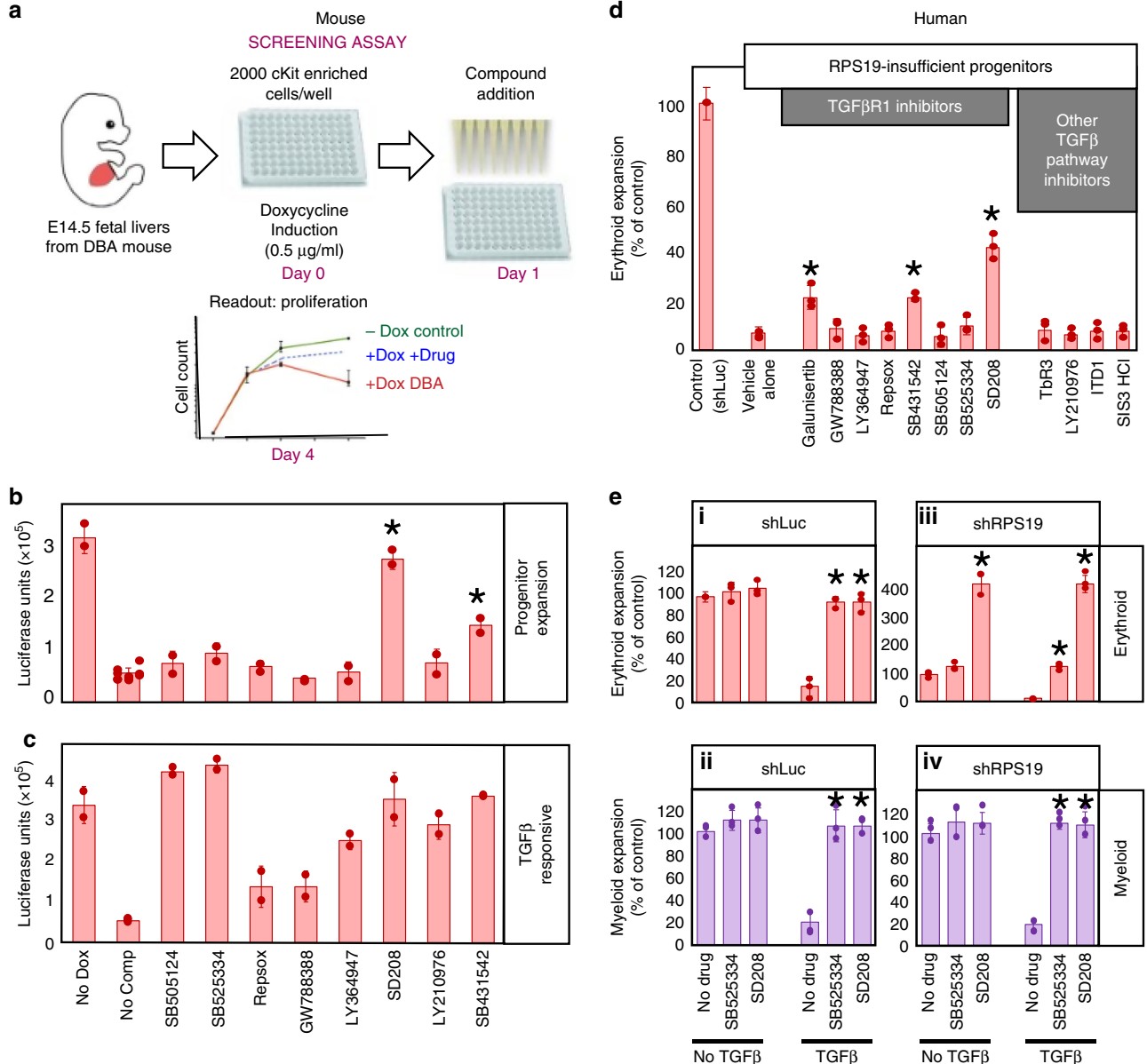

**Fig. 1 TGFβR1 inhibitors that improve erythropoiesis also inhibit NLK activity. a** Schematic of assay utilized to screen compounds for effects on erythroid progenitor cell expansion. Lin-Kit+ fetal liver cells were obtained from mouse embryos expressing tet-on shRNA against RPS19, at day E14.5-15.5. Cells were plated at 2000 cell per well in 96-well plates in the presence or absence of doxycycline. Relative amounts of live cells were quantified by luciferase-based Cell titer-Glo® assay. **b** TGFβR1 inhibitors were assessed for their ability to increase cell expansion in RPS19-insufficiency. As a control, vehicle alone (no doxycycline) is represented at the far left while all other samples were treated with doxycycline to induce RPS19-insufficiency. **c** Kit+ erythroid progenitors were grown in the absence of doxycycline and in the presence of 10 μM of indicated compound. In addition, cells were treated with 5 ng/ml of TGFβ1 for 5 days before being subjected to Cell titer-Glo® assay. **d** Differentiating cord blood CD34+ progenitors were transduced with shRNA against luciferase or RPS19 and treated with inhibitors at working concentrations for TGFβ inhibition every three days. Cells were counted and CD235+ erythroid cells were assessed by flow cytometry after 15 days. **e** Cord Blood CD34+ progenitors were transduced with shRNA against luciferase (i and ii) or RPS19 (iii and iv) differentiated in erythroid media for 15 days alone, or the indicated combinations of 5 ng/ml TGFβ1, SB525334 or SD208 at 5 μM. Cells were counted and CD235+ erythroid (i and iii) and CD11b+ myeloid cell (ii and iv) percentages were determined by flow cytometry. The number of erythroid or myeloid cells is expressed as a percentage of the number of that lineage with no cytokine or drug treatment. Bars represent means ± SD with individual data points overlaid. n = 3 independent experiments performed in triplicate. Statistics: two-tailed Student's t test, significant *p < 0.05. Also see Supplementary Fig. 1. Erythroid expansion is depicted in red while myeloid is in purple. Source data are provided as a Source Data file.

kinases were not studied because compounds that specifically target them did not improve erythropoiesis in our screen. However, SD208 inhibited p38 and NLK kinase activity by 99.6% and 96.8%, respectively at 10 μM (Supplementary Fig. 2a) and the low-specificity p38 inhibitor SB203580 improves erythropoiesis in RPS19-insufficient HSPCs[27]. Interestingly, SB203580 also inhibits NLK at similar concentrations[39–41]. We compared SD208 with

SB203580 in addition to two other p38 inhibitors that do not target NLK[41,42] (PH797804 and SCIO-469) and only SD208 and SB203580 improved erythropoiesis, indicating suppression of p38 does not improve erythropoiesis. All four inhibitors potently reduced p38 phosphorylation (Supplementary Fig. 2b).

To determine if NLK contributes to decreased erythropoiesis in DBA models, we transduced siRNA against NLK and shRNA

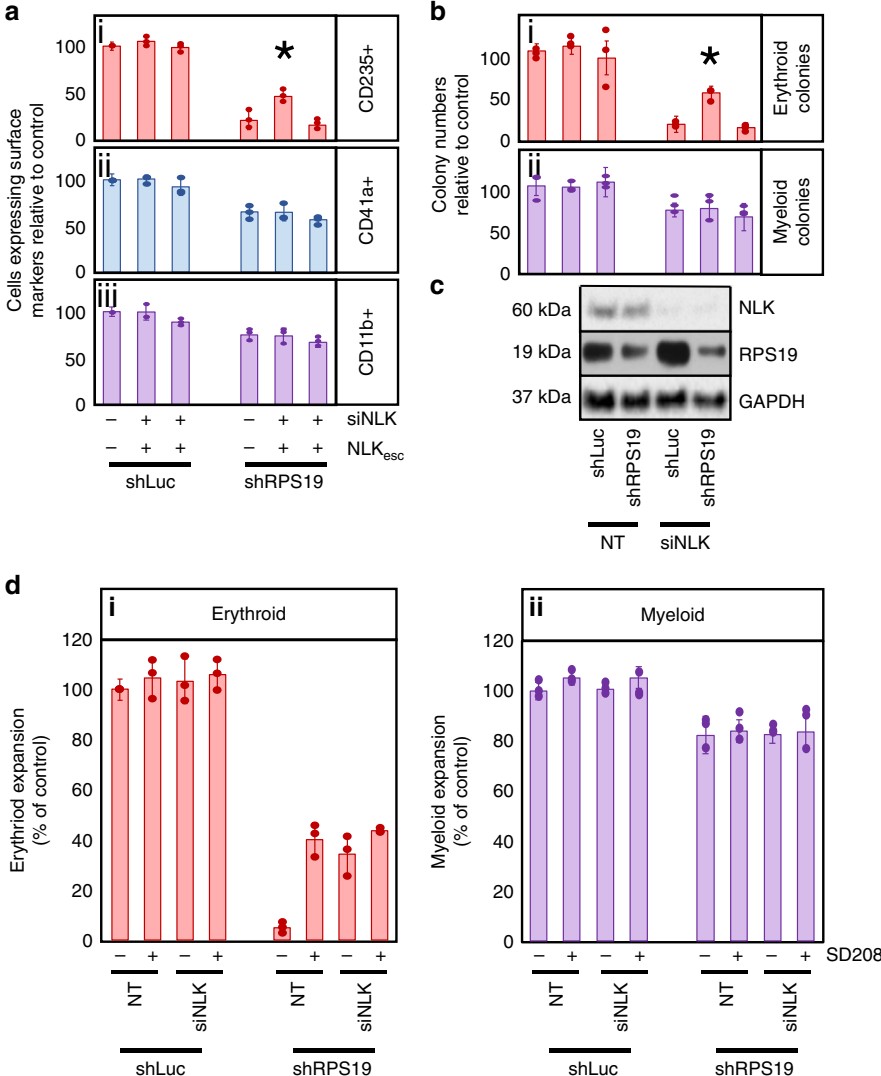

**Fig. 2 NLK expression contributes to erythroid defects in RPS19-insufficiency. a** Cord blood CD34[+] progenitors were transduced with lentivirus expressing shRNA against luciferase (shLuc) or RPS19 (shRPS19) co-expressing GFP, along with siRNA targeting NLK (siNLK) co-expressing RFP and a siNLK-resistant NLK (NLK$_{esc}$) co-expressing puromycin resistance. GFP[+]RFP[+] progenitors were differentiated in erythroid media for 15 days prior to counting and assessment for surface expression of (i) CD235 (erythroid), (ii) CD41a (megakaryocyte) and (iii) CD11b (myeloid) cellular markers by flow cytometry. Within each sample, the total number of cells was multiplied by the percentage of cells expressing each differentiation marker and values were normalized and expressed as a percentage of control (shLuc/NT). **b** Transduced GFP[+],RFP[+] cord blood CD34[+] progenitors were differentiated in methylcellulose for 12–15 days and colonies were scored as either erythroid (i) or myeloid (ii). **c** Expression of NLK, RPS19, and GAPDH were analyzed by Western blot analysis after 5 days of differentiation. **d** CD34[+] progenitors were transduced with shRNA against luciferase or RPS19 and non-targeting or siRNA against NLK. After sorting, samples were split into two groups and either treated with vehicle or SD208 every three days. After 15 days cells were counted and subject to flow cytometry to compare the expansion of maturing CD235[+] erythroid (i) and CD11b[+] myeloid cells (ii). Bars represent means ± SD with individual data points overlaid. $n = 3$ independent experiments performed in triplicate. Statistics: two-tailed Student's $t$ test, significant *$p < 0.05$. Also see Supplementary Fig. 2. Erythroid expansion is depicted in red, megakaryocyte in blue and myeloid in purple. Source data are provided as a Source Data file.

against RPS19 in cord blood (CB) CD34[+] HSPCs and differentiated them in erythroid-promoting media. RPS19-insufficiency reduced maturing CD235[+] erythroblast expansion to 21.2% of controls and silencing NLK improved it to 46.7% ($p = 0.0216$) of ribosome-competent controls (Fig. 2a.i). Non-erythroid myelopoiesis was mildly reduced by RPS19-insufficiency but NLK suppression had a negligible impact ($p = 0.8296$). Likewise, NLK suppression did not affect erythropoiesis or myelopoiesis in ribosome-competent controls ($p = 0.2749$ and 0.9438) (Fig. 2a.ii, iii).

We re-introduced recombinant NLK (including 3′UTR) engineered with a nucleotide sequence unable to be recognized

by the siRNA, as well as using a series of shRNAs against different regions of NLK, to rule out the possibility of off-target effects of NLK siRNA leading to improved erythropoiesis (Supplementary Fig. 2c).

While liquid culture allows us to quantify both immature and mature cell populations, we also assessed the role of NLK in colony assays. Erythroid BFU-E colonies decreased to 18.8% of controls in RPS19-insufficient progenitors. Expression of siRNA against NLK did not affect BFU-E colony numbers in ribosome-competent progenitors ($p = 0.4134$). However, we observed a 2.9-fold increase in erythroid (BFU-E) colonies from 18.8 to 53.6% ($p = 0.0116$) in NLK siRNA treated cells compared with controls

(Fig. 2b.i). NLK siRNA did not affect myeloid CFU-GM colonies ($p = 0.8141$) (Fig. 2b.ii). NLK expression was reduced 80–90% upon expression of siRNA (Fig. 2c). $p$ values were defined by paired Student's $t$ test.

NLK shares a number of conserved regions with cyclin dependent kinases (cdks)[6,43]. The siRNA against NLK was designed not to target other conserved genes, however we examined the impact of the siRNA on expression of kinases with similar substrate profiles by Western blot analysis. No reduction of TAK1, p38, JNK, ERK1/2, Cdk1, or Cdk2 protein was observed upon expression of siRNA against NLK. Mild reductions in p38 (16%), JNK (7%), and ERK1/2 (14%) phosphorylation were observed (Supplementary Fig. 2d).

As observed previously (Figs. 1d, 2a), SD208 treatment alone improved RPS19-insufficient CD235$^+$ erythroblast expansion from 4.9% to 40.3% observed in controls, while siRNA against NLK improved erythropoiesis from 4.9% to 34.2% compared with ribosome-competent controls (Fig. 2d.i). SD208 treatment in RPS19-insufficient erythroid progenitors expressing siRNA against NLK failed to show significant improvement in erythroid expansion over either treatment alone (compare increases from 4.9% to 40.3%, 34.2% and 43.6% for SD208, siNLK and combined, respectively—$p = 0.4633$ and 0.1825, paired Student's $t$ test.) (Fig. 2d.i), suggesting the most relevant target of this compound in ribosomal insufficiency is NLK. No NLK effect was observed in myeloid expansion (Fig. 2d.ii).

**Effect is not through modulation of NLK expression.** Using three different NLK antibodies, we analyzed NLK protein expression by Western blot analysis in CD34$^+$, CD71$^+$ and CD71$^-$ populations (Fig. 3a). CD71 is highly expressed in erythroid progenitors but at lower levels in megakaryocyte and Megakaryocyte/Erythroid Progenitor (MEP) populations[44]. We did not observe differences in NLK expression between control and RPS19-insufficiency in CD71$^+$ or CD71$^-$ populations (Fig. 3a). However, NLK expression was significantly reduced in the CD71$^-$ population relative to the CD71$^+$ and CD34$^+$ HSPC population, suggesting that NLK expression is reduced as cells differentiate along the non-erythroid lineage (Fig. 3a). RPS19-insufficiency did not impact NLK mRNA expression in any lineage, but was much higher in erythroid populations (Fig. 3b).

**NLK Activity is increased in RPS-19 insufficient cells.** NLK has been reported to phosphorylate a number of substrates, including c-Myb[4,13–15,45] and Raptor[16]. Activated NLK is phosphorylated on Thr298[6,46], so we analyzed the phosphorylation status of NLK from control and RPS19-insufficient CD71$^+$ erythroid and CD71$^-$ non-erythroid populations after 10 days of differentiation using a phospho-specific antibody by Western blot analysis. In CD71$^+$ erythroblasts, RPS19-insufficiency increased phosphorylation of NLK at Thr298 2.5-fold of controls (Fig. 3c).

NLK dimerizes upon activation[46]. Corresponding with increased phosphorylation of NLK at Thr298, we observed a 3.9-fold ($p = 0.0124$) increase in dimerization of transduced YFP- and CFP-tagged NLK (as measured by fluorescence resonance energy transfer) in populations expressing shRNA against RPS19 relative to controls (Fig. 3d).

To test NLK activity in small populations of cells, we designed a highly sensitive in vitro kinase assay examining the ability of immunopurified NLK to phosphorylate three known NLK substrates; NLK, c-Myb, and Raptor (see Supplementary Fig. 3a). Corresponding with pThr298-NLK phosphorylation and NLK dimerization, in vitro NLK activity was robustly induced (Fig. 3e. i–iii left panels). CD235$^+$, CD41a$^+$, and CD11b$^+$ are markers of maturing erythroblasts, megakaryocytes, and myeloid

populations respectively, and elevated NLK activity was observed in RPS19-insufficient CD235$^+$ erythroblasts but not CD41$^+$ or CD11b$^+$ populations (Fig. 3e.i–iii right panels). Compared with controls, immunoprecipitated NLK from RPS19-insufficient CD71$^+$ and CD235$^+$ erythroid populations, increased NLK phosphorylation by 2.7- and 3.4-fold ($p = 0.011$ and 0.0025); c-Myb phosphorylation increased by 3.9- and 4.7-fold ($p = 0.0268$ and 0.0175); and Raptor phosphorylation increased by 6.1- and 7.3-fold ($p = 0.0433$ and 0.032), respectively. No significant increase in phosphorylation was detected in other hematopoietic lineages (Fig. 3e) and $p$ values were defined by paired Student's $t$ test. The assay was validated in Kp53A1 myeloid leukemia cells (Supplementary Fig. 3b–d) and specificity of the NLK antibody was also confirmed (Supplementary Fig. 3b, e, f).

We did not see an association of p38 with immunoprecipitated NLK (Supplementary Fig. 3e) however, as an association between the two proteins has been reported[17] we examined this in greater detail. As observed by Ohnishi et al.[17], endogenous p38 could not be detected with immunoprecipitated NLK but NLK could be detected with immunoprecipitated p38 (Supplementary Fig. 4a). This suggests the NLK epitope may be masked by p38 binding and prevent the complex being immunoprecipitated with an NLK antibody, and therefore exclude NLK-associated p38 from our kinase assay.

But as p38 can phosphorylate NLK, Myb and Raptor in vitro, we performed a series of experiments that indicate NLK-associated p38 is not contributing to substrate phosphorylation. We determined the IC$_{50}$ values of a panel of eight p38 inhibitors against p38 and NLK in our in vitro kinase assays. For comparison, reported IC$_{50}$ values from bioassay results submitted to the National Center for Biotechnology Information PubChem Database were tabulated. No compound inhibited NLK with the same specificity as p38. Moreover, for each inhibitor with a reported IC$_{50}$ for NLK, our observed NLK IC$_{50}$ was similar to the reported NLK IC$_{50}$ and differed significantly from that of p38 (Fig. 3f).

The IC$_{50}$ values we obtained for NLK differ from reported values for p38. SD208, SB431542, and Galunisertib inhibit NLK with IC$_{50}$ values of 435 nM, 1.07 μM and 1.34 μM but are reported to inhibit p38 with IC$_{50}$ values of 850 nM, 9 μM and 405 nM, respectively. We observed SD208 inhibited p38 with an IC$_{50}$ value of 910 nM (Supplementary Fig. 4b). This data support our conclusion that these compounds increase erythropoiesis in ribosome insufficiency through the inhibition of NLK and not p38. As it was reported that p38 is required for NLK function in anterior development in *Xenopus*[17], our data suggest the role for p38 in NLK activation is context-dependent. As p38 phosphorylates NLK at a specific serine residue, it is possible redundancy exists between kinases that can phosphorylate this residue under different conditions.

We found no evidence of NLK-associated p38 contributing to the effects of RPS19-insufficiency, however other NLK-associated kinases could not be ruled out. Therefore, in the context of NLK knockdown with siRNA, we re-expressed siRNA-resistant wild-type (WT) or kinase-deficient NLK. No kinase activity was detected after immunoprecipitation of the kinase-deficient mutant, indicating no associated kinases were present (Supplementary Fig. 4c).

**NLK is activated in murine and human models and DBA patients.** Along with RPS19, RPL11 is one of the most commonly mutated genes in DBA[1]. Transduction of CD34$^+$ cells isolated from human cord blood with lentivirus expressing shRNA sequences selected for approximately 50% knockdown efficiency against RPS19 or RPL11 recapitulate erythropoiesis

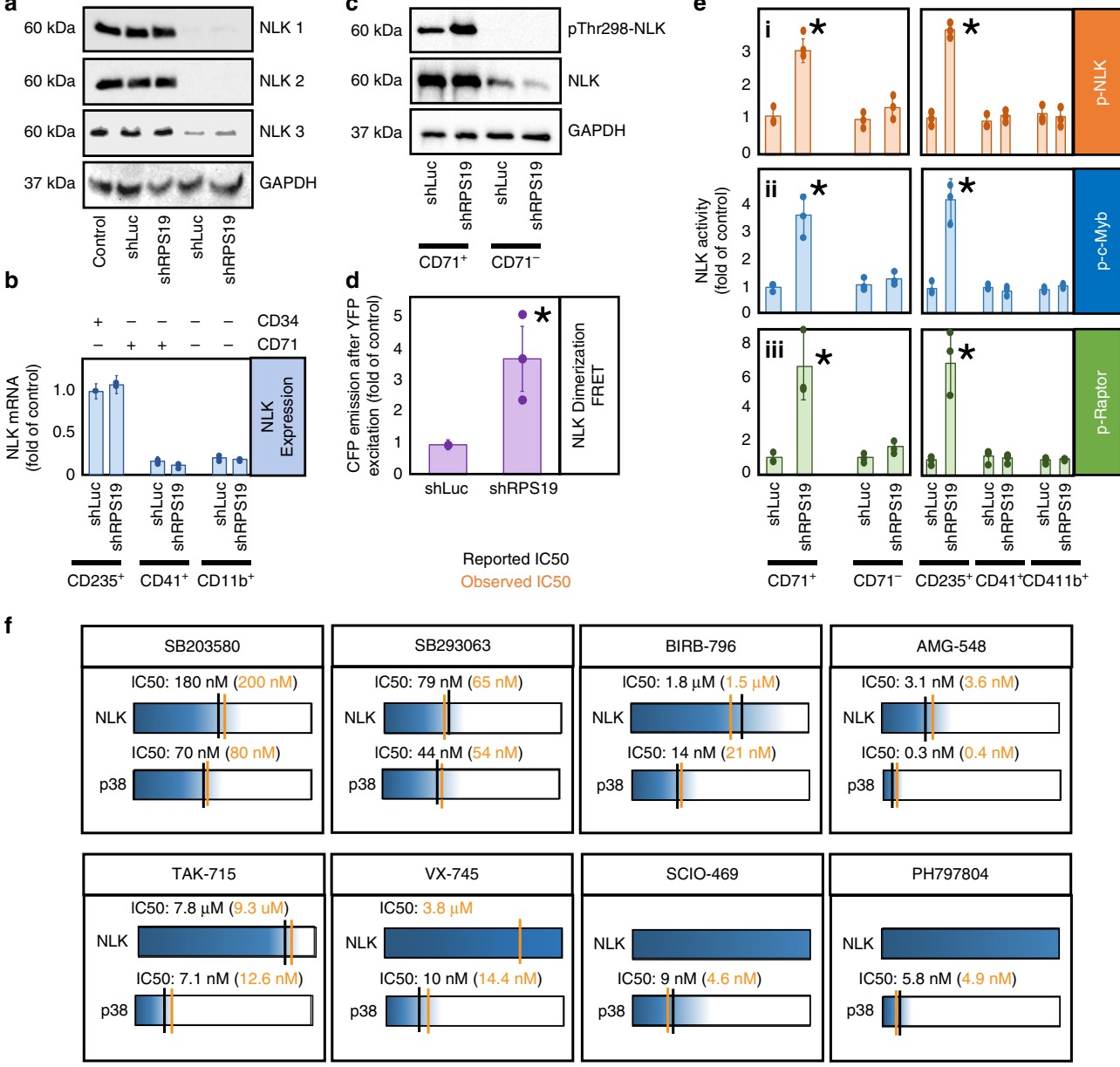

**Fig. 3 NLK expression is higher in erythroid progenitors and is activated in RPS19-insufficiency. a** Transduced CD34+ CB HSPCs were differentiated for 10 days and CD71+ and CD71- fractions were probed by Western blot analysis for NLK using three different NLK antibodies. Equivalent protein was loaded between samples. **b**—blue Transduced progenitors were differentiated for 15 days and sorted for surface expression of CD235 +, CD41 + and CD11b + population NLK mRNA was assessed by qRT-PCR. **c** Control and shRPS19-transduced CB progenitors were differentiated for 10 days prior to separation into CD71+ and CD71- populations and assessed for pThr298-NLK phosphorylation by Western blot analysis. **d**—purple CB CD34 + progenitors were transduced with shRNA against luciferase (shLuc) or RPS19 (shRPS19) along with CFP-NLK and YFP-NLK and differentiated for 10 days. NLK dimerization was quantified by FRET. **e**—left panel NLK was immunoprecipitated from transduced differentiating CD71+ and CD71- populations after 10 days, and incubated in the presence of ATP, Mg$^{2+}$ and dephosphorylated NLK (i orange), c-Myb (ii—blue) and Raptor (iii—green) for 30 min at 37 °C. Phosphorylation was detected by a combination of anti-phosphoserine-HRP and anti-phosphothreonine-HRP antibodies, or mouse anti-phosphoserine and anti-mouse-HRP antibodies. **e**—right panel For comparison, NLK in vitro kinase activity was assessed from CD235+, CD41+ and CD11b+ populations enriched after identical treatments. **f** A panel of eight small molecule p38 inhibitors were titrated into in vitro kinase assays in the presence of activated NLK or p38 from stimulated Kp53A1 cells. The IC$_{50}$ for NLK and p38 for each compound was calculated. The data are represented diagrammatically with IC$_{50}$ values represented as vertical lines along a concentration gradient for NLK and p38. Kinase activity is represented as blue and the extent of inhibition is depicted in white. Our observed values (shown in orange) can be easily compared with documented IC$_{50}$ values (shown in black) for each compound against each kinase. Bars represent means ± SD with individual data points overlaid. $n = 3$ independent experiments performed in triplicate. Statistics: two-tailed Student's $t$ test, significant *$p < 0.05$. Also see Supplementary Figs. 3 and 4. Source data are provided as a Source Data file.

defects of DBA when differentiated in vitro[27]. CB CD34+ cells were transduced with lentivirus expressing GFP and shRNA against RPS19, RPL11, or control (luciferase). Cells were collected each day and assayed for NLK activity against NLK (i),

c-Myb (ii), and Raptor (iii) and expression of NLK (iv), and RPS19 or RPL11 (v). We observed a mild increase in in vitro phosphorylation of all three substrates at day 3 in control cells (Fig. 4a.i–iii), indicating a transient activation of NLK in

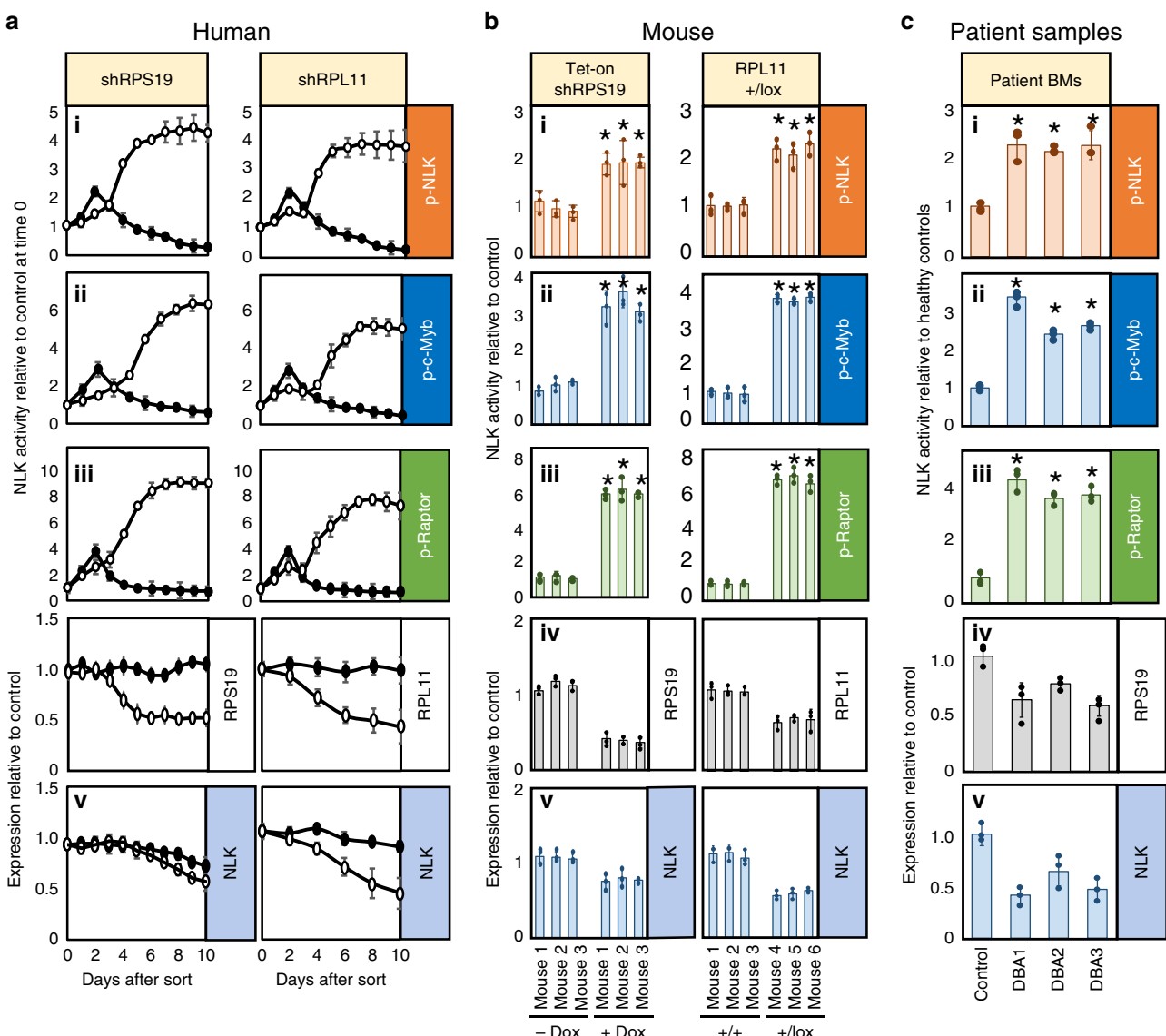

**Fig. 4 NLK is activated in erythroid progenitors from human and murine models of DBA and DBA patient bone marrow. a** Cord blood CD34+ progenitors were transduced with lentivirus co-expressing shRNA against luciferase (shLuc), RPS19 (shRPS19) or RPL11 (shRPL11) and GFP. After 36 h GFP+ cells were differentiated in erythroid media for the indicated days prior to immunopurifying NLK for kinase assay measuring in vitro phosphorylation of NLK (i—orange), c-Myb (ii—blue) and Raptor (iii—green) and assessment of RPS19/RPL11 (iv—gray), and NLK (v—blue) expression by qRT-PCR. Solid circles indicate shLuc while open circles indicate shRPS19 or shRPL11. **b** Lin−Kit+ hematopoietic progenitors were obtained from mouse embryos expressing tetracycline-inducible shRNA against RPS19, at day E14.5. Cells were grown in the presence or absence of doxycycline for 8 days and subjected to NLK kinase assay qRT-PCR for expression of murine RPS19 and NLK (b-left). Lin−Kit+ progenitors were purified from bone marrow of 3 RPL11+/+ and 3 RPL11+/lox tamoxifen-treated mice and analyzed for NLK activity by kinase assay, as well as NLK and RPL11 expression by qRT-PCR (b-right). **c** NLK was immunopurified from 5000 bone marrow mononuclear cells derived from bone marrow aspirates of healthy control and three DBA patients carrying RPS19 mutations. Bars represent means ± SD with individual data points overlaid. $n = 3$ independent experiments performed in triplicate. Statistics: two-tailed Student's $t$ test, significant $*p < 0.05$. Also see Supplementary Fig. 5. Source data are provided as a Source Data file.

control HSPCs during early differentiation. In contrast, a dramatic and sustained increase in NLK activity was induced in RPS19-insufficient cells with NLK, Myb, and Raptor phosphorylation exceeding controls by four-, six-, and nine-fold, respectively (Fig. 4a.i–iii left panels). Activation of NLK in RPL11-insufficiency paralleled RPS19-insufficiency, but was less robust (Fig. 4a.i–iii right panels). It should be noted that NLK activity is being assessed from the entire population of differentiating progenitors (erythroid and non-erythroid combined) so the observed reduction in NLK expression likely reflects a reduced ratio of high NLK-expressing erythroid progenitors relative to low NLK-expressing non-erythroid

progenitors in ribosome-insufficiency (Fig. 4a.v). Sorting of populations into CD71+ and CD71− populations confirmed no difference in NLK expression between control and RPS19-insufficiency within populations (Fig. 3a, b).

Subtle differences in erythroid development have been reported between various sources of CD34+ progenitors[47]. We observed low basal NLK activity in differentiating progenitors derived from healthy control cord blood, fetal liver, and peripheral blood, with significant upregulation of NLK activity in all three progenitor sources during RPS19-insufficiency (Supplementary Fig. 5a).

Despite differences between human and murine hematopoiesis[48], a DBA-like phenotype develops in mice with RPS19 or

RPL11 insufficiency[49,50]. We expanded hematopoietic progenitor Lin⁻Kit⁺ mouse fetal (E14.5) liver cells expressing a tetracycline-inducible shRNA against RPS19. NLK immunoprecipitated from doxycycline-treated cells demonstrated significantly increased activity (1.9-, 3.2-, and 5.3-fold above controls for NLK (i), c-Myb (ii), and Raptor respectively (iii)), (Fig. 4b—left panels).

Similarly, mice expressing a single copy of the RPL11 allele develop anemia[50]. Mice heterozygous for RPL11^flox were treated with tamoxifen for 8 weeks and Lin⁻Kit⁺ hematopoietic progenitors were isolated and NLK activity was increased in all three RPL11-haploinsufficient mice (2.1-, 4.1- and 7.2-fold for NLK (i), c-Myb (ii), and Raptor (iii), respectively) compared with tamoxifen-treated wild type mice (Fig. 4b—right panels). Collectively, these data indicate that NLK is activated in HPSCs from murine models of DBA.

NLK activity was also higher in DBA patient cells than healthy controls. NLK immunopurified from mononuclear cells from the bone marrow aspirates of three DBA patients carrying different RPS19 mutations significantly increased phosphorylation of all three substrates compared with healthy control cells (2.2-, 2.0-, and 2.2-fold NLK phosphorylation (i), 3.3-, 2.4-, and 2.6-fold c-Myb phosphorylation (ii), and 4.8-, 4.1-, and 4.2-fold Raptor phosphorylation (iii)). As expected, reduced NLK expression was observed in bone marrow mononuclear cells from 3 DBA patient samples, reflecting the increased ratio of low NLK-expressing non-erythroid progenitors in the heterogeneous population (Fig. 4c.iv). RPS19 expression was significantly lower in bone marrow cells from DBA patients (0.6, 0.8 and 0.6-fold) compared with control cells (Fig. 4c.v).

Induced-pluripotent stem cells (iPSCs) generated from a DBA patient carrying a mutation in RPS26 (C74A) robustly induced NLK activity. Activation was initiated during early differentiation and was sustained throughout the 10-day time course. Expression of a tetracycline-inducible WT RPS26 significantly reduced NLK activity (Supplementary Fig. 5b). Western blot analysis confirmed a 1.95 fold increase ($p = 0.0261$, paired Student's $t$ test.) in NLK phosphorylation at Thr298 during ribosome-insufficiency in DBA-patient derived iPSCs compared with those with the causal genetic mutation corrected (Supplementary Fig. 5c).

**NLK expression limited in non-erythroid progenitors.** As NLK lacks the TXY motif in the activation loop of other MAPK members, it has been proposed that NLK is constitutively active and over expression alone increases cellular activity levels[17,43,46]. However upstream signaling regulators increase kinase activity without modulating expression[13–16,18–21]. Over expression of NLK cDNA (without NLK 3′UTR) did not appreciably decrease CD235⁺ erythroid differentiation ($p = 0.2249$ and 0.2474, respectively) in control or RPS19-insufficiency (Fig. 5a.i). Similar to erythroid expansion in control cells, over-expressed NLK cDNA did not impact myeloid expansion ($p = 0.0489$). Unexpectedly, NLK cDNA expression resulted in a dramatic reduction ($p = 0.001$) in myeloid expansion (Fig. 5a.ii). P values were defined by paired Student's $t$ test. Consistent with Fig. 2a/b, expression of NLK cDNA with an intact 3′UTR mimicked the endogenous condition with no impact on myeloid expansion (Fig. 5a.ii). These data are consistent with a model in which NLK is not constitutively active and is induced during ribosome insufficiency, but does not explain why NLK without a 3′UTR impacts myelopoiesis while endogenous or recombinant NLK with an intact 3′UTR does not.

Compared with erythroblasts, endogenous NLK expression is significantly reduced in myeloid cells (Fig. 3a–c). NLK expression is highly sensitive to miRNAs[6,22,24,51] that bind to the NLK 3′untranslated region (3′UTR) leading to transcript degradation[6,24,51] Fitting with the NLK 3′UTR suppressing NLK expression in non-erythroid progenitors, fusion of the NLK 3′UTR to the luciferase gene reduced luciferase activity by ~70% in CD41a⁺ megakaryocyte and CD11b⁺ myeloid populations, compared with luciferase activity in CD235⁺ erythroid progenitors (Fig. 5b.i). Luciferase activity was not differentially influenced by the NLK 5′UTR across the different lineages (Fig. 5b.ii).

Downregulation of NLK by miR-181 binding to the 3′UTR has been reported previously[22,23]. Deregulation of miR-181 significantly alters the ratio of lymphoid and myeloid lineages during differentiation[51–55] and is often downregulated in leukemia[51,52,54]. For example, the expression of miR-181 in MEPs determines megakaryocyte versus erythroid commitment[55]. Consequently, we were precluded from direct inhibition of miR-181 in differentiating progenitors. However, we utilized three alternative strategies to determine if miR-181 is involved in regulating NLK expression.

First, since miR-181 induction is critical for lymphoid differentiation[51,53], we anticipated miR-181 expression would be high in the LCL line (DR07). MiR-181 was expressed 8.6 times higher than in CD34⁺ HSPCs. In contrast, NLK expression was 3.5-fold lower (Supplementary Fig 6a). We also expressed a miR-181 sponge that sequesters free miR-181[56] in LCLs from a healthy control (DR07), control transduced with shRNA against RPS19 (DR07 + shRPS19), and three DBA patients (DG0005, DG0006, and DG0079). No significant differences were observed in miR-181 between cell lines with or without miR-181 inhibitor (Supplementary Fig 6b). In contrast, NLK expression was increased between 3.5- and 4.2-fold in all lines upon miR-181 inhibition (Supplementary Fig. 6c). Interestingly, miR-181 inhibition did not impact proliferation in control cells, but reduced proliferation between 46 and 50% in RPS19-insufficient cells (Supplementary Fig. 6d). Although LCLs do not differentiate and are not of erythroid origin, these data indicate NLK expression is inversely correlated with miR-181 and that increasing NLK expression in these cells sensitizes LCLs to ribosomal insufficiency.

Second, although unable to inhibit miR-181 in HSPCs due to the multiple NLK-independent roles of this miRNA in hematopoiesis, we sought to determine if miR-181 was regulating NLK expression by mutating the potential miR-181 binding site in the 3′UTR. Expression of miR-181 was between 6.2- and 6.6-fold higher in non-erythroid cells than erythroblasts but neither RPS19-insuffiiency nor expression of various NLK constructs influenced miR-181 expression (Fig. 5c.i).

As observed in Fig. 3b, NLK was expressed 4–5-fold more in erythroid than non-erythroid cells. Unexpectedly, expression of all recombinant NLK mutants did not dramatically increase already high levels of NLK expression in erythroblasts (Fig. 5c.ii). However, NLK with absent or mutated 3′UTR both increased NLK expression over threefold in non-erythroid lineages, reaching between 75 and 85% of levels expressed in erythroblasts. Transduction of recombinant NLK with an intact 3′UTR failed to significantly increase NLK expression in non-erythroid cells (Fig. 5c.ii).

These data indicate that the miR-181 binding site within the 3′UTR is required for suppression of NLK in non-erythroid cells, but the consequences of "forced" NLK expression in non-erythroid cells was pronounced. In agreement with Fig. 5a, "forced" expression of NLK in non-erythroid progenitors did not influence lineage expansion unless NLK could be activated (Fig. 5c.iii). Few megakaryocytes and <30% of control myeloid populations were recovered from RPS19-insufficient populations expressing NLK with no or mutated 3′UTR (Fig. 5c.iii). Erythropoiesis was only mildly affected by modulation of NLK expression.

Third, the role of the miR181-binding site in the NLK 3′UTR was confirmed by mutating the site using CRISPR-Cas9 in primary

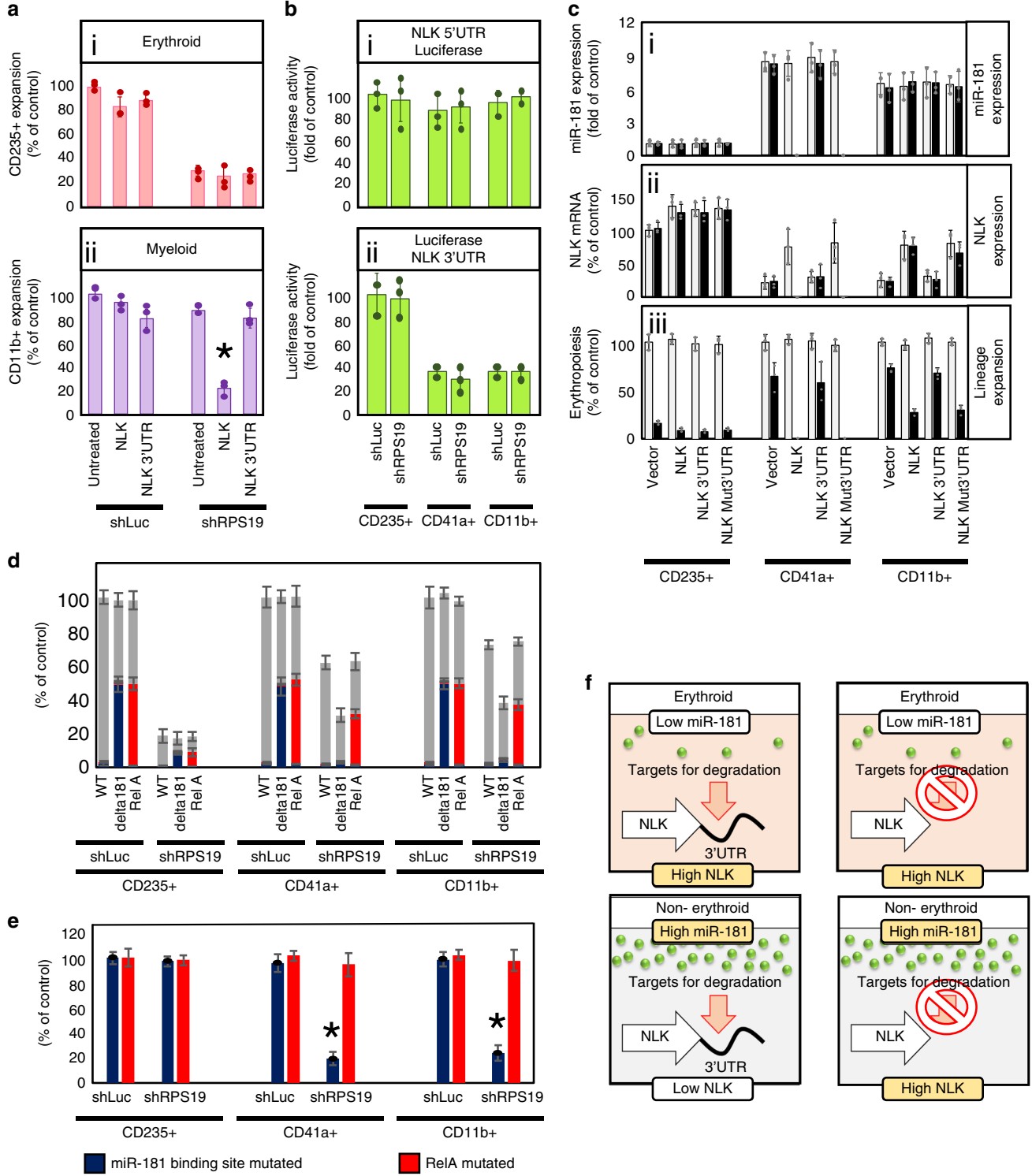

human hematopoietic cells. After screening guide RNAs (gRNAs) that selectively target the potential miR-181 binding site (see Supplementary Fig. 6e), gRNA, along with Cas9, were electroporated into control and RPS19-insufficient CD34+ HSPCs. As a control, gRNA targeting RelA was introduced. Cells were differentiated prior to counting, sorting and sequencing to determine the percentage of cells baring mutations in NLK 3′UTR (a schematic of the experiment is presented in Supplementary Fig. 6f).

Subsequently, CD235+, CD41a+, and CD11b+ numbers were calculated (Fig. 5d gray bars). After sequencing, the percentage of cells baring mutations in NLK 3′UTR (Fig. 5e blue bars) and RelA

(Fig. 5e red bars) was determined. In ribosome-competent cells in excess of 40% of cells were successfully mutated by both NLK 3′UTR- and RelA-targeting in all three cell lineages. Even with 40% cells carrying mutations, no reduction in cell population was observed (Fig. 5d).

A different pattern was observed in RPS19-insufficient progenitor cells. The percentage of cells with NLK 3′UTR or RelA mutations was not significantly different from controls (Fig. 5d blue and red bars) and the mutations do not impact the total cell population (Fig. 5d gray bars). In RPS19-insufficient CD41a+ megakaryocyte and CD11b+ myeloid populations (Fig. 5d gray bars) a significant

**Fig. 5 Upregulation of miR181 suppresses NLK expression in non-erythroid progenitors and protects cells from RPS19-insufficiency. a** Wild type NLK cDNA was generated with or without the NLK 3′UTR. Cord blood CD34 + HSPCs were transduced with NLK construct and siRNA and differentiated for 12 days. Erythroid (i—red) and Myeloid (ii—purple) expansion was calculated by multiplying viable cell counts by % CD235+ or CD11b+, respectively. Results are displayed as a percentage of cells of each lineage relative to untreated controls. **b**—green CD34+ progenitors were transduced with shRNA against a non-targeting sequence (NT) or RPS19 (shRPS19) and the NLK minimal promoter upstream of the luciferase gene (i) or luciferase gene with the NLK 3′UTR downstream (ii). After 12 days of differentiation in erythroid media, cultures were sorted into designated hematopoietic lineages and luciferase activity was assessed. **c** CD34+ progenitors were transduced with shLuc (white bars) or shRPS19 (dark bars). In addition, vectors expressing cDNA for NLK with either a 3′UTR with 3 stop sequences (NLK stop), the wild type NLK 3′UTR (NLK WT 3′UTR) or 3′UTR with the miR181 binding site mutated (NLK Mut 3′UTR). After differentiation, cells were sorted into hematopoietic lineage and assessed for miR181 expression (i) or NLK expression (iii) by qRT-PCR. The relative number of each hematopoietic lineage was determined as a percentage of the number observed in controls expressing only endogenous NLK (iii). **d** After counting and sorting for lineage (gray bars), cells were sequenced to determine the percentage of cells within each population carrying indels in miR181 binding site (blue) and RelA (red). **e** The percentage of cells carrying indels in NLK 3′UTR miR181 binding site (blue) and RelA (red) in each treatment were further compared for impact on population expansion between lineages. **f** A diagrammatic representation of the induction of miR-181 in non-erythroid progenitors leads to binding to the NLK 3′UTR to facilitate degradation of the NLK transcript. Bars represent means ± SD with individual data points overlaid. n = 3 independent experiments performed in triplicate. Statistics: two-tailed Student's t test, significant *p < 0.05.Also see Supplementary Fig. 6. Source data are provided as a Source Data file.

reduction in total population in observed in populations with NLK 3′UTR mutations introduced and less than 8% of these populations carried NLK 3′UTR mutations (Fig. 5d blue bars). This indicates that RPS19-insufficient cells with the mutation were unable to expand (Fig. 5d red bars). As observed in Fig. 4c, mutation of the NLK 3′UTR had no impact on already highly sensitive erythroid differentiation. However CD41a+ megakaryocyte and CD11b+ myeloid populations were 80.4% and 75.1%, respectively, more sensitive to RPS19-insufficiency (Fig. 5e). These data demonstrate that miR-181 suppression of NLK expression protects non-erythroid progenitors from lineage expansion defects resulting from ribosomal insufficiency-induced activation of NLK (see Fig. 5f).

**NLK activation is dependent on p53 activation**. Ribosome-insufficiency induces stabilization of p53 protein and contributes to DBA pathogenesis[27,57–59]. We transduced shRNA against p53 and RPS19 in differentiating HSPCs and observed a greatly reduced activation of NLK compared with RPS19-insufficient cultures without shRNA against p53 (Fig. 6a.i–iii). To further examine the role of p53 in NLK activation in our model and DBA patient samples, we utilized Nutlin-3. Nutlin-3 inhibits the E3 ubiquitin ligase MDM2 binding of p53 and subsequently stabilizes p53 from ubiquitin-dependent degradation[59]. NLK dimerizes upon activation[46] and the detection of FRET from dimerized YFP- and CFP-NLK pairs correlates with NLK activity in RPS19-insufficiency (Fig. 3d). Nutlin-3 increased FRET signaling from 9.0% (Fig. 6b.i) to 50.2% (Fig. 6b.ii) of the population. We gated p53 expression into p53$^{low}$ in which 6.6% of cells activate NLK, and p53$^{hi}$ where 91.9% of cells activate NLK (Fig. 6b.ii). RPS19-insufficiency increased the p53$^{hi}$ population from 8.1 to 34.1%, corresponding to an increase from 9.0 to 36.1% of cells with elevated NLK-dimerization (Fig. 6b.iii).

Next we compared healthy mononuclear bone marrow cells stimulated with (i) vehicle control, (ii) Nutlin-3, (iii) or to a DBA sample pooled from three patients (Fig. 6c). Nutlin-3 again increased the percentage of p53$^{hi}$ cells (from 6.6% to 31.7%), corresponding to an increase of 9.4% to 30.1% of cells with dimerizing NLK (Fig. 6c.ii). Compared with controls, DBA samples demonstrated a modestly higher percentage of cells with elevated p53 levels (6.6–12.8%) which correlated with higher (9.4–14.4%) NLK dimerization (Fig. 6c.iii). NLK in vitro kinase assay was performed and confirmed a correlation between NLK dimerization and activity (Supplementary Fig. 7a). Comparison of values between kinase assay and flow cytometry values are tabulated in Fig. 6d.

In Kp53A1 cells, robust activation of NLK was observed at temperatures supporting p53 activation (Supplementary Fig. 3b–e). Taken together, these data demonstrate that activation of NLK is p53-dependent.

**RPS19-insufficiency increases NLK activation in MEPs**. NLK is transiently and modestly activated in normal erythropoiesis but becomes chronically activated in ribosomal insufficiency (Fig. 4a). NLK is activated in CD71+ and CD235+ erythroblasts (Figs. 3–6) but we sought to determine the stage of differentiation NLK was activated in ribosome insufficiency. As reported[60], Lin-Kit+ progenitors from RPL11$^{+/+}$ and RPL11$^{+/flox}$ mice were sorted into Sca+ HSC and non-committed progenitor populations while Sca$^-$ cells were further sorted into CD34-CD16/32$^-$ MEPs, CD34+CD16/32+ Granulocyte/Macrophage Progenitors (GMPs) and CD34+CD16/32$^-$ Common Myeloid Progenitors (CMPs). Low NLK activity was observed in Lin$^-$Kit+Sca+ and CMP populations from both control and RPL11$^-$insufficient mice. A mild increase in NLK activity was observed in RPL11$^{+/+}$ MEP and GMP populations. A 6.3-fold increase in NLK activity was observed comparing MEPs from ribosome-insufficient mice to controls (Fig. 6e) while no NLK activity was observed in GMPs from either group.

We also sorted transduced human CB progenitors into four distinct progenitor populations after 7 days of differentiation (Supplementary Fig. 7b)[61]. Non-erythroid progenitor population (CD71$^-$CD235$^-$), a population enriched for MEPs and BFU-Es (CD71$^{low}$CD235$^-$), a population enriched for CFU-Es (CD71+ CD235$^-$), and a population enriched in pro-erythroblasts and intermediate erythroblasts (CD71+CD235+). Ribosome-insufficiency does not induce NLK activation in non-erythroid progenitors but NLK activity is over 6-fold above control in all three erythroid progenitor and precursor populations (Fig. 6f).

**Effects of NLK phosphorylation of c-Myb and Raptor**. NLK has many reported downstream effectors, including transcription factors involved in erythropoiesis[16,18,21,62], with the potential to impact DBA pathogenesis. While this study is not designed to determine all factors downstream of NLK contributing to the DBA phenotype, we have determined that both Raptor and c-Myb serve as substrates in vitro (Figs. 3–6) and wanted to determine if they are phosphorylated by NLK in ribosome-insufficiency. In osmotic stress, NLK phosphorylates Raptor at Ser863[16]. Control and RPS19-insufficient progenitors derived from transduced CD34+ HSPCs were differentiated for 10 days and sorted into CD71+ and CD71$^-$ populations, prior to being assessed for Raptor phosphorylation at Ser863 by Western blot analysis. As with NLK activation (Fig. 3c), phosphorylation was primarily restricted to RPS19-insufficiency in CD71+ erythroblasts (Fig. 7a). Raptor expression was equivalent across samples (Fig. 7a) and NLK, pThr298-NLK, and GAPDH expression are documented in Fig. 3c.

Raptor phosphorylation can influence mTOR activity by preventing mTOR-associated Raptor from localizing to the

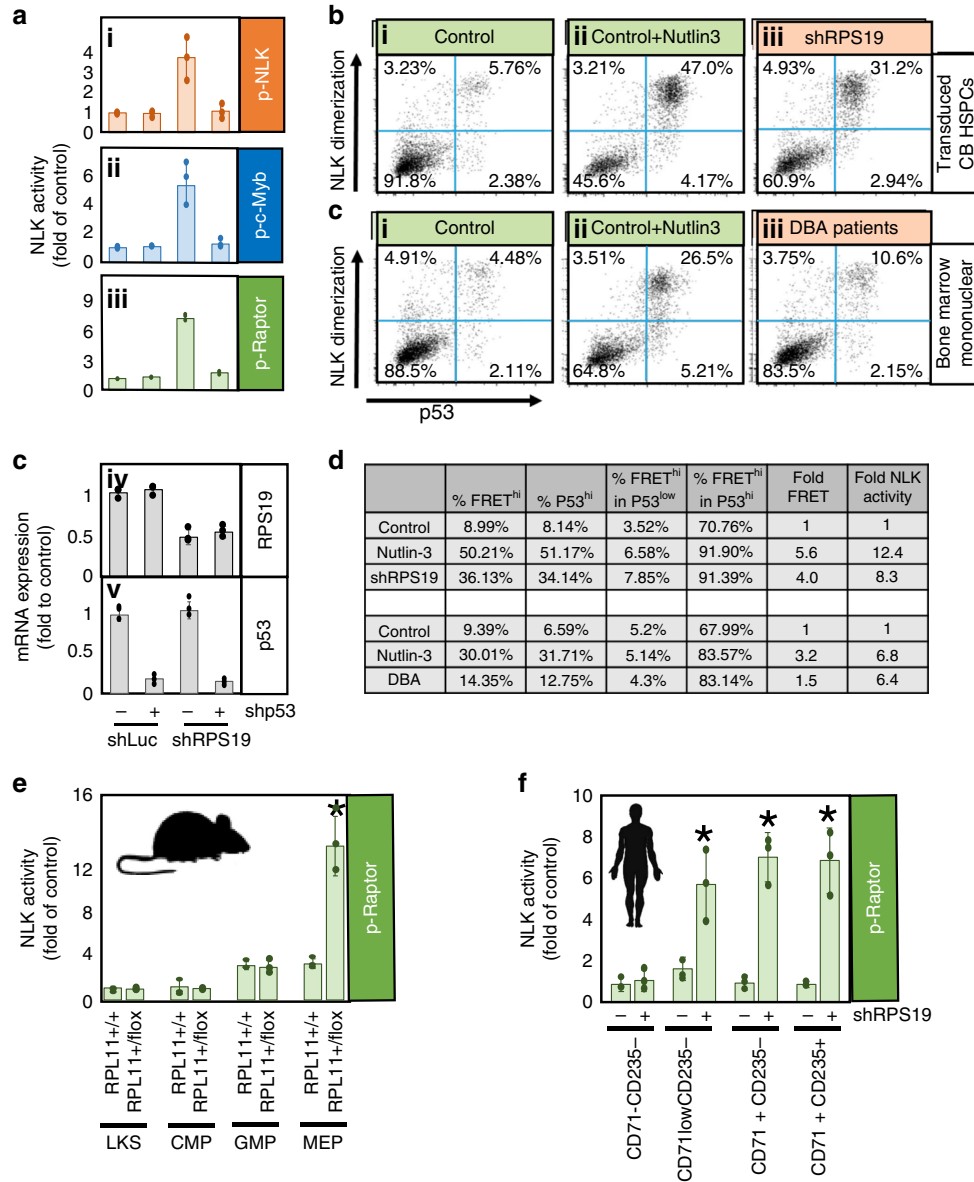

**Fig. 6 NLK is activated in early erythroid progenitors and requires P53 stabilization in RPS19-insufficiency. a** Cord blood CD34$^+$ progenitors were transduced with lentivirus expressing shRNA against luciferase (shLuc) or RPS19 (shRPS19) co-expressing GFP. Cells were also transduced with or without shRNA against p53 co-expressing mCherry. After 36 h, GFP$^+$ and GFP$^+$,mCherry$^+$ cells were differentiated in erythroid media for 8 days, followed by NLK kinase assay (i—NLK: orange, ii—Myb: blue, iii—Raptor: green) and qRT-PCR analysis of RPS19 (iv) and p53 (v) expression. **b** CD34+ cord blood HSPCs were transduced with control shRNA (shLuc) along with YFP- and CFP-tagged NLK and allowed to differentiate for 3 days, in the presence (i) or absence (ii) of Nutlin-3. Cells were stained for p53 and both p53 and FRET was measured by flow cytometry. For comparison, HSPCs were transduced with shRNA against RPS19 (iii). **c** Bone marrow mononuclear cells from healthy donors were transduced with YFP- and CFP-NLK and incubated alone (i), or with Nutlin-3 (ii) for 24 h, prior to p53 and FRET analysis. Bone marrow mononuclear cells from three DBA patients were pooled and analyzed simultaneously (iii). **d** Documentation of the percentage of cells with dimerizing NLK in p53$^{hi}$ and p53$^{low}$ is tabulated and can be compared with NLK in vitro kinase activity. **e**—green Samples were gated to include non-erythroid CD71$^-$CD235$^-$ populations, MEP- and BFU-E-enriched CD71$^{low}$CD235$^-$ populations, CFU-E-enriched CD71$^{hi}$CD235$^-$ populations and proerythroblast and intermediate erythroblast CD71$^{hi}$CD235$^+$ populations. Sorted populations were lysed and immunoprecipitated NLK was subjected to in vitro kinase assay examining Raptor phosphorylation as a substrate. Activity was normalized to the non-erythroid control samples. **f**—green Sorted populations were lysed and immunoprecipitated NLK was subjected to in vitro kinase assay examining Raptor phosphorylation as a substrate. Activity was normalized to the Lin$^-$Kit$^+$Sca$^+$ RPL11$^{+/+}$ samples. Bars represent means ± SD with individual data points overlaid. $n = 3$ independent experiments performed in triplicate. Statistics: two-tailed Student's $t$ test, significant *$p < 0.05$. Also see Supplementary Fig. 7. Source data are provided as a Source Data file.

lysosomal membrane[16]. Strong co-localization (depicted in yellow) of Raptor (red) with the lysosome (green) was observed by fluorescence confocal microscopy in CD71$^+$CD235$^-$ differentiating progenitors in control cells (shLuc) expressing non-targeting or siRNA against NLK (Fig. 7b). Co-localization was disrupted in RPS19-insufficiency but restored when NLK was silenced (Fig. 7b), indicating NLK-dependent phosphorylation and mis-localization of Raptor in RPS19-insufficiency.

NLK-dependent phosphorylation of c-Myb in breast cancer and Wnt signaling has been demonstrated to facilitate rapid ubiquitination and subsequent proteasome degradation[4,14,15,24]. Reduced c-Myb expression has been reported in DBA[5,63] and is

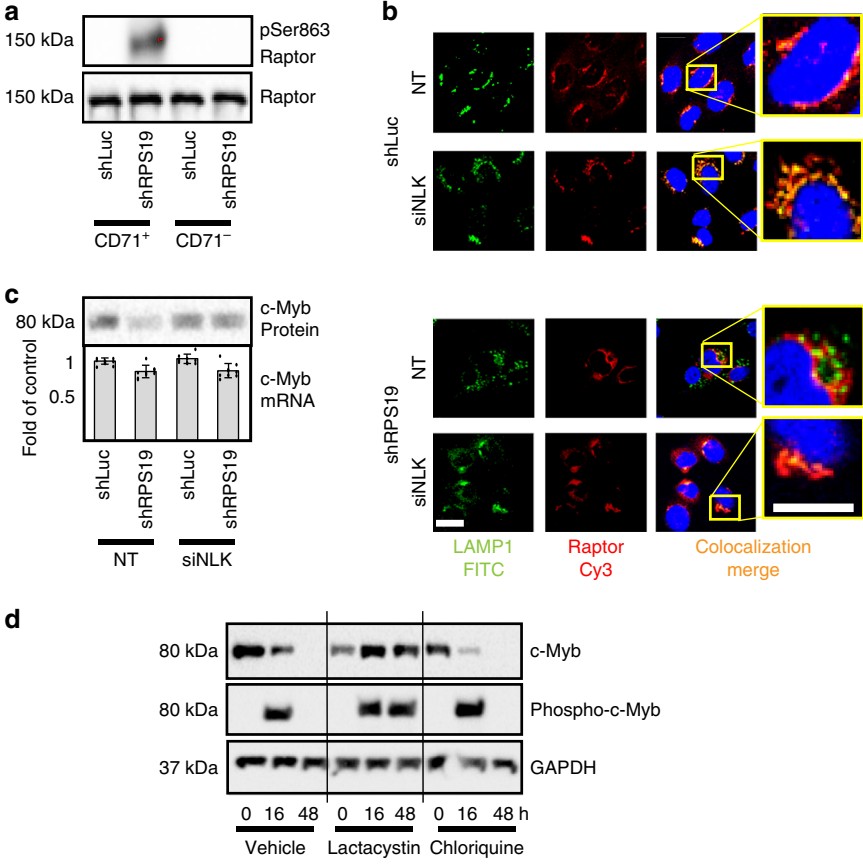

**Fig. 7 NLK phosphorylation of c-Myb results in ubiquitination and proteasome degradation. a** CD34+ CB HSPCs transduced with shRNA against control (shLuc) or RPS19 (shRPS19) were differentiated for 10 days and sorted into CD71+ and CD71− fractions. Three independent experiments were pooled and populations were probed by Western blot analysis examining phospho-Ser863-Raptor and total Raptor. This experiment was performed in conjunction with Fig. 3c. **b** Cord blood CD34+ progenitors were transduced with shRNA against luciferase (shLuc) or RPS19 (shRPS19) and siRNA against NLK (siNLK) or a non-targeting (NT) sequence and differentiated for 8 days. Cells were fixed, permeablized and incubated with Cy3-labeled antibody against Raptor (pseudo-colored red) and lysosomes were visualized by incubation with FITC-labeled antibody recognizing LAMP1 (pseudo-colored green). Areas of co-localization (merge) are indicated in yellow and yellow dotted boxes are magnified in inserts to the far right. Scale bar, 15 μm **c** Fetal liver CD34+ progenitors were transduced with shRNA against luciferase (shLuc) or RPS19 (shRPS19) in conjunction with siRNA against NLK (siNLK) or a non-targeting sequence (NT). Cells were differentiated for 5 days and split into two aliquots. One aliquot was lysed and probed for c-Myb protein expression by Western blot, while the second aliquot was subjected to qRT-PCR to examine c-Myb mRNA expression. **d** Kp53A1 cells were treated with vehicle alone, lactacystin or chloroquine for 30 min, prior to switching cells from 37 to 32 °C for the indicated times. Cells were lysed, normalized for protein and split into two. C-Myb was immunoprecipitated from one sample before Western blot analysis for phosphorylated serine, while the other sample was subjected to Western blot for c-Myb and GAPDH. Bars represent means ± SD with individual data points overlaid. n = 3 independent experiments performed in triplicate. Also see Supplementary Fig. 7. Source data are provided as a Source Data file.

reduced by over 80% in RPS19-insufficient erythroid progenitors relative to controls, when differentiated in vitro. Expression of siRNA against NLK prevents c-Myb degradation in RPS19-insufficiency (Fig. 7c). Quantitative RT-PCR revealed c-Myb mRNA in RPS19-insufficiency was only mildly downregulated (13% reduction) compared with controls (Fig. 7c) which would be anticipated if c-Myb was being degraded after NLK-mediated phosphorylation. Myb serves a number of cellular roles in erythropoiesis, including transcriptional upregulation of KLF1 and LMO2[64]. In controls we observed inductions of 1.8-fold and 2.4-fold of LMO2 and KLF1, respectively, but RPS19-insufficient cultures displayed only 1.3- and 1.8-fold inductions. Upon silencing of NLK, LMO2, and KLF1 induction was restored to 1.6-fold and 2.3-fold, respectively (Supplementary Fig. 7c).

We sought to determine if NLK-mediated phosphorylation was responsible for proteasomal degradation during RPS19-insufficiency, Kp53A1 cells cultured at 32 °C activate p53 and NLK (Supplementary Fig. 3b–e), c-Myb protein is reduced and silencing of NLK prevents c-Myb loss (Supplementary Fig. 7d). C-Myb is undetectable after 48 h of NLK activation so we examined c-Myb expression and phosphorylation at 0, 16, and 48 h after stimulation. To determine if the lysosome or proteasome were involved in the degradation of c-Myb, we also treated cells with chloroquine or lactacystin. Despite a reduction in c-Myb expression, c-Myb phosphorylation was readily detectable after 16 h of NLK activity (Fig. 7d). Expression and phosphorylation of c-Myb was not influenced by chloroquine but both were sustained out to 48 h with lactacystin treatment (Fig. 7d). Our data indicate both Raptor and c-Myb are phosphorylated downstream of NLK activation in RPS19-insufficiency.

**Chemical inhibition of NLK enhances erythropoiesis in DBA.** For direct comparison of the effects of SD208 on NLK activity and erythropoiesis between murine and human DBA models, we isolated CD34+ (human) and Lin−Kit+ (murine) HSPCs from control and disease states and differentiated them in the presence or absence of 5 μM SD208.

SD208 did not significantly impact erythroid expansion of human HSPCs transduced with control shRNA (100% to ~105%), however SD208 increased erythroid expansion in both RPS19- and RPL11-insufficiency from 8.7% to 38.7% ($p = 0.0056$) and 5.8% to 36.7% ($p = 0.0135$) of control, respectively (Fig. 8a).

In mice, Lin$^-$Kit$^+$ cells from three mice expressing a tetracycline-inducible shRNA against RPS19 were differentiated for 8 days in the absence or presence of doxycycline and/or SD208. RPS19-insufficiency reduced Ter119$^+$ erythroblasts to 24.3, 20.3, and 18.5% of controls differentiated in the absence of doxycycline. The presence of SD208 with doxycycline improved erythropoiesis to 83.2, 81.1, and 76.6% of controls ($p = 0.0011$) (Fig. 8b).

Three mature RPL11$^{+/lox}$ mice were treated with tamoxifen for eight weeks to induce heterozygous deletion of RPL11. As controls, three different RPL11$^{+/+}$ littermates of the same mouse strain were treated with tamoxifen. Lin$^-$Kit$^+$ HSPCs were isolated and differentiated in the presence or absence of SD208 for 8 days. As observed in other DBA models, RPL11-insufficiency reduced erythropoiesis but SD208 improved erythropoiesis from 31.2% to 57.3%, 26.4% to 56.8% and 18.6% to 50.3% of controls ($p = 0.0059$) (Fig. 8b).

NLK phosphorylation at Thr298 was elevated in RPS19-insufficient differentiating human erythroblasts 2.5-fold above controls using conventional Western blot analysis (Fig. 3c). Capillary electrophoresis using the Peggy-Sue™ platform (Protein Simple©), revealed a similar 2.3-fold increase in NLK phosphorylation at Thr298 in RPS19-insufficiency, while RPL11-insufficiency revealed a 1.96-fold increase (Fig. 8c). SD208 reduced NLK activity from 2.3-fold to 1.3-fold ($p = 0.025$) above controls in RPS19-insufficiency and 1.96-fold to 1.2-fold ($p = 0.0431$) in RPL11-insufficiency. Parallel results were obtained by analysis of in vitro NLK activity (Supplementary Fig. 8a).

In HPSCs from DBA mouse models, detection of NLK Thr298 phosphorylation by capillary electrophoresis increased 3.1-fold and 2.5-fold over controls in RPS19- and RPL11-insufficiency respectively. SD208 reduced this to 1.3-fold ($p = 0.0027$) and 1.3-fold ($p = 0.0393$) of controls (Fig. 8d) and correlated with in vitro NLK activity (Supplementary Fig. 8b).

**Increased erythropoiesis in SD208-treated DBA BM MNCs.** HSPCs isolated from three DBA patients or healthy controls were cultured in the presence or absence of 5 μM SD208. In the absence of SD208, differentiation of CD235$^+$ erythroblasts from DBA patient HSPCs was reduced to 23.0, 11.6, and 8.4% of healthy controls. The presence of SD208 increased this to 43.7, 24.7, and 19.4% ($p = 0.1806$) (Fig. 8e). This correlates to a mean increase of 2.11-fold. Capillary electrophoresis revealed a 1.68-fold ($p = 0.0487$) increase in NLK phosphorylation at Thr298 in DBA patient samples relative to healthy controls (Fig. 8f). SD208 reduced phosphorylation to 1.11-fold ($p = 0.0253$) of controls. Corresponding decreases were observed in immunopurified NLK (Supplementary Fig. 8c). All P values were defined by paired Student's $t$ test. Collectively, SD208 reduced NLK activity and improved erythropoiesis in all models examined, indicating that targeting NLK activation with small molecules has the potential of improving erythropoiesis in DBA patients.

## Discussion
Advances in DBA therapy have greatly improved life expectancy and outcomes[1,57,65]. Stem cell transplantation can cure the hematological manifestations of the disease, but is associated with a significant risk of life threatening complications such as graft versus host disease[1,57]. Iron overload and other complications are associated with chronic red blood cell transfusions[1,57]. New

targets and therapies are needed to improve patient outcome and quality of life.

Kinases, are highly druggable targets[66]. DBA is a genetic disease, but causal mutations can exist within one of at least 22 genes, so identification of aberrantly regulated kinases common to all genetic variants offers particular promise. Our study documents sustained activation of NLK in human and murine models of DBA with mutations in different ribosomal subunits and chemical inhibition of NLK improved erythropoiesis in all of the models examined.

Systemic inhibition of NLK will not only suppress NLK in erythroid progenitors, but also every other cell type exposed to it. We demonstrated NLK is minimally expressed in non-erythroid hematopoietic lineages and SD208 treatment did not impact differentiation in these lineages in either control or ribosome-insufficient cells. Beyond hematopoietic cells, NLK is modestly expressed in most tissues with highest expression in the brain[6]. NLK knockout mice that are born display smaller size, pulmonary[67] and some neurological defects[68]. Therefore, systemic inhibition of NLK may have adverse effects, particularly in the vascular and nervous systems. Aberrant stromal development in the bone marrow was also observed in NLK null mice[68], which could impact erythropoiesis in DBA patients.

Despite differences in mechanisms regulating erythroid production in humans and mice, DBA-like phenotypes are observed in both species in response to ribosomal insufficiency[49,50], suggesting a conserved molecular mechanism. Although our data demonstrate that small molecules are capable of inhibiting NLK activity, these compounds are not yet ready for clinical application. Our observation that NLK activation is conserved and contributes to erythroid defects highlights the relevance of using these model systems to study DBA and develop novel approaches to target NLK in the future.

## Methods
**Cell culture**. Primary human CD34$^+$ hematopoietic stem and progenitor cells were purified from cord or peripheral blood (New York Blood Center) or from human fetal liver tissue (Advanced Bioscience Resources and University of California, Los Angeles Center for AIDS Research) by using magnetic-activated cell sorting (Miltenyi Biotec) and were cryopreserved. Upon thawing, cells were cultured in x-Vivo15 media (Lonza) containing 10% fetal bovine serum, fms-related tyrosine kinase 3 (50 ng/ml), thyroid peroxidase (50 ng/ml), interleukin-3 (IL-3; 20 ng/ml), interleukin-6 (IL-6; 20 ng/ml), and stem cell factor (50 ng/ml). When applied, TGFβ1 was added at 5 ng/ml. Kp53A1 cells were obtained from Javier Leon and cultured in DMEM supplemented with 10% fetal bovine serum at 37 or 32 °C. Stable cell lines expressing shRNA against RPS19 with differing efficiencies; shRPS19#8 (high), shRPS19#1 (moderate), and shRPS19#3 (low) were generated by co-transfecting shRNA-carrying vectors (pLVTH) with neomycin-carrying vector (pcDNA3.1) using Lipofectamine® 2000 (Thermo Fisher). Individual clones were harvested and expanded in 100 μg/ml neomycin and RPS19 expression examined by Western blot and qRT-PCR. CD34$^+$ progenitors were obtained from mononuclear bone marrow DBA patient samples using magnetic-activated cell sorting (Miltenyi Biotec) and differentiated for 12 days. Patient samples were obtained and provided by Dr. Hanna Gazda and their collection and use was approved by the institutional review board at Boston Children's Hospital. Informed consent was obtained from affected individuals and their family members participating in the study.

**Lentiviral transduction**. Primary CD34$^+$ cells were transduced using spinoculation (30 min at 800 $g$) at MOI 50 with lentivirus expressing shRNA against RPS19, RPL11, p53, NLK, or luciferase (Luc), siRNA against NLK or a non-targeting sequence, or cDNA expressing NLK with wild type 3′UTR, a mutated 3′UTR or no 3′UTR or "escape" NLK. YFP- or CFP- tags were inserted 3′ of NLK using restriction enzyme digestion and ligation. Kinase-deficient NLK was generated by introducing the K115M mutation using point mutagenesis as reported previously[21]. Virus co-expressed GFP, RFP, mCherry, or puromycin to enable selection.

**Compounds**. Small molecule inhibitors were purchased from SelleckChem, with the exception of SD208 (Tocris), ITD-1 (Adooq Bioscience), and were diluted in dimethylsulfoxide (DMSO). Inhibitors were added to cells at indicated concentrations with a final DMSO concentration of 0.5%. Lactacystin and chloroquine were purchased from Sigma Aldrich and diluted according to manufacturer's instructions, and added to cells at a concentration of 1 μM, 10 μM and 30 μM,

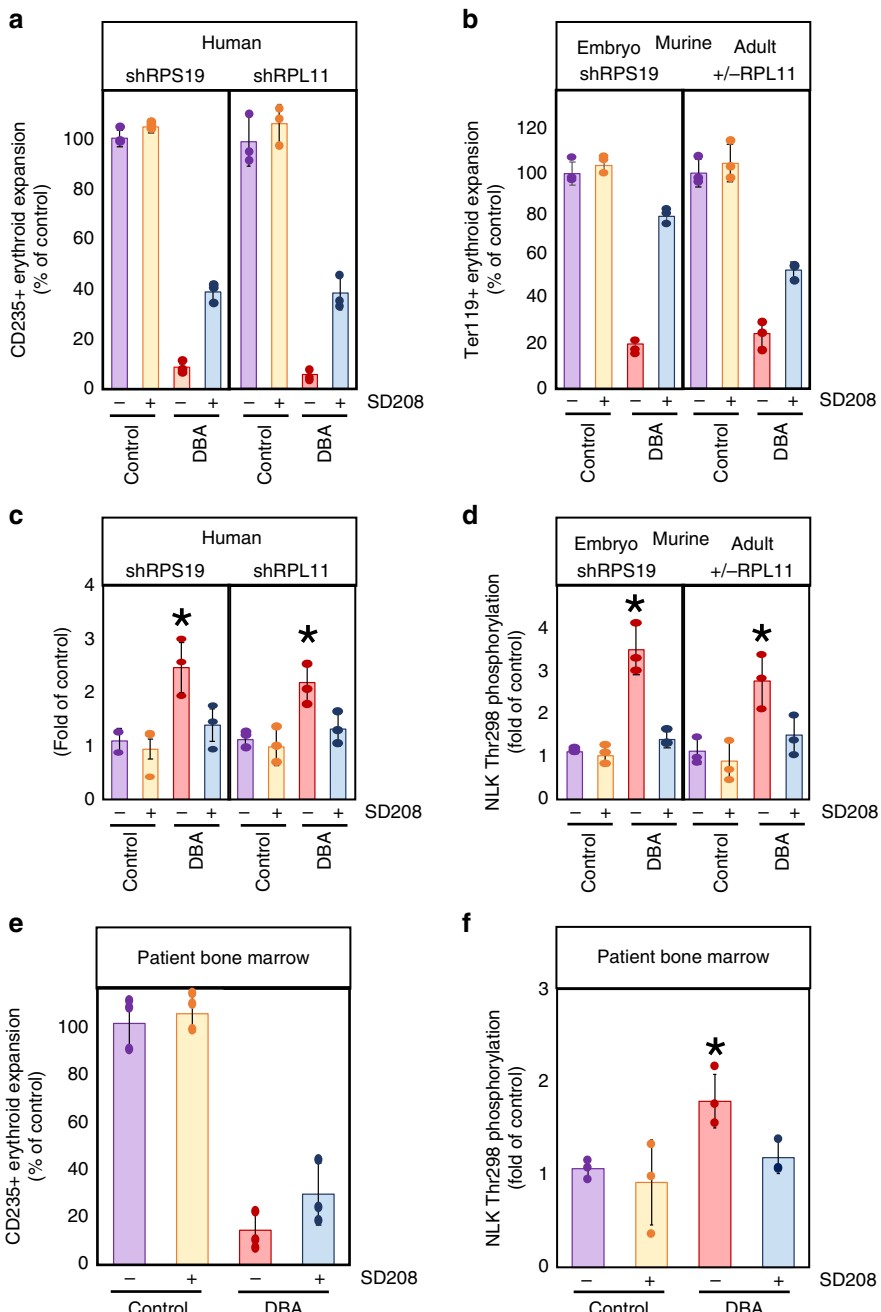

**Fig. 8 NLK inhibition increases expansion of erythroid progenitors from human and murine models of DBA. a** Human cord blood CD34+ progenitors were transduced with lentivirus co-expressing GFP with shRNA against luciferase (shLuc), RPS19 (shRPS19), or RPL11 (shRPL11). After 36 h GFP+ cells were differentiated in erythroid media in the presence or absence of 5 μM SD208 for 15 days. Cells were counted and assessed for cell surface expression of CD235. **b** Lin-Kit+ hematopoietic progenitors were obtained from three mouse embryos expressing tetracycline-inducible shRNA against RPS19 at day E14.5 or three untreated mice, and thee mature RPL11+/+ or three mature RPL11+/lox mice treated with tamoxifen for eight weeks. Cells were grown in the presence or absence of doxycycline and/or SD208 for 8 days prior to counting and assessing for Ter119 surface expression. Values were expressed as a percentage relative to untreated controls. **c, d** Intracellular phosphorylation of NLK at Thr298 was determined by capillary electrophoresis using the Peggy Sue™ Automated Western blotting platform. After lysis, 4 μg of protein from the CD235+ population was probed against pThr298-NLK and normalized to GAPDH. Detected NLK phosphorylation is plotted relative to untreated control. **e** CD34+ HSPCs were isolated from three healthy control and three DBA patient mononuclear bone marrow aspirates by magnetic bead sorting and differentiated in the presence or absence of SD208 for 14 days. After counting the total cell population, the ratio of CD235+ erythroblasts was determined by flow cytometry and number of CD235+ erythroblasts calculated. This was expressed as a percentage of the average number of erythroid cells present in untreated healthy controls. **f** NLK phosphorylation was assessed by capillary electrophoresis as above. Bars represent means ± SD with individual data points overlaid. Purple depicts untreated controls, yellow depicts controls treated with SD208, red depicts untreated DBA progenitors, while blue depicts DBA progenitors treated with SD208. n = 3 independent experiments performed in triplicate. Statistics: two-tailed Student's t test, significant *p < 0.05. Also see Supplementary Fig. 8. Source data are provided as a Source Data file.

respectively. $IC_{50}$ calculations were obtained using 8–9 concentrations of compound and generated using the IC50 Calculator software by AAT Bioquest®. NLK, p38, or ALK5 for in vitro kinase assays were immunopurified from Kp53A1 cells. NLK activation was induced by incubating cells at 32° for 24-48 h, p38 by stimulating with IL-3, IL-6, SCF, and Epo at the same concentrations used for differentiating erythroid progenitors. $IC_{50}$ values for each kinase and compound were plotted along a horizontal bar with our observed value superimposed next to documented values (if available). $IC_{50}$ values were obtained for each kinase against each substrate (e.g., NLK phosphorylation of NLK, c-Myb, and Raptor), but as IC50 curves were virtually identical for each substrate, only one is presented.

**Colony assays.** Sorted hematopoietic cells were seeded in methylcellulose medium containing IL-3, stem cell factor, granulocyte macrophage–colony-stimulating factor, and erythropoietin (H4434; STEMCELL Technologies) in triplicate, with 1000 cells per plate. Erythroid (burst-forming unit erythroid) and myeloid (colony-forming unit, granulocyte-macrophage) colonies were counted 14–18 days later. In some cases, 3000 (shRPL11) or 5000 (shRPS19 + TGFβ) cells per plate were added to ensure enough colonies for robust statistical analysis.

**Flow cytometry.** For cell surface flow cytometry, cells were incubated with human Fc receptor binding inhibitor (#14-9161−73; eBioscience) followed by primary antibodies CD235−APC (#306607; BioLegend) CD41-FITC (#303703; BioLegend) and CD11b-PE/Cy5 (#101209; BioLegend) or CD-71-APC (BD551374; BD Biosciences), p53-PECy7 (NB200-171; Novus), Sca-1-PE (#12-5981-81; eBioscience), CD34-Fluor®450 (#48-0341-82; eBioscience) and CD16/32-FITC (#101305; BioLegend). FRET measurements were obtained as described[69]. To measure ECFP and FRET cells were excited with the 405 nm laser and fluorescence was collected in the ECFP channel with a standard 450/40 filter, while the FRET-signal was measured with a 529/24 filter. To measure EYFP, cells were excited with the 488 nm laser while emission was also taken with a 529/24 filter. Data were collected on a DxP10 flow cytometer (Cytek) and analyzed by using FlowJo Software, v.9.7.2.

**Kinase and ubiquitination assays.** For NLK kinase analysis, cultures were treated as indicated and lysed for 30 min at 4 °C in 750 μl of kinase lysis buffer (50 mM Tris [pH 7.4], 5 mM EDTA, 250 mM NaCl, 0.1% Triton X-100, 50 mM NaF, 0.1 trypsin inhibitor unit of aprotinin per ml, 50 μg of phenylmethylsulfonyl fluoride per ml, 100 μM sodium vanadate, 1 μg of leupeptin per ml). For TGFβR1 kinase analysis, cells were lysed in membrane kinase buffer (10 mM Tris-HCl (pH8.0), 140 mM NaCl, 300 mM KCl, 0.5% Triton X-100, and 0.5% sodium deoxycholate with complete protease inhibitor cocktail (Roche)). Extracts were clarified, and equivalent protein was incubated overnight at 4 °C with antibody (NLK: #AB97642; Abcam, TGFβR1: V22; Santa Cruz Biotechnology). Immune complexes were collected with Catch and Release® V2.0 Reversible Immunoprecipitation System and diluted in kinase buffer (25 mM Tris [pH 7.4], 10 mM $MgCl_2$, 1 mM dithiothreitol). When cell numbers were limiting, NLK activity was amplified by adding 0.5 μg dephosphorylated NLK prior to performing kinase reaction. 50 μl of kinase buffer containing NLK or TGFβR1 sample and 5 μM ATP was incubated in the presence of biotinylated substrate (NLK, c-Myb, or Raptor for NLK and Smad2 or Smad3 for TGFβR1) immobilized on streptavidin-coated 96-well plates. The kinase reaction was allowed to proceed for 30 min at 37 °C before kinase was removed. After vigorous washing in kinase buffer, substrates were incubated for 60 min with antibody against phospho-Serine/Threonine (#525280; Calbiochem), and detected by HRP-conjugated anti-mouse antibody (#170-6516; BioRad) or a combination of phospho-Serine-HRP (ab9334; abcam) and phospho-Threonine-HRP (sc-5267; Santa Cruz Biotechnology) and SuperSignal® West Pico Chemiluminescent Substrate (Thermo Scientific). Signal was detected at 428 nm by Synergy™ H1 multi-mode microplate reader (BioTek®). Prior to kinase analysis, NLK c-Myb, Raptor, Smad2, and Smad3 were immunopurified by Catch and Release® V2.0 Reversible Immunoprecipitation System and biotinylated as per manufacturer's instructions (EZ Link™ NHS Biotin; Thermo Scientific) and immobilized on Pierce® NeutrAvidin-coated 96-well plates (Thermo Scientific). Background phosphorylation was removed by 30 min incubation in the presence of 0.1 unit/ml calf intestinal phosphatase (New England Biolabs). Ubiquitination assays were performed as above, except cells lysates were applied directly to plates in the presence of lactacystin for 2 h prior to incubation with anti-ubiquitin antibody (#P4D1; Santa Cruz Biotechnology).

**qRT-PCR.** RNA was extracted by using total RNA mini kit (Bio-Rad). RNA was transcribed into cDNA by using the iScript cDNA Synthesis Kit (Bio-Rad). The quantitative RT-PCR (qRT-PCR) reaction was run with iQ SYBR Green MasterMix (Bio-Rad) using the CFX384 Touch Real-Time PCR Detection System (Bio-Rad). 7SL small cytoplasmic RNA[27] was used as an internal control. miRNA was quantified using TaqMan® Small RNA Assays (Applied Biosystems) as per manufacturer's directions and normalized to snoRNA. Fold change of mRNA and miRNA was calculated by using the comparative $C_t$ method. List of primers is provided in Supplementary Table 1.

**Luciferase assay.** The NLK minimal promoter (1019 5′ nucleotides) and NLK 3′UTR (1885 3′ nucleotides) were cloned upstream, or downstream respectively,
of firefly luciferase in pLenti-GIII-CMV-RFP-2A-Puro (abm). Transduced into CD34$^+$ cord blood progenitors were differentiated in 1 μg/ml puromycin for 6 days. Transduction efficiency was normalized by RFP expression and firefly luciferase activity determined by Luciferase Assay Reagent II from Dual-Luciferase® Reporter (DLR™) Assay System (Promega). Luminescence was assessed using a Synergy™ H1 hybrid multi-mode microplate reader (BioTek®).

**Immunoprecipitation and blotting.** Antibodies against RPS19 (#AB40833; Abcam; 1:200 dilution), NLK (#AB97642; Abcam; 1:1000 dilution, #PAS-21877; Invitrogen; 1:1000, #PAS-25953; Invitrogen; 1:1000, #94350; Cell Signaling; 1:1000, #3535; Cell Signaling; 1:500), phosphoThr298-NLK (orb157946; biorbyt; 1:350 dilution), c-Myb (#12319; Cell Signaling; 1;1000 dilution), Raptor (#AB26264; Abcam 1;1000 dilution), phospho-Ser863 Raptor (#PAS-64849; ThermoFisher Scientific; 1:500), phospho-Ser792 Raptor (#2083; Cell Signaling; 1:500), phospho-Serine (#525280; Calbiochem; 1:1000 dilution), TAK1 (ab109526; abcam; 1:1000), pERK, pJNK, pp38 (#9910; Cell Signaling; 1:1000), ERK (orb160959; biorybt; 1:1000), JNK (#9252; Cell Signaling; 1:1000), p38 (#8690; Cell Signaling; 1:1000), cdc2/CDK1 (#77055; Cell Signaling; 1:1000), and CDK2 (#2546; Cell Signaling; 1:1000), and GAPDH (#MAB374; Millipore; 1:10000) were used according to manufacturer's instructions. The target proteins were analyzed by using SuperSignal® West Pico Chemiluminescent Substrate for horseradish peroxidase (Thermo Scientific). Densitometry was performed using Image J software (http://rsb.info.nih.gov/ij/). When indicated, proteins were co-immunoprecipitated prior to immunoblotting. Cell lysates were normalized for protein before preclearing with Protein A/G Agarose (Upstate) for 30 min, before incubating with indicated antibody overnight. Immune complexes were precipitated with Protein A/G Agarose and separated by SDS-PAGE, before being subjected to immunoblotting. Molecular size relative to size markers of bands corresponding with the protein of interest accompanies each blot. Uncropped and unprocessed scans of blots are provided in the Source Data file.

**Capillary electrophoresis.** 4 μg of protein was lysed in RIPA lysis buffer and subjected to capillary electrophoresis using the Peggy-Sue™ platform (ProteinSimple©) as per manufacturer's instructions. Antibody raised against phospho-Thr298 NLK (biorbyt) was used at 1:50.

**Mice.** The RPS19-deficient mouse model contains a doxycycline-regulatable Rps19-targeting shRNA (shRNA-D) located downstream of the collagen A1 locus, allowing dose-dependent downregulation of Rps19 expression[57]. Mice were maintained at the Lund University animal facility (Sweden) and all animal experiments were performed with consent from the Lund University animal ethics committee.

Inducible RPL11 heterozygous deletion mice[50] were fed a standard chow diet ad libitum. When indicated, standard chow diet was replaced by tamoxifen diet (Teklad, Harlan Laboratories) to induce activation of the CreERT2 transgene. All animals were maintained at the Spanish National Cancer Research Centre (CNIO) under specific pathogen-free conditions, in agreement with the recommendations of the Federation of European Laboratory Animal Science Association (FELASA). All animal procedures were evaluated and approved by the Ethical Committee of the Carlos III Health Institute, Madrid, Spain (#54-2013-v2).

**Cloning and CRISPR/Cas9.** Generation of lentiviral constructs expressing NLK cDNA, NLK cDNA with various 3′UTR mutants, and NLK with an alternative nucleotide sequence at the siRNA targeting site (escape NLK), as well as luciferase gene fused to the NLK 5′ or 3′UTR, were generated with standard molecular biology techniques. For CRISPR–Cas9 disruption of the miR181-binding sequence within the NLK 3′UTR, a series of sgRNAs were purchased from Synthego and screened for the ability to disrupt miR181 binding. K562 cells stably expressing luciferase fused to the NLK 3′UTR were electroporated with sgRNAs and Cas9. Cells were differentiated towards erythroid or myeloid lineages with hemin or TCA respectively. After 36 h, luciferase activity was assessed. Control TPA-treated cells induce miR181 and have reduced luciferase activity so maintained elevated luciferase in TPA treated group indicates miR181-binding site disruption. A control sgRNA, a poor NLK-targeting sgRNA and an efficient NLK-targeting sgRNA were electroporated with Cas9 into CD34$^+$ progenitors and transduced with shRNA against RPS19 and luciferase. After 15 days differentiation populations were counted and assessed by flow cytometry for expression of CD235, CD41a and CD11b. The indel frequency of each sorted population was determined after DNA sequencing. Designing primers and analyzing indel frequency was performed using SnapGene® Schematic and K562 screening data is presented in Supplementary Fig. 6e.

**Statistics.** $P$ values for statistical significance were obtained by using a paired Student $t$ test. Significance was designated as $p < 0.05$. The data are representative of at least three independent experiments.

**Reporting summary.** Further information on research design is available in the Nature Research Reporting Summary linked to this article.

## Data availability

The data underlying all figures are provided as a Source Data file. The authors declare that all data supporting the findings of this study are available within the paper (and its supplementary information files). Details are available from the corresponding author upon reasonable request.

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

## Acknowledgements
We thank Javier Leon for supplying Kp53A1 cells, Hanna Gazda for supplying DBA patient BM MNCs, Lars Johansson for small molecule screening infrastructure and medicinal chemistry support and Michelle Monje for critical reading of the manuscript. This work was funded by NIH T32 training grant (DK098132) and Maternal Child Health Research Institute fellowship (M.C.W), DBA Foundation, Department of Defense (BM180024), and SPARK program at Stanford (K.M.S). This chemical screen was supported by grants to J.F. from the Ragnar Söderberg Foundation, DBA Foundation, Captain Courageous Foundation, DBA Canada and the Swedish Research Council. Work in the laboratory of M.S. was funded by the IRB and by grants from the Spanish Ministry of Economy co-funded by the European Regional Development Fund (ERDF) (SAF2017-82613-R), the European Research Council (ERC-2014-AdG/669622) and "La Caixa" Foundation.

## Author contributions
M.C.W., M.S., and J.F. and K.M.S. conceived and planned the experiments. M.C.W., K.S., J.C., G.V. carried out the experiments. T.N. and H.N. advised and assisted on iPSC generation while D.P.D. and M.H.P. advised on CRISPR/Cas9 experiments. M.Y.Y. contributed to shRNA and lentivirus preparation. M.Y.Y., H.C., A.N., B.G., H.T.G., M.S., S.L., C.E.R., and K.M.S. contributed to the interpretation of the results. F.E., R.O., T.L., and J.F. designed and implemented the small molecule screen. M.C.W. took the lead in writing the manuscript. All authors provided critical feedback and helped shape the research, analysis and manuscript.

## Competing interests
The authors declare no competing interests.
