## [Peer Review File · Nature Communications]

Reviewers' Comments:

Reviewer #1:

Remarks to the Author:

The authors discovered that an atypical MAPK Nemo-like kinase (NLK) is hyperactivated in erythroid progenitors and DBA (Diamond Blackfan Anemia) model cells, using their developed in vitro kinase (ivk) assay systems. NLK activation in DBA model cells were dependent on p53 and miR181. In addition, functional analyses using RNAi and inhibitors showed that, in DBA model, activated NLK regulates c-Myb protein stability negatively and autophagy positively and inhibits erythropoiesis. They also found that subset of TGF β inhibitors increased erythropoiesis through off-target NLK in DBA model and DBA patient HPSCs. Based on these findings, the authors concluded that aberrant erythropoiesis in DBA is mediated by hyperactive NLK and NLK may be a potential target for therapy. The authors' hypothesis is very interesting. However, their data are insufficient to conclude that NLK is hyperactivated in DBA. In addition, although the authors show many data, some of them are small collections and insufficient to support the authors' conclusion. And these small data are not well-organized as a story (I feel several data does not strengthen the authors' models). Moreover, there seem to be many mislabeling, and explanation of Figures are also insufficient. So, these make difficult to read this manuscript. I think that there are many points which need to be addressed.

Comments (Main):

1) To measure the NLK kinase activity, the authors developed and used the ELISA-based ivk (in vitro kinase assay). However, current data is insufficient to convince the readers that this ivk correctly measure the NLK activity.

For example, in their ivk system, the authors detected the phosphorylation of NLK at Ser residue to measure NLK activity. However, I think they should detect the phosphorylation of NLK at Thr residue to measure NLK activity because the NLK autophosphorylation site is a Thr residue. In addition, NLK Ser-510 is phosphorylated by p38, which binds to and activates NLK (Ohnishi et al., MCB 2009). p38 can also phosphorylate both c-Myb and Raptor (Bies et al., JBC 2013; Wu et al., JBC 2011). These suggest the possibility that the immunoprecipitated NLK in ivk may include p38 and the p38 activity may affect the ivk results. Therefore, the authors should exclude these possibilities. For example, they can check whether treatment with p38 inhibitor affect the ivk results.

In addition, their ivk is an artificial system. So, this experimental system is insufficient to demonstrate the endogenous NLK activation. Further confirmation is required. For example, they can check whether pThr-298 NLK levels are upregulated in the patients and the mouse model, using western blotting (Fig 1a, c, d, 2c, 4b, d).

Related to this comment, it is surprising that their ivk results indicate that NLK was activated in the patients, while NLK expression was decreased in the patients. Previous many studies showed that overexpression of NLK can induce autoactivation of NLK, suggesting that NLK expression levels seems to positively correlate with NLK activity. Although the authors measured the activity of immunoprecipitated NLK, it remains unclear that total activity of endogenous NLK (NLK activity per cells) are really activated in the patients and the model animals.

2) The authors stated NLK activation is dependent on p53. However, p53 knockdown dramatically reduced both NLK expression levels and NLK activity (Fig 1D), indicating that "NLK expression" is dependent on p53. On the other hand, the authors showed that RPS19 knockdown promoted NLK activity but it didn't increase NLK expression levels. These suggest that RPS19 and p53 independently regulates NLK through different mechanisms.

3) Although the authors stated NLK expression is downregulated in non-erythroid hematopoietic progenitors by miR181, they did not provide the data sufficient to state this. I agree that the potential miR181-binding sequence is important. But, to demonstrate the importance of miR181, they should test the effect of miR181 knockdown.

4) The authors showed that NLK regulates c-Myb, raptor, and autophagy. However, it remains unclear whether NLK promotes aberrant erythropoiesis through the regulation of c-Myb, raptor, and autophagy in DBA. I recommend the authors to check whether NLK RNAi-induced erythroïd (CD235+ cell) expansion is reversed by forced inhibition of c-Myb or mTOR signaling or forced activation of autophagy.

5) The authors repeatedly stated "pharmacological inhibition of NLK" in this manuscript, including abstract section. This statement gives readers the impression that the authors used NLK specific inhibitor. However, they used several TGF β R1 inhibitors as NLK inhibitors (non-specific inhibitors) and they didn't test whether the inhibitors block NLK activity directly or indirectly. Thus, this statement may mislead readers. So, I recommend the authors to rephrase this. Related to this comment, the authors should confirm the effects of these inhibitors more carefully. For example, although the authors found that TGF β R1 inhibitors can inhibit NLK activity, it is a predictable result because NLK is known as a downstream of TGF- β -activated kinase 1 (TAK1) and as a substrate of p38 MAPK, which is also activated by TGF β signaling. Therefore, TGF β inhibitors may indirectly inhibit NLK activity by blocking the activity of TGF β R1 in vivo. To conclude the inhibitors blocked NLK activity not through TGF β R1 and downstream kinases, they should show the effects of the inhibitors on the phosphorylation levels (or activity) of TGF β R1, TAK1, and p38. At least, they should check whether SD208 and SB431542 don't affect the activities of p38 and TAK1, which bind to and activate NLK.

In addition, although the authors stated that Kinase Profiling of SD208 indicated NLK to be a robust target and kinase activity is 96.8% inhibited at 10 μ M, they did not show the data in this manuscript. The authors should provide this data and show the method. And, in this Kinase Profiling, did they used bacterially purified NLK or immunopurified NLK from mammalian cells? If they used bacterially purified one, it is likely that SD208 can directly inhibit NLK activity. But, if they used mammalian cell-derived one, SD208 may indirectly inhibit it. Moreover, the authors treated cells with inhibitors at 1 μ M (Fig 6d, e), while they stated 10 μ M SD208 inhibited NLK activity in vitro Kinase Profiling. Does 1 μ M SD208 can efficiently inhibit NLK activity in vitro? If it cannot, it is likely that SD203 indirectly inhibit NLK in cells.

6) It is difficult to grasp a complete view of this manuscript. I recommend the authors to improve the text and Figures (Sentence order, Figure order...).

For example, although the explanation of Kp53A1 cells were shown in page 17, this cells is used in Fig S1 experiments (shown in page 6).

Other comments:

1) The authors should mention the reason why they focused on NLK in the introduction or the beginning of result section.

2) In page 6, the authors stated "Western blot analysis demonstrated that immuno-precipitated NLK, c-Myb and raptor were phosphorylated at serine/threonine residues (Fig S1C)". However, they tested only serine phosphorylation. They should check the phosphorylation of threonine, or they should rephrase this.

3) Supplementary Figure 1 may include mislabeling. So, it is hard to understand this. For example, in Fig S1B, p-NLK is upregulated in NLK-knockdowned cells (siNLK). NLK and its substrate are phosphorylated at 32 $^{\circ}$ C but not at 37 $^{\circ}$ C in S1C and S1D, while they are strongly phosphorylated at 37 $^{\circ}$ C in S1B.

4) Although shRPS19 treatment reduced NLK expression levels in Fig 1b and 1c, it did not reduce NLK expression in Fig 1d. Is RPS19 RNAi-induced reduction of NLK expression is really reproducible? (and the authors' experimental system is stable?)
In addition, "RPL11" in Fig 1d must be in correct. "RPL19" may be correct.

5) It is difficult to understand Fig3A. For example, transfection of NLK-expressing vector did not

increase NLK expression levels in Cd235+ cells. In addition, NLK overexpression couldn't stimulate NLK activity. As I mentioned, in many cell lines, NLK overexpression is sufficient to activate NLK. Why does NLK was inactive even in NLK-overexpressed cells?

6) The authors should provide the information about miR181 binding sequence, its mutant sequence, gRNA sequence, and CRISPR-edited sequence.

7) It is difficult to understand the meaning of Fig S4C and Fig 3C and their relationship. What are Guide 1, 2, and 3 in S4C? Is "RelA" in Fig S4C mean gRNA against RelA? In Fig 3C, Guide 1 was electroporated? Guide 1 is delta181??

8) "induction of active p53 (cultured at 37C)" in page 18, lane 3-4 may be wrong. "induction of active p53 (cultured at 32C)" may be correct. There are many similar mistakes.

9) In Fig4E, the authors used Staurosporine, which inhibits all Ser/Thr-kinase. I think this experiment does not strengthen the authors' model. They should use NLK siRNA or inhibitors instead of Staurosporine.

10) Based on Fig 5CD results, the authors concluded that shRPS19 increased autophagy and this increase was suppressed by siNLK. However, their experiments are insufficient. For example, they should detect LC3 processing by western blotting.

11) In Figure S2, I expected that the phosphorylation levels of NLK, Myb, and Raptor decreased in Dox-treated cells and they are kept at the steady levels in Dox-untreated cells. However, unexpectedly, the data in Fig S2 shows these phosphorylation levels increased in untreated cells. The authors should explain this phenomenon.

Reviewer #2:

Remarks to the Author:

The manuscript proposes a novel mechanism partially responsible for the anemia in DBA. In particular, the Authors indicate that over activation of NLK targets and phosphorylates several proteins (NLK itself, cMYB and RAPTOR) which contribute the decreased erythroid proliferation and production of red cells.

Positive:

The manuscript provides strong evidences that the activity of NLK is altered in DBA patient cells and may be a major source of abnormal erythropoiesis in DBA

Major:

1) The manuscript is quite cumbersome and very long, a major revision in the structure and figures is required. Style should be revised as well. Difficult to follow.

2) Some of the paragraphs and experiments described are confusing and requires major efforts to understand the structure of the experiments or interpretation of the data. For instance, although it is obvious what is the goal of this assay, the supplemental figure 1 and legend are confusing.

3) One of the major point of this manuscript is that NLK activity is increased in cells with DBA associated mutations. However, in many experiments, the expression level of NLK is reduced. This suggest that another protein may be responsible for the increased activity of NLK, irrespective of expression level. This is not addressed.

4) Authors indicate that NLK is not deleterious in non-erythroid cells because of miR181. It is not clear why over expression of the NLK-cDNA (Figure 2) is not detrimental to non erythroid colonies

+ shRPS19. Does this cDNA does or does not contain the 3'UTR region? If not, how do they reconcile this observation? If yes, it is expected that over expression of this mRNA should have some effect also in non erythroid cells expressing miR181 + shRPS19. Or the mRNA is degraded despite it is overly expressed? Please, clarify.

5) Fig. 1: are the patient cells P53 deficient?

6) Fig. 3: some of the columns are missing, only the standard deviation bar is visible.

7) In the section "NLK activation in RPS19-insufficiency increases Autophagy through mTOR deregulation" is too long. The transition between Raptor, mTor1 and ten autophagy is confusing. Please, fix several typos or missing punctuation.

8) Pag.20. RSK seems to be responsible for targeting S6K. However, this set of experiments are providing little information and no mechanistic relationship between DBA, NLK and Raptor.

9) Description of the candidate drug is extensive and could easily be separated and expanded in a second manuscript. No justification why the candidate drug was not utilized in vivo. How does the drug targets NLK? Additional targets? These topics could be expanded in a second manuscript.

10) In light of this, my suggestions are to clarify how NLK is activated in erythroid cells and the role of P53 in patient cells

Minor:

1) Results, first paragraph: please shorten and simplify the description of the assay.

2) Pag. 7: We observed a mild increase in in vitro phosphorylation of all three substrates at day 3 compared to control cells.: this is confusing...which cells? at what time?

3) Pag. 11: "Modulation of NLK had no impact on erythroid colony formation in shLuc controls" should be in underlined

4) Pag. 3...please, rephrase: "Erythroid specificity of ribosomal insufficiency in DBA is largely due to reduced translational efficiency of genes possessing a short, complex 5'UTR that is highly upregulated,..."?

5) Pag.8: "In the presence of doxycycline, CD34+ clonal HSC colonies expressed reduced RPS19 expression, however NLK activity showed no difference from cells cultured in the absence of doxycycline (Fig S2A)." is confusing...it looks like the activity is increased in S2A, top panel.

6) pag.13: "RPS19-insufficiency had no impact ON luciferase-NLK 3'UTR fusion expression"

7) Pag.16: baring = bearing?

8) pag.17 vs pag.18: is P53 activated at 32 or 37 C?

Reviewer #3:

Remarks to the Author:

The manuscript "Diamond Blackfan is mediated by hyperactive Nemo-like kinase" by Wilkes et al attempts to show a role for a kinase known as the Nemo-like kinase (NLK) in the pathogenesis of DBA. The work is original and potentially of importance as the kinase reported here may be a candidate for developing drugs to treat DBA. While the manuscript describes a large number of experiments which appear to support a role for the NLK in DBA I have a number of concerns that I

feel need to be addressed before I could recommend this manuscript for publication. These concerns are included below.

- The introduction introduces the reader to DBA and then to the Nemo-like kinase, two seemingly disparate lines of investigation with no obvious links given between the two subjects. The manuscript then moves to the results and a description of the Nemo-like kinase assay used throughout the study. This is followed by all the changes in Nemo-like kinase activity associated with DBA in various model systems. I was left wondering why the authors began studying the Nemo-like kinase in the first place. Did these studies arise from some type of random screen linking NLK activity with DBA? Since there is no obvious link between NLK and DBA given in the introduction (perhaps this is derived from a connection with autophagy, which is underdeveloped), some type of description for what drew the authors attention to NLK would be helpful in understanding the rationale for the studies outlined.
- The NLK assay is very complex – NLK is immunoprecipitated from cells, incubated with immobilized substrates and then phosphorylation of substrates assessed by ELISA. All these manipulations would seem quite challenging given the small number of cells involved given the best of circumstances, but many of these experiments are carried out on cells haplo-insufficient for ribosomal proteins induced to differentiate along the erythroid lineage, where multiple studies including those from the authors themselves show that p53 is induced and many of the cells become apoptotic.
 - o The authors show NLK is induced 4 days or so after knockdown of RPS19. It is important to know whether there is significant cell death at this point and if there is a subsequent reduction in the number of cells compared to baseline.
 - o Investigation into cell death and reductions in cell number is important because the authors state in supplemental figure 1 that if there was a significant reduction in cell numbers observed extracts were supplemented with purified dephosphorylated NLK to aide in dimerization. If this is true for some measurements and not others, it would be helpful to know which samples received the supplements as they would be expected to correlate with cell number loss induced by RP haplo-insufficiency and so bias data interpretation.
 - o There is a lack of correlation between the Western blotting reported in supplemental figure 1 and the ELISA data. The ELISA data show huge increases in all three substrates, whereas the Western data for raptor shows a very high background and barely detectable levels of phosphorylated raptor.
- In general, the overall reproducibility of the hundreds of assay measurements reported in this manuscript is stunning given the complexity of the manipulations involved and the biological issues associated with working with dead and dying cells. In this regard it would be helpful to see some (clearly not all) of the raw data obtained for some of the key data, including any normalizations and other types of manipulations used to report the data. This would be helpful in determining whether appropriate statistical analyses were used, especially since all statistical analysis seemed to use a paired Student T test.
- Regarding statistical analyses, the authors state when referring to Fig. 2 that an increase in colony number to 105.5% of control was insignificant but then continue to report out their data as 72.4% etc. My point is why include the decimal in reporting the data if this does not present a significant figure.
- Referring to the expression data in Fig. 1 the authors mention protein levels in the text but only show RNA levels in the figure.
- The authors state that haploinsufficiency for ribosomal proteins induces NLK activity despite the fact that amounts of NLK are reduced. While I can accept this, the authors go on to state that it is the lower amount of NLK (not the activity of NLK) resulting from the upregulation of miR181 that prevents other hematopoietic lineages from being affected by RP haploinsufficiency. Given the disconnect between NLK levels and activity reported in the erythrocyte lineage, drawing the aforementioned conclusions for other hematopoietic lineages seems a bit of an overreach.
- Including the results from patients where the gene affected is unknown muddies the waters a bit and would seem to require some type of explanation for how to incorporate these data into the rest of the story. In the discussion and throughout the manuscript for that matter the authors

equate inductions in NLK activity with ribosomal protein haploinsufficiency. Given that this induction also occurs for cells from DBA patients where the gene is unknown, I can't help but wonder whether ribosome biogenesis is affected in these cells and if not, what implications this would have on the conclusions drawn in the manuscript.

- Factors downstream of activated p53 have been extensively studied in many different systems. Has NLK been encountered in any of these studies, or is its presumed role as a downstream effector of p53 unique to the erythrocyte lineage?

We would like to thank the reviewers for their helpful comments and suggestions. We have addressed each of the concerns raised by the reviewers. We also significantly revised the manuscript, increasing flow and clarity, and removed less relevant data. We have focused on the central observations that NLK is hyperactivated in DBA models; inhibition of NLK activity improves erythroid expansion; and mechanistic data with miR-181. We have added new data that strengthens these conclusions, specifically by analysis of intracellular NLK activation (to complement *in vitro* kinase assays) and additional validation of the assays (with controls) and reagents previously used. The revised manuscript is substantially improved due to the recommendations of the reviewers.

New data generated for inclusion in the re-submission include:

Fig.1e

Fig.2a,b

Fig.3c

Fig.6b,e,h

Fig.7b,f

Supplementary Fig.1a-c

Supplementary Fig.2e,f

Supplementary Fig.3d

Supplementary Fig.4b

Supplementary Fig.6d,e

Supplementary Fig.7a-d,e (lower)

Reviewers' comments

Reviewer #1

General:

The authors discovered that an atypical MAPK Nemo-like kinase (NLK) is hyperactivated in erythroid progenitors and DBA (Diamond Blackfan Anemia) model cells, using their developed *in vitro* kinase (*ivk*) assay systems. NLK activation in DBA model cells was dependent on p53 and miR181. In addition, functional analyses using RNAi and inhibitors showed that, in DBA model, activated NLK regulates c-Myb protein stability negatively and autophagy positively and inhibits erythropoiesis. They also found that subset of TGF β inhibitors increased erythropoiesis through off-target NLK in DBA model and DBA patient HPSCs. Based on these findings, the authors concluded that aberrant erythropoiesis in DBA is mediated by hyperactive NLK and NLK may be a potential target for therapy. The authors' hypothesis is very interesting. However, their data are insufficient to conclude that NLK is hyperactivated in DBA. In addition, although the authors show many data, some of them are small collections and insufficient to support the authors' conclusion. And these small data are not well organized as a story (I feel several data does not strengthen the authors' models). Moreover, there seem to be many mislabeling, and explanation of Figures is also insufficient. So, these make difficult to read this manuscript. I think that there are many points, which need to be addressed.

Response: Thank you for the very helpful comments and suggestions.

- We sincerely apologize to the reviewer for the errors in the manuscript. We have completely re-written the manuscript, corrected the errors, and reformatted the figures. New data have been added to strengthen our conclusions. Confusing and unnecessary data have been removed to streamline the results and conclusions.

Specific (Main):

Comment 1) To measure NLK activity, the authors developed and used the ELISA-based *in vitro* kinase assay). However, current data are insufficient to convince the readers that this *in vitro* correctly measure the NLK activity. For example, in their *in vitro* system, the authors detected the phosphorylation of NLK at Ser residue to measure NLK activity. However, I think they should detect the phosphorylation of NLK at Thr residue to measure NLK activity because the NLK autophosphorylation site is a Thr residue. In addition, NLK Ser-510 is phosphorylated by p38, which binds to and activates NLK (Ohnishi et al., MCB 2009). p38 can also phosphorylate both c-Myb and Raptor (Bies et al., JBC 2013; Wu et al., JBC 2011). These suggest the possibility that the immunoprecipitated NLK in *in vitro* may include p38 and the p38 activity may affect the *in vitro* results. Therefore, the authors should exclude these possibilities. For example, they can check whether treatment with p38 inhibitor affect the *in vitro* results. In addition, their *in vitro* is an artificial system. So, this experimental system is insufficient to demonstrate the endogenous NLK activation. Further confirmation is required. For example, they can check whether pThr-298 NLK levels are upregulated in the patients and the mouse model, using western blotting (Fig 1a, c, d, 2c, 4b, d). Related to this comment, it is surprising that their *in vitro* results indicate that NLK was activated in the patients, while NLK expression was decreased in the patients. Previous many studies showed that overexpression of NLK can induce autoactivation of NLK, suggesting that NLK expression levels seem to positively correlate with NLK activity. Although the authors measured the activity of immunoprecipitated NLK, it remains unclear that total activity of endogenous NLK (NLK activity per cells) is really activated in the patients and the model animals.

Response:

- With regard to the *in vitro* kinase assays, we have attempted to use conventional techniques to compare intracellular NLK activity with *in vitro* NLK activity in primary cells. After an extended effort to examine intracellular NLK phosphorylation using flow cytometry, background staining was problematic. As an alternative, we pooled cells from multiple experiments to provide enough lysate to perform Western blot analysis with a phospho-specific NLK antibody. To analyze intracellular NLK phosphorylation in more limited mouse and DBA patient samples, we employed capillary electrophoresis technology using the Peggy-Sue™ platform from Protein Simple©. As observed by *in vitro* kinase activity, intracellular NLK phosphorylation significantly increased in DBA models by both conventional Western blot analysis (included in Figs. 2a, S3, 3c) and capillary electrophoresis (included in Fig. 6b, e, h). The increase in NLK phosphorylation in DBA models was not as dramatic as detected by *in vitro* kinase assay (see Figs. 2a, S3 and 3c). However, while NLK phosphorylation will correlate with activity, the activity of a kinase toward each specific substrate differs and is influenced by a number of other factors.

We have also demonstrated by conventional Western blot analysis that raptor (an NLK substrate) phosphorylation is also restricted to the population with high NLK phosphorylation in differentiating erythroid progenitors (included as Fig. 3c). This result complements the data previously provided in cultured Kp53A1, thus validating the requirement of NLK for raptor phosphorylation in this model system (see Fig. S2c and d).

We compared *in vitro* kinase activity with NLK phosphorylation in iPSCs derived from a DBA patient by conventional Western blot analysis. NLK phosphorylation was significantly higher in clones from an RPS26-haploinsufficient DBA patient but was reduced upon re-expression of WT RPS26 (now included as Fig. S3d).

- In terms of antibody specificity for the *in vitro* kinase assays, we appreciate the concerns of Reviewer 1. As well as extending the analysis of NLK activity to include intracellular phosphorylation of NLK (included in Figs. 2a, S3, 3c, 6b, e, h) and raptor (Fig. 3c) in differentiating progenitors and previously examined in Kp53A1 cultured cells (Fig S2c and d), the specificity of NLK antibodies and siRNA used in the *in vitro* assays were critically examined. After immunoprecipitation with the antibody utilized to pull down NLK for the *in vitro* kinase assays, flow-through from the catch-&-release® columns was subjected to SDS-PAGE and probed by Western blot for other MAPK family members (TAK1, ERK, JNK and p38) as well as similar kinases that can phosphorylate c-Myb and Raptor *in vitro* (cdk1 and cdk2). None of these proteins were pulled down with the NLK antibody (introduced as Fig S2e). Furthermore, a similar pattern of NLK *in vitro* activity was observed when NLK was immunopurified from ribosome-insufficient samples using 3 separate NLK antibodies recognizing different epitopes in the protein (included as Fig. S2f). This supports our conclusion that NLK is responsible for the observed kinase activity in our NLK *in vitro* assays.

Expression of siRNA against NLK reduces NLK activity (Fig. 1c and d), which supports that the observed kinase activity can be attributed to NLK. To further validate our results, we confirmed that NLK siRNA did not impact related MAPKs/cdks (now included as Fig. S1b) and generated 3 different shRNA sequences against NLK. While shRNA knockdown was not as extensive as the siRNA, a corresponding increase in erythroid expansion inversely correlated with NLK knockdown (this is included as Fig. S1a).

- With regard to phosphorylation of NLK at Thr residue, while the antibody used is listed as a phospho-Serine, it recognizes phospho-Threonine with similar affinity. In the conditions of this assay (and with the p-Ser and p-Thr antibodies we tried) it was too difficult to compare between the extent of Ser versus Thr phosphorylation with any degree of confidence. As new phospho-specific antibodies become available, this could be re-examined. While we are interested in the phosphorylation status of NLK at particular residues (especially Ser-510) to assist in our understanding of NLK activation in these cells, this is beyond the scope of the current study and will be investigated in the future. Also, examination of *in vitro* phosphorylated NLK does not detect the phosphorylation status of the intracellular NLK, only the ability of immunoprecipitated NLK to phosphorylate other NLK *in vitro*. As the reviewer points out, it is likely

intracellular NLK is phosphorylated by other kinases (at other residues) other than just other NLK molecules.

- In terms of p38 as the potential kinase, as an atypical MAPK family member there is high degrees of redundancy in substrates for these kinases, and we agree it has been very important to more rigorously test our results because of this. Indeed ERK, JNK, p38 as well as CDK1 and CDK2 can also phosphorylate Myb and Raptor in the non-rate limiting conditions of an *in vitro* kinase assay. In the case of ERK, it has been demonstrated that these phosphorylation events do not correspond with intracellular ERK activation. To address if kinases other than NLK could be contributing to substrate phosphorylation in the *in vitro* kinase assays, we have examined the specificity of the antibody used to immunoprecipitate NLK. ERK, JNK, p38, TAK1, Cdk1 and Cdk2 have been documented to phosphorylate c-Myb and Raptor, but none of these proteins were detected by Western blot analysis after immunoprecipitation with the NLK antibody (included as Fig. S2e). To further eliminate the possibility of artifacts due to cross-reactivity of our NLK antibody, we performed the *in vitro* kinase assays after immunoprecipitating with 4 separate antibodies, each with a unique epitope. Each of the 3 antibodies yielded strikingly similar results (introduced as Fig. S2f).

We have decided not to utilize p38/TAK1 inhibitors to address this issue for the following reasons: 1) p38 and TAK1 inhibitors have off-target effects on other kinases, including NLK and 2) P38 and TAK1 have been implicated as being upstream regulators of NLK; thus, if the compounds do have an effect it could be because of indirect inhibition of NLK, independent of NLK, direct inhibition of NLK due to off-target inhibition.

- With regard to *in vitro* kinase assays being an artificial system, we have now addressed this concern. We obtained enough ribosome-insufficient and -competent erythroid and non-erythroid progenitors to perform either conventional Western blot analysis (included as Figs. 2a, 3c) or capillary electrophoresis (included as Fig. 6b, e, h). NLK phosphorylation was increased in ribosome-insufficient erythroid progenitors, although the increase was less pronounced than the increase observed by *in vitro* kinase assays.

In addition, we performed the assay with differentiated cells from DBA-patient derived iPSCs with similar results (presented Fig. S3d). It should also be noted that we utilized CD71+ erythroid progenitors, rather than CD235+ progenitors, as CD71 comes up earlier during differentiation and we could obtain more cells prior to loss from the RPS19-insufficient population. To allow comparison between NLK activity in CD71+ versus CD235+ populations, *in vitro* kinase was performed in both populations and the new data is presented (see Fig. 2b).

- In terms of our statement that the *in vitro* kinase results suggest that NLK was activated in the patients while NLK expression was decreased, we apologize for the confusion. The apparent decrease in NLK expression in these assays is the result of different hematopoietic progenitors in the population. As this was not well explained in

the initial submission, we have now clarified this by reorganizing the manuscript and including new data. Examination of NLK expression between erythroid and non-erythroid progenitors indicates that NLK expression is significantly lower in non-erythroid progenitors. We have included new data confirming that the protein level (see Figs. 1e, 2a, S3d, 3c) correlates with previous mRNA analysis (see Fig. 1f). Therefore, NLK expression does not decrease in erythroid progenitors from DBA patients. As these assays are normalized by cell numbers, the proportion of low-NLK-expressing non-erythroid progenitors will be increased in the anemic samples.

Similar to other kinases that autoactivate upon dimerization, an increase in concentration should increase autoactivation, particularly *in vitro*. However, as we observed no increase in NLK expression in erythroid progenitors we attribute the increase in NLK activation to deregulation of upstream regulators, rather than modulation of protein expression. Interestingly, attempted over-expression of recombinant NLK in erythroid progenitors did not dramatically increase total NLK expression but did increase in non-erythroid progenitors, perhaps suggesting a mechanism to limit NLK expression in these cells (Fig. 7f).

- In response to the concern that although the authors measured the activity of immunoprecipitated NLK, it remains unclear that total activity of endogenous NLK (NLK activity per cells) is really activated in the patients and mouse models; we thank the reviewer for this comment. While we hoped to address this by phospho-NLK using flow cytometry with primary patient samples and DBA mouse hematopoietic stem and progenitor cells, the antibodies had very high background staining. In addition, there were too few cells for conventional Western blot analysis. However, we were able to generate iPSC clones from a DBA patient expressing a mutant RPS26, and observed limited hematopoietic differentiation. CD71+ clones demonstrated high NLK phosphorylation and this decreased when wild type RPS26 was expressed (included as Fig. S3d) which correlates with NLK *in vitro* kinase activity observed previously (see Fig. S3c). We also examined intracellular NLK phosphorylation in mouse and patient samples using capillary electrophoresis and documented statistically increased levels of Thr298 phosphorylation (introduced as Fig.6 b,e,h).

Comment 2) The authors stated that NLK activation is dependent on p53. However, p53 knockdown dramatically reduced both NLK expression levels and NLK activity (Fig 1D), indicating that “NLK expression” is dependent on p53. On the other hand, the authors showed that RPS19 knockdown promoted NLK activity but it didn't increase NLK expression levels. These suggest that RPS19 and p53 independently regulates NLK through different mechanisms. On the other hand, the authors showed that RPS19 knockdown promoted NLK activity but it didn't increase NLK expression levels. These suggest that RPS19 and p53 independently regulates NLK through different mechanisms.

Response: We apologize for the error in the figure. The bottom panel was meant to be p53 expression, not NLK. This has been corrected and is included in the new Fig. S3b. We and others previously showed that RPS19-deficiency induces p53 stability in erythroblasts (Danilova, et al. Blood, 2008; Danilova et al. Mech Dev, 2008). We do not

observe NLK activation when p53 is silenced by shRNA. As we do not see NLK expression altered between control and diseased states, we concluded that NLK expression is most likely p53-independent.

Comment 3) Although the authors stated NLK expression is downregulated in non-erythroid hematopoietic progenitors by miR181, they did not provide the data sufficient to state this. I agree that the potential miR181-binding sequence is important. But, to demonstrate the importance of miR181, they should test the effect of miR181 knockdown.

Response: Thank you. MiR-181 is critical for differentiation of all hematopoietic lineages, including influencing the fate of common megakaryocyte/erythroid progenitors (MEPs). Particularly relevant to us, the knockdown of miR181 in hematopoietic stem and progenitor cells shifts the differentiation of MEPs dramatically to the erythroid lineage (independently of NLK). It would be difficult to interpret results after such disruption in normal hematopoiesis. However, in cultured lymphoblastic cell lines (LCLs) from DBA patients that ordinarily express high levels of miR-181 and low levels of NLK, inhibition of miR181 did increase NLK expression (now included as Fig. S7c). Even more interestingly, LCLs derived from DBA patients have similar proliferation to controls, but proliferation was greatly impacted by miR181 inhibition of DBA-derived LCLs with negligible impact in controls (included as Fig. S7d). Because of the above mentioned NLK-independent effects of miR-181 knockdown, we chose to examine the effects of miR181 on hematopoiesis by disrupting miR-181 binding to the 3'UTR of NLK by mutation or CRISPR/Cas9-mediated disruption (see Fig. 7).

Comment 4) The authors showed that NLK regulates c-Myb, raptor, and autophagy. However, it remains unclear whether NLK promotes aberrant erythropoiesis through the regulation of c-Myb, raptor, and autophagy in DBA. I recommend the authors to check whether NLK RNAi-induced erythroid (CD235+ cell) expansion is reversed by forced inhibition of c-Myb or mTOR signaling or forced activation of autophagy.

Response: Thank you for this suggestion. We agree that modulation of c-Myb, mTOR or autophagy to counteract NLK knockdown in RPS19-insufficient cells was not tested. As mTOR stimulation has been proposed as a treatment for DBA, this is something we are investigating further. In attempts to focus this manuscript, we have removed mTOR and autophagy data, including raptor phosphorylation only as a surrogate of intracellular NLK activation. As the preliminary studies that guided us to examine NLK focused on c-Myb, that data has been included (Figs.1 and 3) but because modulation of endogenous levels of c-Myb has dramatic phenotypes, and because the role of c-Myb in erythropoiesis is well documented, we have chosen not to modulate it specifically.

Comment 5) The authors repeatedly stated “pharmacological inhibition of NLK” in this manuscript, including abstract section. This statement gives readers the impression that the authors used NLK specific inhibitors. However, they used several TGFβR1 inhibitors as NLK inhibitors (nonspecific inhibitors) and they didn't test whether the inhibitors block NLK activity directly or indirectly. Thus, this statement may mislead readers. So, I recommend the authors to rephrase this.

Related to this comment, the authors should confirm the effects of these inhibitors more carefully. For example, although the authors found that TGF β R1 inhibitors can inhibit NLK activity, it is a predictable result because NLK is known as a downstream of TGF β -activated kinase 1 (TAK1) and as a substrate of p38 MAPK, which is also activated by TGF β signaling. Therefore, TGF β inhibitors may indirectly inhibit NLK activity by blocking the activity of TGF β R1 *in vivo*.

To conclude the inhibitors blocked NLK activity not through TGF β R1 and downstream kinases, they should show the effects of the inhibitors on the phosphorylation levels (or activity) of TGF β R1, TAK1, and p38. At least, they should check whether SD208 and SB431542 don't affect the activities of p38 and TAK1, which bind to and activate NLK.

In addition, although the authors stated that Kinase Profiling of SD208 indicated NLK to be a robust target and kinase activity is 96.8% inhibited at 10 μ M, they did not show the data in this manuscript. The authors should provide these data and show the methods. And, in this Kinase Profiling, did they use bacterially purified NLK or immunopurified NLK from mammalian cells? If they used bacterially purified one, it is likely that SD208 can directly inhibit NLK activity. But, if they used mammalian cell-derived one, SD208 may indirectly inhibit it.

Moreover, the authors treated cells with inhibitors at 1 μ M (Fig 6d, e), while they stated 10 μ M SD208 inhibited NLK activity in vitro Kinase Profiling. Does 1 μ M SD208 efficiently inhibit NLK activity in vitro? If it cannot, it is likely that SD203 indirectly inhibit NLK in cells.

Response: Thank you, these are excellent points.

- The panel of TGF β R1 inhibitors was tested for *in vitro* NLK and TGF β R1 inhibition. As this was not clear, we have revised the manuscript and re-emphasized this point. The results have been moved from the main data in the manuscript to the supplementary data section, but is stated more clearly in the results and discussion sections (see Fig. S6). In terms of "specific" NLK inhibition, to our knowledge, no specific NLK inhibitor compounds are available and there are very few inhibitors available that do not have affect other "off-target" molecules, often inhibiting with similar IC₅₀s as the advertised target. This is particularly true of TGF β R1 and p38 inhibitors and by the reviewer's logic, would therefore be misleading to advertise as pharmacological inhibitors of TGF β R1. To better define the NLK inhibitory properties of SD208, SB431542 and Galunisertib we performed dose response curves *in vitro*. The generated NLK IC₅₀ values can be compared with TGF β R1 IC₅₀ values we performed concurrently. Our obtained TGF β R1 IC₅₀ values closely align with published values.

- We examined whether TGF β signaling was implicated in our observed results. TGF β does suppress erythroid and myeloid expansion (see Fig. 5a and Aglietta et al, 1999. Experimental Hematology 17:296-299, Sakamaki et al, 1999. Blood 94:1961-70) but a number of our results indicate TGF β is not involved in the pathogenesis of DBA. Primarily, we documented a broad panel of TGF β inhibitors could suppress recombinant TGF β growth effects in mouse and human differentiating progenitors (see Figs. 4c and

S5a) and TGF β R1 inhibitors prevented the phosphorylation of TGF β R1 substrates, Smad2 and 3 *in vitro* (Fig. S6b), but only a small subset could improve erythropoiesis (see Figs. 4b,d and S6c). The TGF β R1 inhibitors that improved erythropoiesis were the same inhibitors that reduced kinase activity from NLK immunoprecipitated with inhibitor-treated cells (see Fig. 4e) in *in vitro* kinase assays (Fig. S6c). Furthermore, addition of recombinant TGF β suppressed the growth of RPS19-insufficient progenitors by the same percentage, which does not support a model in which TGF β is already signaling in DBA. While it is possible TAK1 or p38 signaling may be upstream regulators that may also be targeted by these compounds, the inability of other TGF β inhibitors to impact RPS19-insufficiency, strongly argues against a model in which TGF β is contributing. As the ability of these compounds to directly inhibit activated NLK when added *in vitro* (Fig. S6c), also suggests NLK is a direct pharmacological target inside the cell.

- As mentioned above, direct addition of inhibitors into *in vitro* kinase assays for both NLK and TGF β R1 strongly supports a role for SD208, SB431542 and Galunisertib as direct inhibitors of NLK. As documented in Fig. S2e, no TAK1 or p38 is present with immunopurified NLK in these assays. As other TGF β R1 inhibitors did not impact erythropoiesis (but did impact TGF β R1 intracellular and *in vitro* activity) it is challenging to envision a scenario in which TGF β is involved. There are multiple upstream kinases of TAK1 and p38 activation and as both have been implicated in hematopoiesis and NLK activation. We are currently studying activators of NLK in DBA, however, we feel this is beyond the scope of the current study. As stated above, there is no evidence that SD208 inhibits TAK1 or p38, but SD208 inhibits NLK directly with an IC₅₀ of 440nM.

- In terms of kinase profiling, we have now included the top 22 hits from the kinase profiling data (included as Fig. S4b) and the requested information is included in supplemental methods.

- In terms of the concentration of small molecules, SD208 does inhibit NLK at 1uM *in vitro* and activation of NLK was significantly inhibited in *in vitro* assays by SD208 below 0.1uM (Fig. S8). We have expanded and replaced these data with complete dose response curves and calculated IC₅₀ values for SD208, SB431542 and Galunisertib. Although the Kinase profiling only utilized one concentration, 10uM of SD208 inhibited NLK by 96.8% (Fig S4b).

Comment 6) It is difficult to grasp a complete view of this manuscript. I recommend the authors to improve the text and Figures (Sentence order, Figure order...). For example, although the explanation of Kp53A1 cells were shown in page 17, these cells were used in Fig S1 experiments (shown in page 6).

Response: Thank you. We have completely re-written and reformatted the manuscript in an attempt to focus and clarify our central conclusions.

Other comments:

1) The authors should mention the reason why they focused on NLK in the introduction or the beginning of result section.

Response: Thank you. We now explain that we became interested in NLK because NLK has been reported to modulate c-Myb protein levels through ubiquitin-dependent proteasome degradation. Expression of siRNA against NLK not only restored c-Myb protein to control levels, but also improved erythroid expansion (see Fig. 1). This led to our studies on NLK.

2) In page 6, the authors stated "Western blot analysis demonstrated that immunoprecipitated NLK, c-Myb and raptor were phosphorylated at serine/threonine residues (Fig S1C)". However, they tested only serine phosphorylation. They should check the phosphorylation of threonine, or they should rephrase this.

Response: We appreciate the reviewer's comment. The antibody used recognizes both serine and threonine phosphorylation. Although we attempted to differentiate between the two with multiple antibodies, all had various degrees of cross-reactivity.

3) Supplementary Figure 1 may include mislabeling. So, it is hard to understand this. For example, in Fig S1B, p-NLK is upregulated in NLK-knockdown cells (siNLK). NLK and its substrate are phosphorylated at 32°C but not at 37°C in S1C and S1D, while they are strongly phosphorylated at 37°C in S1B.

Response: We apologize for the mislabeling. We have now corrected the errors (see Fig. S2b).

4) Although shRPS19 treatment reduced NLK expression levels in Fig 1b and 1c, it did not reduce NLK expression in Fig 1d. Is RPS19 RNAi-induced reduction of NLK expression is really reproducible? (and the authors' experimental system is stable?) In addition, "RPL11" in Fig 1d must be in correct. "RPL19" may be correct.

Response: Thank you for the comment. NLK expression is reduced in the total population of DBA model cells because the ratio of the high-NLK-expressing erythroblasts and low-NLK-expressing non-erythroblasts, which decreases. Second, the graph labeled "NLK" in previous Fig 1d, actually depicted p53 expression. Thus, RPS19 was incorrectly labeled "RPL11". These data are now corrected and included as Figs. 2d and S3b.

5) It is difficult to understand Fig3A. For example, transfection of NLK-expressing vector did not increase NLK expression levels in Cd235+ cells. In addition, NLK overexpression couldn't stimulate NLK activity. As I mentioned, in many cell lines, NLK overexpression is sufficient to activates NLK. Why was NLK inactive even in NLK-overexpressed cells?

Response: We apologize for the confusion. The lower panel of this figure (now Fig. 7f) was mislabeled. We have now corrected this. The increase in NLK expression is much lower in erythroblasts (with high endogenous NLK) than observed in other myeloid lineages (with low endogenous NLK). We postulate there may be an upper limit of NLK expression for viability, but this is speculative. With regard to the possibility of increased transfection increasing expression and activity proportionally, we propose that upstream modulators are increasing the catalytic activity of the existing NLK leading to hyperactivity – without modulating NLK expression. We also suggest that NLK expression is being influenced by a number of cellular variables in these cells that

compound the simple transfection:expression:activity ratio observed in other cell models.

6) The authors should provide the information about miR181 binding sequence, its mutant sequence, gRNA sequence, and CRISPR-edited sequence.

Response: Thank you. This information has been now been included (see new Fig. S7e).

7) It is difficult to understand the meaning of Fig S4C and Fig 3C and their relationship. What are Guide 1, 2, and 3 in S4C? Is “RelA” in Fig S4C mean gRNA against RelA? In Fig 3C, Guide 1 was electroporated? Guide 1 is delta181??

Response: We have now reformatted this figure and included a descriptive schematic diagram (Fig. 7b). We also revised the figure legend and results section.

8) “induction of active p53 (cultured at 37C)” in page 18, lane 3-4 may be wrong. “induction of active p53 (cultured at 32C)” may be correct. There are many similar mistakes.

Response: We apologize for the error. Every effort has been made to correct such oversights.

9) In Fig4E, the authors used Staurosporine, which inhibits all Ser/Thr-kinase. I think this experiment does not strengthen the authors’ model. They should use NLK siRNA or inhibitors instead of Staurosporine.

Response: We agree with the reviewer. The mechanism of NLK-mediated degradation of c-Myb has been well previously reported. To improve the focus of this manuscript, we have decided to remove the staurosporine data.

10) Based on Fig. 5CD results, the authors concluded that shRPS19 increased autophagy and this increase was suppressed by siNLK. However, their experiments are insufficient. For example, they should detect LC3 processing by western blotting.

Response: We agree that definitively examining autophagy in erythropoiesis requires further expansion than we can include in this study. Consequently, we have removed autophagy from the current manuscript and are examining this in the context of mTOR signaling in a separate study.

11) In Figure S2, I expected that the phosphorylation levels of NLK, Myb, and Raptor decreased in Dox-treated cells and they are kept at the steady levels in Dox-untreated cells. However, unexpectedly, the data in Fig S2 shows these phosphorylation levels increased in untreated cells. The authors should explain this phenomenon.

Response: We apologize for the confusion. Unlike Fig. S2a, in which dox induces the insufficiency in healthy background, the iPSCs are generated from the ribosomal-insufficient state, and dox induces expression of the healthy WT allele (i.e. phenotypic rescue). To avoid confusion, the current submission includes only iPSCs from the DBA patient, +/- induction of the WT (Fig S5c/d).

Reviewer 2 Comments

General:

The manuscript proposes a novel mechanism partially responsible for the anemia in DBA. In particular, the Authors indicate that over activation of NLK targets and phosphorylates several proteins (NLK itself, cMYB and RAPTOR) which contribute the decreased erythroid proliferation and production of red cells.

Positive:

The manuscript provides strong evidences that the activity of NLK is altered in DBA patient cells and may be a major source of abnormal erythropoiesis in DBA

Specific (Major):

Comment 1) The manuscript is quite cumbersome and very long, a major revision in the structure and figures is required. Style should be revised as well. Difficult to follow.

Response: We would like to apologize to the reviewer. We concur and a more succinct manuscript has been submitted.

Comment 2) Some of the paragraphs and experiments described are confusing and requires major efforts to understand the structure of the experiments or interpretation of the data. For instance, although it is obvious what is the goal of this assay, the supplemental Figure 1 and legend are confusing.

Response: Thank you. We have addressed this in the revised manuscript. Several variables must be accounted for, including different progenitors within a sample as well as distinct stages of differentiation across samples. Lineage cell numbers, ratios, size, viability, differentiation, proliferation and apoptosis are all influenced during normal differentiation and impacted by ribosomal-insufficiency. While every effort has been made to clarify the experiments in the text, an element of complexity is inherent in the experimental design.

Comment 3) One of the major points of this manuscript is that NLK activity is increased in cells with DBA associated mutations. However, in many experiments, the expression level of NLK is reduced. This suggests that another protein may be responsible for the increased activity of NLK, irrespective of expression level. This is not addressed.

Response: We have now clarified that the apparent reduction in NLK expression (in the mixed population) during ribosomal-insufficiency has to do with the increased ratio of low-NLK-expressing non-erythroid progenitors, rather than a loss of NLK expression in erythroid progenitors. The decrease in NLK expression in these assays is the result of different hematopoietic progenitors in the population. As this was not well explained in the initial submission, we have now clarified this by revising and reorganizing the manuscript, and have included new data. Examination of NLK expression between erythroid and non-erythroid progenitors indicates that NLK expression is significantly lower in non-erythroid progenitors. We have included new data confirming that the protein level (see Figs. 1e, 2a, S3d, 3c) correlates with previous mRNA analysis (see Fig. 1f). Therefore, NLK expression does not decrease in erythroid progenitors in DBA patients. As these assays are normalized by cell numbers, the proportion of low-NLK-expressing non-erythroid progenitors will be increased in the anemic samples.

Similar to other kinases that autoactivate upon dimerization, an increase in concentration should increase autoactivation, particularly *in vitro*. However, as we observe no increase in NLK expression in erythroid progenitors we attribute the increase in NLK activation to deregulation of upstream regulators, rather than modulation of protein expression. Interestingly, attempted over-expression of recombinant NLK in erythroid progenitors did not dramatically increase total NLK expression but did increase in non-erythroid progenitors, perhaps suggesting a mechanism to limit NLK expression in these cells (Fig. 7f). To summarize, NLK is only expressed at high levels in erythroid progenitors (irrespective of ribosome status), and ribosomal-insufficiency leads to activation of that NLK.

Comment 4) Authors indicate that NLK is not deleterious in non-erythroid cells because of miR181. It is not clear why over expression of the NLK-cDNA (Figure 2) is not detrimental to non-erythroid colonies + shRPS19. Does this cDNA contain the 3'UTR region? If not, how do they reconcile this observation? If yes, it is expected that over expression of this mRNA should have some effect also in non-erythroid cells expressing miR181 + shRPS19. Or the mRNA is degraded despite it is over expressed? Please, clarify.

Response: This is correct. We made this observation when trying to re-introduce NLK cDNA (without 3'UTR) into progenitors expressing siRNA against NLK. We had expected to restore erythroid failure, but we observed an unexpected accompanying lineage collapse. This observation was the catalyst to probe the literature and examine the role of miR-181. For this reason, the restored erythroid failure in Fig. 1c and d does require re-introduction of NLK_{esc} cDNA with an intact 3'UTR. We agree that gross overexpression of NLK should have effects in ribosomal-insufficient progenitors. We suspect we haven't observed that when we express our NLK (with 3'UTR) cDNA because intracellular miR-181 expression is high enough to bind to the increased NLK mRNA. MiR-181 induction is quite robust in non-erythroid lineages. It should be noted that NLK cDNA is not driven by a tremendously strong promoter, and it is hard to get a high MOI with HSPCs. We have attempted to further address the role of miR-181 in this process by including data performed in non-differentiating LCLs from DBA patients (Fig. S7). The ability of miR-181 to degrade a high proportion of NLK has been well documented in other cell types (e.g. Cichocki et al, J.Immunol 2011, Junfang et al, Hepatology 2009 and Hutchison et al, 2013).

Comment 5) Fig. 1: are the patient cells P53-deficient?

Response: p53 Expression has not been examined in these cells, but current models of DBA would predict an increase in p53, as it is a hallmark of DBA.

Comment 6) Fig. 3: some of the columns are missing, only the standard deviation bar is visible.

Response: We apologize and the error has now been corrected. The graph was from a separate experiment and has been replaced (see Fig. 7f).

Comment 7) In the section "NLK activation in RPS19-insufficiency increases Autophagy through mTOR deregulation" is too long. The transition between Raptor, mTor1 and ten autophagy is confusing. Please, fix several typos or missing punctuation.

Response: We agree. For these reasons we have omitted these data. We have retained NLK phosphorylation of Raptor, but only as a surrogate readout of NLK activity.

Comment 8) Page 20. RSK seems to be responsible for targeting S6K. However, this set of experiments is providing little information and no mechanistic relationship between DBA, NLK and Raptor.

Response: We agree and the study is beyond the central focus of this study. Consequently, it has been removed.

Comment 9) Description of the candidate drug is extensive and could easily be separated and expanded in a second manuscript. No justification why the candidate drug was not utilized *in vivo*. How does the drug target NLK? Additional targets? These topics could be expanded in a second manuscript.

Response: The drug is a poor candidate for *in vivo* studies due to poor solubility and bioavailability in animals. The drug also has many off-target effects. Therefore, we did not include *in vivo* data. Inhibition of active NLK in *in vitro* kinase assays (see Fig. S6) and Kinase Profiling (see Fig. S4B) indicate SD208 inhibits NLK directly. SD208 inhibits TGF β R1 by inhibiting ATP binding to the ATP-binding pocket, and as NLK shares a high degree of conservation in this region with TGF β R1, we predict a similar mechanism is responsible for NLK inhibition.

Comment 10) In light of this, my suggestions are to clarify how NLK is activated in erythroid cells and the role of P53 in patient cells

Response: Thank you. We have now modified the manuscript to clarify NLK activation in erythroid cells. The role of p53 protein stabilization in DBA is well characterized and we plan to understand the link between p53 and NLK activation in the future.

Specific (Minor):

- 1) Results, first paragraph: please shorten and simplify the description of the assay.
- 2) Pag. 7: We observed a mild increase in *in vitro* phosphorylation of all three substrates at day 3 compared to control cells.: this is confusing...which cells? at what time?
- 3) Pag. 11: "Modulation of NLK had no impact on erythroid colony formation in shLuc controls" should be in underlined
- 4) Pag. 3...please, rephrase: "Erythroid specificity of ribosomal insufficiency in DBA is largely due to reduced translational efficiency of genes possessing a short, complex 5'UTR that is highly upregulated..."?
- 5) Pag.8: "In the presence of doxycycline, CD34+ clonal HSC colonies expressed reduced RPS19 expression, however NLK activity showed no difference from cells cultured in the absence of doxycycline (Fig S2A)." is confusing...it looks like the activity is increased in S2A, top panel.
- 6) pag.13: "RPS19-insufficiency had no impact ON luciferase-NLK 3'UTR fusion expression"
- 7) Pag.16: baring = bearing?

8) pag.17 vs pag.18: is P53 activated at 32 or 37 C?

Response: Thank you for the suggestions. We have addressed these points in the revised manuscript.

Reviewer 3

General:

The manuscript “Diamond Blackfan is mediated by hyperactive Nemo-like kinase” by Wilkes et al attempts to show a role for a kinase known as the Nemo-like kinase (NLK) in the pathogenesis of DBA. The work is original and potentially of importance as the kinase reported here may be a candidate for developing drugs to treat DBA. While the manuscript describes a large number of experiments which appear to support a role for the NLK in DBA I have a number of concerns that I feel need to be addressed before I could recommend this manuscript for publication. These concerns are included below.

Specific:

Comment 1) The introduction introduces the reader to DBA and then to the Nemo-like kinase, two seemingly disparate lines of investigation with no obvious links given between the two subjects. The manuscript then moves to the results and a description of the Nemo-like kinase assay used throughout the study. This is followed by all the changes in Nemo-like kinase activity associated with DBA in various model systems. I was left wondering why the authors began studying the Nemo-like kinase in the first place. Did these studies arise from some type of random screen linking NLK activity with DBA? Since there is no obvious link between NLK and DBA given in the introduction (perhaps this is derived from a connection with autophagy, which is underdeveloped), some type of description for what drew the authors’ attention to NLK would be helpful in understanding the rationale for the studies outlined.

Response: We apologize for this oversight. We initially studied NLK because this kinase is known to phosphorylate the transcription factor c-Myb, which is highly expressed in hematopoietic stem and progenitor cells and plays an important role in the regulation of erythropoiesis. Phosphorylation by NLK results in ubiquitination and degradation of c-Myb. This led us to examine both the expression and activation of NLK during normal erythropoiesis and DBA models. This is now described in the Introduction section of the manuscript (2nd paragraph). We also included data with NLK phosphorylation of c-Myb in Figure 1.

Comment 2) The NLK assay is very complex. NLK is immunoprecipitated from cells, incubated with immobilized substrates and then phosphorylation of substrates assessed by ELISA. All these manipulations would seem quite challenging given the small number of cells involved given the best of circumstances, but many of these experiments are carried out on cells haploinsufficient for ribosomal proteins induced to differentiate along the erythroid lineage, where multiple studies including those from the authors themselves show that p53 is induced and many of the cells become apoptotic.

Response: We appreciate the reviewer’s comment and apologize for the lack of clarification. Limited number of cells was a challenge and often required us to push our methodologies to the threshold of reliable analysis. As outlined above, we have included new data to confirm our conclusions using multiple approaches. In specific reference to

erythroid populations with induced p53 and apoptosis, we have determined that NLK activation does not occur in the absence of p53 (Fig. S3b) and have performed a pilot study of the extent of annexin V staining at day 8 and 11 during differentiation in control and RPS19-insufficient hematopoietic progenitors derived from human cord blood CD34+ HSPCs (see Figure 1 below). Our preliminary data revealed an increase in annexin V staining from 8.92 to 20.13% in RPS19-insufficiency and this returned to 7.2% when NLK was knocked down. While this supports a model in which cells that activate NLK are more likely to undergo apoptosis, it is unclear which specific progenitor population is being affected and what percentage of cells that activate NLK undergo apoptosis. It also remains unclear if apoptosis is a direct consequence of NLK activity, or indirect and only occurs in a subset of cells in which NLK is activated. This is part of intense ongoing work examining the mechanism by which NLK activation impacts erythropoiesis (i.e. decreased proliferation, decreased translation, cell cycle arrest, or differentiation block) and will be the topic of future studies.

Data Supporting Responses to Reviewer 3, Comments 2 and 3.

Figure 1. Analysis of apoptosis by flow cytometry using DAPI and Annexin V staining in transduced CD34+ Human cord blood progenitors during erythroid differentiation. Cord blood CD34+ progenitors were transduced with shRNA against luciferase (Luc-sh) or RPS19 (RPS19-sh) and siRNA against NLK (siNLK) or a non-targeting sequence (NT) and cultured in erythroid media for 4-11 days. Apoptotic cells were assessed by Annexin-V/DAPI double staining at indicated time points. Flow cytometric

profiles represent one out of three experiments with similar results. Values represent mean \pm SEM (n = 3).

Comment 3) The authors show NLK is induced 4 days or so after knockdown of RPS19. It is important to know whether there is significant cell death at this point and if there is a subsequent reduction in the number of cells compared to baseline.

Response: We have performed annexin V analysis comparing control and RPS19-insufficient differentiating hematopoietic progenitors from transduced CD34+ human cord blood at Day 4. We observed no significant induction of cell death in any population of cells, and the expression or suppression of NLK did not influence this. Therefore, NLK activation precedes cell death and is not subsequent to it (see Figure 1). As we detect NLK activation in CD71+ and CD235+ erythroblasts at later time-points, it is unlikely NLK activation alone is required for early progenitors undergoing apoptosis. We will examine this in much greater detail in future studies.

Comment 4) Investigation into cell death and reductions in cell number is important because the authors state in supplemental Figure 1 that if there was a significant reduction in cell numbers observed extracts were supplemented with purified dephosphorylated NLK to aide in dimerization. If this is true for some measurements and not others, it would be helpful to know which samples received the supplements as they would be expected to correlate with cell number loss induced by RP haplo-insufficiency and so bias data interpretation.

Response: We apologize if this was unclear. Addition of unphosphorylated NLK prior to *in vitro* kinase assay (to amplify signals below threshold of detection) was uniformly applied to every sample in the assay. As the reviewer points out, failure to do so would prevent comparison across samples with or without signal amplification. To clarify, addition of the amplification step increased assay sensitivity, not signal intensity of individual samples.

Comment 5) There is a lack of correlation between the Western blotting reported in supplemental Figure 1 and the ELISA data. The ELISA data show huge increases in all three substrates, whereas the Western data for raptor shows a very high background and barely detectable levels of phosphorylated raptor.

Response: This is an important point. The *in vitro* kinase assay assesses the extent of phosphorylation that immunopurified NLK can catalyze in non-rate limiting *in vitro* conditions. While this allows us to make comparisons between the activity of NLK in one sample relative to another, it does not accurately reflect the ability of the kinase to phosphorylate the substrate within the cell. To address this, we have included new data that quantifies the extent of intracellular phosphorylation of Raptor at Ser863 in differentiating erythroid and non-erythroid progenitors (included as Fig. 3c). In addition, data with phosphorylation of NLK at Thr-298 in differentiating cord blood progenitors (reported to represent activated NLK) (see Figs. 2a and 3c), as well as in cord blood, mouse and DBA patient progenitors by capillary electrophoresis were added (included as Fig 6b,e,h). The “barely detectable” raptor phosphorylation in previous Fig S1C (now Fig. S2c) is the level of phosphorylated serine, detected by Western blot analysis after immunoprecipitation of raptor. This contrasts with the level phosphorylation detected by

Western blot with an antibody raised specifically against pS863 of Raptor (see Fig. S2d). While band intensity is as much dependent on antibody affinity as protein expression, we argue that the “barely detectable” signal is probably more due to suboptimal Western/co-immunoprecipitation conditions, than reflecting a lack of intracellular phosphorylation. Robust Raptor phosphorylation was detected in ribosome-insufficient erythroid progenitors with considerably less sample present (see Figs. 3c).

Comment 6) In general, the overall reproducibility of the hundreds of assay measurements reported in this manuscript is stunning given the complexity of the manipulations involved and the biological issues associated with working with dead and dying cells. In this regard it would be helpful to see some (clearly not all) of the raw data obtained for some of the key data, including any normalizations and other types of manipulations used to report the data. This would be helpful in determining whether appropriate statistical analyses were used, especially since all statistical analysis seemed to use a paired Student T test.

Response: This is an excellent point as many variables have been normalized in an attempt to facilitate meaningful comparisons. As this observation is central to the manuscript and represents calculations for both kinase assays and erythroid expansion we have included the raw data and calculations for Fig 6a and c.

Comment 7) Regarding statistical analyses, the authors state when referring to Fig. 2 that an increase in colony number to 105.5% of control was insignificant but then continue to report out their data as 72.4% etc. My point is why include the decimal in reporting the data if this does not present a significant figure.

Response: Thank you for bringing this to our attention. The number of significant figures is determined by the sensitivity of the assay and the standard deviation. As such we feel three significant figures adequately represent the sensitivity of the assay.

Comment 8) Referring to the expression data in Fig. 1 the authors mention protein levels in the text but only show RNA levels in the figure.

Response: We apologize. This was an oversight and has been corrected. Optimally, kinase activity would be compared against protein expression, but due to sample size limitations, mRNA (although not optimal) was shown as a surrogate.

Comment 9) The authors state that haploinsufficiency for ribosomal proteins induces NLK activity despite the fact that amounts of NLK are reduced. While I can accept this, the authors go on to state that it is the lower amount of NLK (not the activity of NLK) resulting from the upregulation of miR181 that prevents other hematopoietic lineages from being affected by RP haploinsufficiency. Given the disconnect between NLK levels and activity reported in the erythrocyte lineage, drawing the aforementioned conclusions for other hematopoietic lineages seems a bit of an overreach.

Response: We apologize for the confusion. As discussed above, NLK expression is not reduced in any lineage during ribosome-insufficiency (the ratio of lineages changes). NLK expression is lower in non-erythroid lineages (due to miR-181), irrespective of ribosome status. In ribosome-insufficiency, NLK is only activated in erythroid progenitors, but our data suggest this is because it is only expressed in erythroid

progenitors. We tried not to imply that reduced NLK expression is the only reason non-erythroid are less sensitive to ribosomal-insufficiency. Instead we state reduced NLK expression contributes to the lack of sensitivity.

Comment 10) Including the results from patients where the gene affected is unknown muddies the waters a bit and would seem to require some type of explanation for how to incorporate these data into the rest of the story. In the discussion and throughout the manuscript for that matter the authors equate inductions in NLK activity with ribosomal protein haploinsufficiency. Given that this induction also occurs for cells from DBA patients where the gene is unknown, I can't help but wonder whether ribosome biogenesis is affected in these cells and if not, what implications this would have on the conclusions drawn in the manuscript.

Response: Thank you. As the inclusion of the different iPSCs added confusion to the previous manuscript, we have removed data with the unknown mutation in iPSCs and controls expressing shRNA against RPS19 (Fig. S3c and d).

Comment 11) Factors downstream of activated p53 have been extensively studied in many different systems. Has NLK been encountered in any of these studies, or is its presumed role as a downstream effector of p53 unique to the erythrocyte lineage?

Response: This is an excellent question. Physical interaction between p53 and NLK has been previously reported (Zhang et al, 2014 Cell Death Differ. 21:1656-1663).

While NLK activation downstream of p53 is not well characterized, NLK activation in response to p53 effectors TNFalpha, NFkappaB and Wnt have been published (Li et al, 2014 Biochim Biophys Acta. 1843:1365-72, Smit et al, 2004 JBC. 279:17232-40, Ishitani et al, 2003 Mol and Cell Biology. 22:1379-89, Ishitani et al, 1999 Nature. 399:798-802).

Reviewers' Comments:

Reviewer #1:

Remarks to the Author:

The authors addressed our questions. However, a part of them has not been correctly addressed. I think that additional revision is required.

Reviewer1 concerns that there is the possibility that several NLK activities shown in this study might include other kinase activity. Other kinases might contribute to the ivk results. Compound might inhibit not only NLK but also other kinase. To conclude that compound works through inhibiting NLK, the authors should exclude the possibility that compound works through other molecules. However, in the rebuttal, the authors stated that testing these possibilities is not essential because this study focus on NLK. I understand that this study is focusing on NLK, but I think the authors should exclude the possibilities that other kinases contribute to the results of ivk and compound assays.

About ivk:

Reviewer1 suggested that the authors should exclude the possibility that p38 activity may affect the ivk results because p38 binds to NLK and phosphorylates c-Myb and raptor, which are thought to be NLK substrates in this study. The authors seem to think that Reviewer1 concerns that anti-NLK binds to other kinase nonspecifically and the immunoprecipitates includes other kinases, and they showed that TAK1 and p38 are not precipitated by using western blotting. I think that TAK1 and p38 should be immunoprecipitated with anti-NLK because they are the NLK-binding kinases and their experimental system (Catch and Release® kit) can be usable for co-IP. By extending the exposure time and changing antibody dilution, p38 and TAK1 should be detected in western blotting analysis.

To exclude the possibility the involvement of other kinases, Reviewer1 recommend treatment with kinase inhibitor. However, the authors stated "p38 and TAK1 inhibitors have off-target effects on other kinases, including NLK" in rebuttal, and they did not perform this recommended experiment. To my knowledge, there are no reports showing p38 and TAK1 inhibitors have off-target effects on NLK. So, I am not convinced their opinion.

To correctly show that NLK is really activated and activated NLK phosphorylates substrates, these experiments can be done:

1) Treatment of anti-NLK immunoprecipitates with p38 inhibitor (after IP, but not before IP) does not affect ivk results.

OR

1) Western blotting analysis or RadioIsotope ivk assay showing increase of NLK autophosphorylation activity (Graph data is not suitable. SDS-PAGE gel data with molecular maker is suitable to show the activity of NLK itself)

2) The ivk assays of exogenous tagged NLK WT and Kinase Dead (KD). For example, the authors express exogenous FLAG-tagged NLK WT or KD in cells expressing shLuc or shRPS19 and immunoprecipitate FLAG-NLK WT or KN and perform ivk. NLK WT would be activated in shRPS19, while NLK KD would be not activated. If NLK KD is also activated, other kinase contribute to the ivk results

Fig 2A shows NLK is completely inactive in the absence of CD71. However, in Fig 2B, NLK activity in the absence of CD71 (lane1) is as same as that in the presence of CD71 (lane 3). This means that the results of their ivk system does not reflect real NLK activity. They should explain this discrepancy.

In the ivk, the authors stated antibody against phosphor-Serine/Threonine in kinase assays, in the method section. However, in legends of several Figure, they stated that anti-pSer is used. Which is correct?

In the method section, the authors stated that phospho-Serine (#525280) was used. #525280 is a

kit including 4 antibodies. Which antibody was used? They should specify this in the method section.

The authors mentioned that phospho-Serine recognizes phospho-Threonine with similar affinity in rebuttal. That's new to me. Is there any evidence?

The authors stated that NLK siRNA did not impact related MAPKs/cdks, although they showed only that NLK siRNA did not affect their protein levels. To state this, they should show their activity but not protein levels. For example, they can use anti-phospho-MAPKs antibodies.

A part of Fig 2a panels are identical with Fig 3c panels. They are redundant. I think redundant panels should be eliminated.

About anti-pThr298:

To show NLK activation in vivo, anti-pThr298 is a good tool. So, I recommended the authors to use it.

In rebuttal, the authors stated that they showed pT298 phosphorylation levels of intracellular NLK in Fig 6beh. However, in the revised manuscript, Fig6beh shows NLK in vitro activity. Which is correct? What in vitro mean? If this in vitro means in vitro kinase assay, the authors did not answer Reviewer1's question.

About compounds:

I agree that SD inhibit NLK but I disagree SD works "through inhibiting NLK".

SD also inhibit p38 strongly. So, I feel "through inhibiting NLK" is overstatement.

Additional comments on the authors' response to reviewer #2:

Comment 2) Some of the paragraphs and experiments described are confusing and requires major efforts to understand the structure of the experiments or interpretation of the data. For instance, although it is obvious what is the goal of this assay, the supplemental Figure 1 and legend are confusing.

Authors' Response: Thank you. We have addressed this in the revised manuscript. Several variables must be accounted for, including different progenitors within a sample as well as distinct stages of differentiation across samples. Lineage cell numbers, ratios, size, viability, differentiation, proliferation and apoptosis are all influenced during normal differentiation and impacted by ribosomal-insufficiency. While every effort has been made to clarify the experiments in the text, an element of complexity is inherent in the experimental design.

My comments:

The authors have changed Figures and text significantly. The previous version of supplemental Figure 1 (Current version of supplemental Figure 2) is also improved. I feel that the problems pointed out in comment 2 have been improved.

Comment 3) One of the major points of this manuscript is that NLK activity is increased in cells with DBA associated mutations. However, in many experiments, the expression level of NLK is reduced. This suggests that another protein may be responsible for the increased activity of NLK, irrespective of expression level. This is not addressed.

Authors' Response: We have now clarified that the apparent reduction in NLK expression (in the mixed population) during ribosomal-insufficiency has to do with the increased ratio of low-NLK-

expressing non-erythroid progenitors, rather than a loss of NLK expression in erythroid progenitors. The decrease in NLK expression in these assays is the result of different hematopoietic progenitors in the population. As this was not well explained in the initial submission, we have now clarified this by revising and reorganizing the manuscript, and have included new data. Examination of NLK expression between erythroid and non-erythroid progenitors indicates that NLK expression is significantly lower in non-erythroid progenitors. We have included new data confirming that the protein level (see Figs. 1e, 2a, S3d, 3c) correlates with previous mRNA analysis (see Fig. 1f). Therefore, NLK expression does not decrease in erythroid progenitors in DBA patients. As these assays are normalized by cell numbers, the proportion of low-NLK expressing non-erythroid progenitors will be increased in the anemic samples. Similar to other kinases that autoactivate upon dimerization, an increase in concentration should increase autoactivation, particularly *in vitro*. However, as we observe no increase in NLK expression in erythroid progenitors we attribute the increase in NLK activation to deregulation of upstream regulators, rather than modulation of protein expression. Interestingly, attempted over-expression of recombinant NLK in erythroid progenitors did not dramatically increase total NLK expression but did increase in non-erythroid progenitors, perhaps suggesting a mechanism to limit NLK expression in these cells (Fig. 7f). To summarize, NLK is only expressed at high levels in erythroid progenitors (irrespective of ribosome status), and ribosomal-insufficiency leads to activation of that NLK.

My comments:

The authors' explanation is confusing and roundabout. This reviewer concerned that, although the expression level of NLK is reduced in many experiments, NLK activity is increased. And he/she requested the authors to address another protein which is responsible for the increased activity of NLK, irrespective of expression level.

Actually, not only in original Fig 1b, 1c, 2c, S2b, S2c, S3a, and S3b but also revised Fig 2c, 2d, 2e, S3b, and S3c, NLK was reduced but activated. However, the authors' response is "NLK expression does not decrease in erythroid progenitors". I disagree this response.

I can agree their hypothesis that NLK was activated by deregulation of upstream regulators even NLK was reduced, if their experimental systems are reasonable and they show the upstream regulator activating NLK. But, their kinase assay system seems to be problematic and they didn't show the mechanism. Therefore, I think the authors should show that NLK proteins are really activated by ribosomal-insufficiency. For example, the authors can check whether NLK RNAi inhibits shRPS19/11-induced c-Myb phosphorylation and degradation and p-Raptor phosphorylation and almost completely eliminates p-NLK in their kinase assay system. (A part of them had been shown in the original Fig2, but they were omitted in the revised one.)

As this reviewer also suggested, the authors should examine the mechanisms of NLK activation. For example, the authors can check whether shRPS19/11 affect the activities of known NLK activators (TAK1 and p38 etc...). At least, they can check the effect of shRPS19/11 on NLK nuclear localization.

I also feel a part of their response is rough. They should cite references when they refer to the past findings. For example, they should cite reference showing "Similar to other kinases that autoactivate upon dimerization, an increase in concentration should increase autoactivation, particularly *in vitro*".

Comment 4) Authors indicate that NLK is not deleterious in non-erythroid cells because of miR181. It is not clear why over expression of the NLK-cDNA (Figure 2) is not detrimental to non-erythroid colonies + shRPS19. Does this cDNA contain the 3'UTR region? If not, how do they reconcile this

observation? If yes, it is expected that over expression of this mRNA should have some effect also in non-erythroid cells expressing miR181 + shRPS19. Or the mRNA is degraded despite it is over expressed? Please, clarify.

Response: This is correct. We made this observation when trying to re-introduce NLK cDNA (without 3'UTR) into progenitors expressing siRNA against NLK. We had expected to restore erythroid failure, but we observed an unexpected accompanying lineage collapse. This observation was the catalyst to probe the literature and examine the role of miR-181. For this reason, the restored erythroid failure in Fig. 1c and d does require re-introduction of NLKesc cDNA with an intact 3'UTR. We agree that gross overexpression of NLK should have effects in ribosomal-insufficient progenitors. We suspect we haven't observed that when we express our NLK (with 3'UTR) cDNA because intracellular miR-181 expression is high enough to bind to the increased NLK mRNA. MiR-181 induction is quite robust in non-erythroid lineages. It should be noted that NLK cDNA is not driven by a tremendously strong promoter, and it is hard to get a high MOI with HSPCs. We have attempted to further address the role of miR-181 in this process by including data performed in non-differentiating LCLs from DBA patients (Fig. S7). The ability of miR-181 to degrade a high proportion of NLK has been well documented in other cell types (e.g. Cichocki et al, J.Immunol 2011, Junfang et al, Hepatology 2009 and Hutchison et al, 2013).

My comments:

This is a pointed question. If the authors' hypothesis is correct, overexpression of the NLK-cDNA without 3'UTR affect shRPS19-treated non-erythroid colonies. However, this was no effect (revised Fig 1cd). So, the reviewer requested the authors to explain this discrepancy. But the authors have not explained this clearly.

On the other hand, in revised Fig 7g, overexpression of the NLK-cDNA without 3'UTR affect CD11b+ shRPS19-treated non-erythroid colonies. Only this data supports the authors' hypothesis. This situation (Fig 1cd is inconsistent and only one data is consistent) weaken their hypothesis. To strengthen the hypothesis, the authors clearly explain the discrepancy in revised Fig 1cd. And they can also confirm that overexpression of the NLK-cDNA without 3'UTR or with mutated 3'UTR affect CD41a+ shRPS19-treated non-erythroid colonies.

In summary, the authors' explanations are confusing and roundabout. It is difficult for reviewers to judge how they improved their manuscript.

Reviewer #3:

Remarks to the Author:

The manuscript "Diamond Blackfan is mediated by hyperactive Nemo-like kinase" by Wilkes et al attempts to show a role for a kinase known as the Nemo-like kinase (NLK) in the pathogenesis of DBA. The work is original and potentially of importance as the kinase reported here may be a candidate for developing drugs to treat DBA.

I had a number of concerns in my initial review of this manuscript that have been adequately addressed in the revised version of the manuscript and in the author's rebuttal letter.

Having said this I think the manuscript is still in need of significant copying editing as I found slogging through the results section exceedingly laborious with many instances where I had to read and reread sections to try and discern what the authors were saying. I also found the manuscript extremely long with many experiments repeated in multiple different systems. While this repetition may be necessary to solidify the argument that the Nemo-like kinase has a role in DBA pathophysiology, the length made me less tolerant of the authors extrapolations in the

discussion suggesting that the Nemo-like kinase may be a target of glucocorticoids and leucine therapy. Thus, I think the discussion could be shortened considerably.

Specific examples of copy-editing issues are included below.

Introduction

Authors: Erythroid specificity of ribosomal insufficiency in DBA is largely due to reduced translational efficiency of genes possessing a short, complex 5'UTR that is highly upregulated, particularly GATA13.

Reviewer comment: reduced translational efficiency, highly upregulated? Are the authors meaning reduced translational efficiency of a mRNA that is normally highly upregulated during erythropoiesis?

Results

Authors: Each day, cells were collected and assayed for NLK activity and expression of NLK, RPS19 and RPL11. We observed a mild increase in in vitro phosphorylation of all three substrates at day 3 in control cells.

Reviewer comment: The second sentence mentions three substrates, which are not defined. Given that this follows a sentence where three proteins are mentioned, one could erroneously conclude that the three proteins in the previous sentence are the substrates. Only later in the paragraph does the reader find out that the three substrates are NLK, Myb, and raptor.

Authors: Several TGF β inhibitors in the murine RPS19-insufficient model were thus tested at 10 μ M and only SB431542 and SD208 significantly rescued c-Kit⁺ erythroid expansion, while six other TGF β inhibitors displayed no significant effect (Fig. 4b). Erythroid growth in murine RPS19-insufficiency by SB431542 and SD208 improved with EC50s of 5 μ M and 0.7 μ M respectively (Supplementary Fig. 4a). All of the compounds inhibit TGF β in the RPS19-insufficient murine model, as all rescued proliferation of TGF β -treated c-Kit⁺ cells to varying extents (Fig. 4c). Collectively, this indicates suppression of the TGF β pathway is not the mechanism of action for SB431542 and SD208 during of erythroid expansion in murine RPS19-insufficiency.

Reviewer comment: There has to be a more straightforward way for the authors to make the point that only a subset of TGFbeta inhibitors rescue erythroid expansion in the Rps19-insufficient model, and these are the ones that also inhibit NLK.

Authors: SD208 did not significantly impact erythroid expansion of human CB CD34⁺ HSPCs transduced with control shRNA (100% to approximately 105%), however SD208 increased erythroid expansion in both RPS19- and RPL11-insufficiency from 8.7% to 38.7% and 5.8% to 36.7%, respectively (Fig. 6a).

Reviewer comment: Sometimes results are presented as above (which I prefer) many other times they are simply as a percent of ribosome-competent controls. Consistency would improve readability.

Authors: The presence of SD208 with doxycycline improved erythropoiesis to 83.2%, 81.1% and 76.6% (Fig. 6d).

Reviewer comment: compared to what?

Discussion: In patients that qualify, stem cell transplantation can cure the disease, but is associated with a significant risk of serious, and life threatening complications such as graft versus host disease and malignancies spawned from chemotherapy prior to transplantation^{1,40}.

Reviewer comment: Stem cell transplant only cures hematological manifestations of the disease.

Additional comments on the authors' response to reviewer #2:

Comment 5) Fig. 1: are the patient cells P53-deficient?

Response: p53 Expression has not been examined in these cells, but current models of DBA would predict an increase in p53, as it is a hallmark of DBA.

The response to this comment is not satisfactory. The author's have not measured p53 levels in their patient cells but instead refer to previous work suggesting that p53 levels should be raised in these patient samples since it reported to be a hallmark of DBA. The issue of whether the authors measured p53 levels in patient cells could be a minor issue if this were a tangential point in the current manuscript. However, the authors wish to make the case that hyperactivation of Nemo-like kinase is in response to elevated p53 levels resulting from ribosomal protein haploinsufficiency. Thus, it would seem that measuring p53 levels in patient cells would close the loop in their proposed model.

- Technically, measuring p53 levels would certainly appear to be within the author's capabilities as p53 levels are manipulated in their CD34+ model system in supplemental figure 3 to make the case that Nemo-like kinase activation requires p53.
- It is conceivable that the authors may not be able to measure p53 levels in the patient cells because these samples are no longer available. If this were the case, they could at least measure p53 protein levels in their CD34+ RPS19 knockdown experiments (as reported in supplemental figure 3) and address reviewer 2's comments that way. While not optimal, this would at least close the loop in their model in one of the systems they employ.

Comment 10) In light of this, my suggestions are to clarify how NLK is activated in erythroid cells and the role of P53 in patient cells

Response: Thank you. We have now modified the manuscript to clarify NLK activation in erythroid cells. The role of p53 protein stabilization in DBA is well characterized and we plan to understand the link between p53 and NLK activation in the future.

Regarding comment 10, I agree with the authors that providing a mechanism through which p53 activates Nemo-like kinase seems beyond the scope of the current manuscript. Addressing potential mechanisms at this point would add additional speculation to the discussion, which does not seem warranted.

Reviewer #4:

Remarks to the Author:

In this manuscript, the authors reported the contribution of NLK to erythropoiesis failure in multiple DBA human and murine models. They first observed that ribosomal-insufficiency induced chronic hyperactivation of NLK, and the subsequent phosphorylation of NLK substrates including c-Myb and Raptor. They further showed that chemical and genetic inhibition of NLK increased erythroid expansion in murine and human models of DBA as well as in bone marrow stem and progenitor cells from DBA patients. While these findings are interesting, there are many concerns that need to be addressed. Even after extensive re-writing, the logic to study NLK in DBA as well as the overall flow of the study is still not well presented and not easy to follow. For example, the key point of current manuscript is about the hyperactive NLK in DBA, however they started their studies by examining the effect of NLK expression (rather than NLK activation) on c-Myb expression. Moreover, the importance of studying miRNA181 is not clear. Specific comments are listed below:

Major concerns:

1. Fig 1a shows that knockdown of RPS19 resulted in decreased c-Myb protein level. Interestingly, knockdown of NLK in the context of RPS19 knockdown restored c-Myb protein level. These results reveal the correlation between NLK expression and c-Myb protein level but do not necessarily support the authors' conclusion that expression of NLK facilitates c-Myb degradation. To reach this kind of conclusion, one needs to show that overexpression of NLK directly promotes c-Myb

degradation rather than loss of NLK restores c-Myb protein level caused by RPS19 deficiency. Additionally, it is important to understand why knockdown of RPS19 led to decreased c-Myb protein level.

2. Why for experiments shown in Fig 1a fetal liver CD34+ cells were used and the cells were differentiated for 5 days, while for Fig 1b, cord blood CD34+ cells and the cells were differentiated for 12 days?

3. What is the point to show the results under the subtitle "NLK is differentially expressed between hematopoietic lineages but is not influenced by RPS19-insufficiency"? Fig 1a already showed that RPS19 knockdown does not affect NLK expression.

4. Fig 2a shows that knockdown of RPS19 led to increased phosphorylation of NLK in CD71+ cells after differentiation of CB CD34+ cells for 10 days. Based on this finding, they conclude that NLK is phosphorylated at Thr298 and hyperactivated in erythroid progenitors. It should be noted that most of cells cultured for 10 days are erythroid precursors (erythroblasts) but not erythroid progenitors (BFU-E and CFU-E). Additionally, why the CD34+ cells were differentiated for 10 days here but were differentiated for 5 days and 12 days in Fig 1?

5. Fig 3, the figure legend is a little confusing.

6. It is not clearly whether NLK is selectively activated in erythroid progenitors or it is also activated in HSPCs.

Minor concerns:

1. c-Myb is a master transcription factor for hematopoiesis but not specifically for erythropoiesis. The statement that c-Myb is a master TF for erythropoiesis is somewhat misleading.

2. The authors gave a long introduction about NLK, some of them are not relevant to the current study.

3. NLK is widely expressed in other tissues and cell types. Targeting NLK may lead to many side effects.

4. Why the authors keep emphasizing "intracellular" phosphorylation? Is this extracellular phosphorylation?

5. Please clarify what do you mean by "chronic hyperactivation"?

6. After stating the changes in %, there is no need to repeat fold change.

7. Fig 1c, for colony assays, the authors differentiated GFP+RFP+ progenitors 12-15 days. It should be noted that human CFU-E colonies should be counted on day 7.

8. The quality of some western blots is poor and needs to be improved.

Response to the Previous Reviewer's Critiques (first revision)

We would like to thank the reviewers for their helpful comments and suggestions. We have addressed each of the reviewer's concerns point by point (see below and are editing the manuscript.

New data generated for inclusion in the re-submission include:

Fig.3d-f

Fig.5a

Fig.6b-f

Fig.7a, b, e

Supplementary Fig.2c, e – right panel

Supplementary Fig.4

Supplementary Fig.7a, b, e

Reviewer #1 Comments (Remarks to the Author):

The authors addressed our questions. However, a part of them has not been correctly addressed. I think that additional revision is required.

Reviewer1 concerns that there is the possibility that several NLK activities shown in this study might include other kinase activity. Other kinases might contribute to the ivk results. Compound might inhibit not only NLK but also other kinase. To conclude that compound works through inhibiting NLK, the authors should exclude the possibility that compound works through other molecules. However, in the rebuttal, the authors stated that testing these possibilities is not essential because this study focus on NLK. I understand that this study is focusing on NLK, but I think the authors should exclude the possibilities that other kinases contribute to the results of ivk and compound assays.

About ivk:

Reviewer1 suggested that the authors should exclude the possibility that p38 activity may affect the ivk results because p38 binds to NLK and phosphorylates c-Myb and raptor, which are thought to be NLK substrates in this study.

Response: Many different MAPK family members can phosphorylate these substrates in vitro, including ERK and p38. We have focused on addressing the role of p38 in c-Myb and Raptor phosphorylation. We have performed the recommended experiments and included other data we feel examines this possibility thoroughly, see below.

Summary: we detect no difference in p38 activity between control and DBA models, and inhibition of p38 does not impact erythropoiesis in DBA models.

Reviewer #1 Comments: The authors seem to think that Reviewer1 concerns that anti-NLK binds to other kinase nonspecifically and the immunoprecipitates includes other kinases, and they showed that TAK1 and p38 are not precipitated by using western blotting. I think that TAK1 and p38 should be immunoprecipitated with anti-NLK because

they are the NLK-binding kinases and their experimental system (Catch and Release® kit) can be usable for co-IP. By extending the exposure time and changing antibody dilution, p38 and TAK1 should be detected in western blotting analysis.

Response: p38 association with NLK has been reported previously (Ohnishi et al, 2010). Much of the experimental data is derived from over expression of epitope-tagged constructs, but endogenous NLK and p38 were detected in a complex in murine neuroblastoma cells. As we observed (even after long exposure time), p38 could not be detected after immunoprecipitation with NLK. In contrast, Ohnishi et al, detected NLK after p38 immunoprecipitation. Consequently, we have performed immunoprecipitation of p38 and also detected an association between NLK and p38 (see **Experiment 1 and Supplementary Fig. 4a**). This result has caused us to more closely examine the role of p38 in our response. We thank the reviewer for bringing this to our attention.

Summary: NLK-associated p38 was detected. NLK could be detected after immunoprecipitation with p38, but p38 could not be detected after immunoprecipitation with NLK.

Reviewer #1 Comments: To exclude the possibility the involvement of other kinases, Reviewer1 recommend treatment with kinase inhibitor. However, the authors stated “p38 and TAK1 inhibitors have off-target effects on other kinases, including NLK” in rebuttal, and they did not perform this recommended experiment. To my knowledge, there are no reports showing p38 and TAK1 inhibitors have off-target effects on NLK. So, I am not convinced their opinion.

Response: P38 association with NLK has been reported previously (Ohnishi et al, 2010). Much of the experimental data is derived from over expression of epitope-tagged constructs, but endogenous NLK and p38 were detected in a complex in murine neuroblastoma cells. As we observed (even after long exposure time), p38 could not be detected after immunoprecipitation with NLK. In contrast, Ohnishi et al, detected NLK after p38 immunoprecipitation. Consequently, we have performed immunoprecipitation of p38 and also detected an association between NLK and p38 (see **Experiment 1 and Supplementary Fig. 4a**). This result has caused us to more closely examine the role of p38 in our response. We thank the reviewer for bringing this to our attention.

Experiment 1: Immunoprecipitation with NLK, p38, or IgG antibodies

Kp53A1 leukemia cells were prepared using the same lysis conditions utilized in our in vitro kinase (ivk) assays. Approximately 500 μg of protein was immunoprecipitated with IgG, or antibodies against p38 or NLK at 4°C overnight. Immunoprecipitates were subjected to SDS-PAGE and probed by Western blot analysis for p38 or NLK. As a positive control, 2% of the lysate (input) was run – far left lane.

Conclusion: NLK-p38 complexes from cell lysates were not detected after immunoprecipitation with NLK antibody, but were detected after immunoprecipitation with p38 antibody. Therefore, it is unlikely that immunopurified NLK contains p38 in the complex.

No direct association between TAK1 and NLK has been reported (even through yeast two hybrid) (Tohru and Shizuka Ishitani, 2013) and complex association can only be detected in the presence of a scaffold, such as TAB2 (Li et al, 2010 JBC) or HIPK2 (Kaneii Ishii, et al). TAK1 has also not been shown to be able to phosphorylate c-Myb or Raptor, and we previously found no association of TAK1 with immunoprecipitated NLK.

Reviewer #1 Comments: To correctly show that NLK is really activated and activated NLK phosphorylates substrates, these experiments can be done:

- 1) Treatment of anti-NLK immunoprecipitates with p38 inhibitor (after IP, but not before IP) does not affect ivk results.

Response: We have now examined the contribution of p38 in our NLK in vitro kinase assays (after IP) using p38 inhibitors. Eight p38 inhibitors were titrated to determine the IC₅₀ of each compound against activated NLK or p38. Each of the inhibitors have previously reported IC₅₀ values for both p38 and NLK, with three inhibitors reported to have no off-target inhibition of NLK.

Summary: In our experiments, NLK was inhibited at a similar IC₅₀s as reported, and three specific p38 inhibitors had no effect on NLK activity (**Experiment 2 and Fig. 3f**).

Experiment 2. Inhibition of NLK from RPS19-insufficient cells is not associated with P38 inhibition.

A

Compound	NLK		P38	
	Reported	Observed	Reported	Observed
SB203580	180 nM	200 nM	70 nM	80 nM
SB239063	79 nM	64 nM	44 nM	54 nM
BIRB-796	1.8 μ M	1.5 μ M	14 nM	21 nM
AMG-548	3.1 μ M	3.6 μ M	0.3 nM	0.4 nM
TAK-715	7.8 μ M	9.3 μ M	7.1 nM	12.6 nM
VX-745	No Effect	3.8 μ M	10 nM	14.4 nM
SCIO-469	No Effect	No Effect	9 nM	4.6 nM
PH797804	No Effect	No Effect	5.8 nM	4.9 nM

B.

A panel of eight small molecule p38 inhibitors were titrated into in vitro kinase assays in the presence of immunopurified NLK or P38, as well as ATP, Mg²⁺ and dephosphorylated c-Myb substrate. Determining the extent of c-Myb phosphorylation at each drug concentration allowed us to calculate the IC₅₀ for NLK and p38 for each compound. (A) Comparison between our results and previously reported results are presented in the table. (B) Data are shown as figures with IC₅₀ values represented as vertical lines along a concentration gradient comparing NLK and p38 inhibitors. Kinase activity is represented as blue and the extent of inhibition is depicted in white. Our observed values (shown in orange) can be easily compared with documented IC₅₀ values (shown in blue) for each compound against each target.

Conclusion: Our results suggest that p38 inhibitors do not inhibit NLK activity at the same IC₅₀ as p38 activity. Therefore, it is unlikely that NLK-associated p38 is contributing significantly to NLK activity in our assays.

Additional experiments were performed to examine if p38 contributes to the observed kinase activity.

- 1) STAT3 has been reported to be a substrate of NLK and not p38 (Ohnishi et al, 2010), so we compared phosphorylation of immobilized STAT3 by immunoprecipitated NLK and p38. NLK from RPS19-insufficient hematopoietic progenitor cells increased STAT3 phosphorylation relative to controls.

Summary: No difference in phosphorylation was observed between p38 from either condition (**Experiment 3 and Supplementary Fig. 4c**).

Experiment 3. Immunoprecipitated NLK from RPS19-insufficient cells can phosphorylate STAT3 *in vitro*.

P38 and NLK were immunoprecipitated from RPS19-insufficient (shRPS19) or control (shLuc) cells derived from human cord blood CD34⁺ hematopoietic stem and progenitor cells at Day 7. Immunopurified p38 or NLK were incubated in the presence of ATP, Mg²⁺ and immobilized, dephosphorylated STAT3 to assess the ability of each kinase to phosphorylate STAT3 *in vitro*.

Conclusion: NLK from RPS19-insufficient progenitor cells had significantly increased enzymatic activity towards STAT3 (p=0.0024). No increase in STAT3 phosphorylation is detected with p38 (p=0.0838). Therefore, increase in STAT3 phosphorylation is due to NLK, not p38 activity.

- 2) If p38 was contributing to the observed increase in phosphorylation of substrates, we reasoned p38 activity may be increasing similarly.

Summary: Examination of immunoprecipitated P38 from control and RPS19-insufficiency did not reveal an increase in p38 kinase activity against NLK, c-Myb or Raptor (**Experiment 4 and Supplementary Fig. 4b**).

Experiment 4. Immunopurified NLK but not p38 activity is increased in RPS19-insufficient cells.

P38 and NLK were immunoprecipitated from RPS19-insufficient (shRPS19) or control (shLuc) cells derived from human cord blood CD34+ hematopoietic stem and progenitor cells at Day 7. Immunopurified p38 or NLK was incubated in the presence of ATP, Mg²⁺ and immobilized, dephosphorylated critical substrates in hematopoiesis (NLK, c-Myb and Raptor) to determine whether RPS19-insufficiency affects NLK or P38 *in vitro* activity.

Conclusion: While RPS19-insufficiency increased NLK-mediated phosphorylation of NLK, c-Myb and Raptor, P38 activity did not increase these substrates.

- 3) A number of MAPK kinases can phosphorylate NLK, c-Myb and Raptor *in vitro* but with different specificities. We compared specificity of active P38 and NLK and observed a clear difference.

Summary: P38 phosphorylated NLK and c-Myb with high specificity, but Raptor with low specificity. In contrast, NLK phosphorylated Myb and Raptor with high specificity but low specificity against itself (**Experiment 5 and Supplementary 4d**).

Experiment 5. Activated p38 and NLK have different kinase specificity for NLK, c-Myb, Raptor and STAT3.

Immunopurification of activated NLK and p38 were incubated in the presence of ATP, Mg^{2+} and immobilized, dephosphorylated substrates. After 30 mins, the degree of substrate phosphorylation was determined. Of the four substrates examined, NLK was the most phosphorylated substrate detected with p38. The extent of phosphorylation of the other substrates by p38 was normalized to NLK as substrate, i.e. NLK-100%, c-Myb-79%, Raptor-33%, STAT3-18%. Raptor was the most phosphorylated substrate of NLK. Consequently, phosphorylation of other substrates by NLK is expressed as a percentage of NLK-mediated Raptor phosphorylation, i.e. NLK-37%, c-Myb-80%, Raptor-100%, STAT3-39%.

Conclusion: While p38 and NLK can phosphorylate NLK, c-Myb and Raptor in vitro, we observe different specificities between immunoprecipitated NLK and p38. This would suggest that NLK activity is most likely not due to NLK-associated p38.

Reviewer #1 Comments:

1) Western blotting analysis or Radiolotope ivk assay showing increase of NLK autophosphorylation activity (Graph data is not suitable. SDS-PAGE gel data with molecular maker is suitable to show the activity of NLK itself)

Response: We performed the Western blot analysis as requested. Purified NLK incubated with immunopurified NLK from treated samples, were separated by SDS-PAGE and immunoblotted with pThr298-NLK antibody. Molecular markers are included and NLK phosphorylation at Thr298 is increased in RPS19-insufficient cells, but not in control, when co-expressed with siRNA against NLK, or treated with SD208. In contrast, phosphorylation of NLK is not suppressed by the p38-specific inhibitor PH797804. Equal loading of NLK substrate is ensured by Ponceau staining prior to Western blot analysis (**Experiment 6**).

Summary: NLK phosphorylation by Western blot analysis correlated closely with ELISA-based assay.

Experiment 6. SDS-PAGE and pThr298-NLK Western blot analysis of NLK activation in RPS19-insufficient cells.

NLK was immunoprecipitated from RPS19-insufficient (shRPS19) and control (shLuc) cells derived from human cord blood CD34+ hematopoietic stem and progenitor cells at Day 7. Cells were transduced with siRNA against NLK or control lentivirus, or incubated in the presence of NLK inhibitors SD208 or PH797804. Immunopurified NLK was incubated in the presence of ATP, Mg²⁺ and dephosphorylated NLK. After 30 min incubation at 37°C, the kinase/substrate mix was subjected to SDS-PAGE and transferred to immobilon-P membrane. Ponceau staining (right panel) ensured equal loading. The membrane was probed for pThr298-NLK by Western blot analysis and the result is superimposed over the colorimetric image to allow correlation of the bands with protein size markers (as requested).

Conclusion: SDS-PAGE followed by Western blot analysis with pThr298-NLK antibody shows comparable results as ELISA-based *in vitro* kinase assay.

Reviewer #1 Comments: The *in vitro* assays of exogenous tagged NLK WT and Kinase Dead (KD). For example, the authors express exogenous FLAG-tagged NLK WT or KD in cells expressing shLuc or shRPS19 and immunoprecipitate FLAG-NLK WT or KN and perform *in vitro* kinase assay. NLK WT would be activated in shRPS19, while NLK KD would be not activated. If NLK KD is also activated, other kinases contribute to the *in vitro* kinase assay results.

Response: We have now performed this experiment (see **Experiment 7 and Supplementary Fig. 4f**). Wild type and Kinase-deficient NLK were expressed after endogenous NLK was knocked down and immunoprecipitated kinase-deficient NLK could not phosphorylate substrates. It should also be noted that a constitutively-active NLK mutant increases c-Myb degradation (**Experiment 8 and Fig. 7e**) and erythroid failure (**Experiment 9 and Fig. 5a**).

Experiment 7. Immunoprecipitated Kinase-deficient NLK does not catalyze phosphorylation of NLK, Myb, Raptor.

NLK was immunoprecipitated from RPS19-insufficient (shRPS19) or control (shLuc) cells derived from human cord blood CD34+ hematopoietic stem and progenitor cells at Day 7. Cells were transduced with shRNA against RPS19 or control lentivirus, cultures were transduced with siRNA against NLK alone, or with either a wild type (WTesc) or kinase-deficient (KDesc) siRNA-resistant NLK cDNA (with intact 3'UTR). NLK activity was determined by *in vitro* kinase assays examining NLK, c-Myb and Raptor as substrates. NLK mRNA expression was assessed by qRT-PCR.

Conclusion: No substrate phosphorylation was detected after immunoprecipitation of kinase-deficient NLK, indicating that observed NLK-mediated substrate phosphorylation is most likely mediated by NLK directly and not by an NLK-associated kinase.

Experiment 8. Constitutively-active NLK induces proteasome-dependent degradation of c-Myb.

Lentiviral constructs containing Wild type (WT-NLK) or constitutively-active (CA-NLK) NLK cDNA were generated (NLK 3'UTR present). Human cord blood CD34+ hematopoietic stem and progenitor cells were transduced with the appropriate lentiviral constructs and differentiated for 7 days. Cells were treated in the presence or absence of the proteasomal inhibitor lactacystin. Triplicate samples were pooled and the CD71+

population was assessed for c-Myb, NLK and GAPDH by Western blot analysis after cell lysis and protein normalization.

Conclusion: Over-expression of wild type NLK did not reduce c-Myb expression in control or RPS19-insufficiency. In contrast, constitutively-active NLK reduced c-Myb expression, even in ribosome-competent cells.

Experiment 9. Overexpression of recombinant cDNA expressing constitutively-active (but not wild type)

NLK reduces CD235+ erythroid and CD11b+ myeloid expansion. Fusion of NLK 3'UTR to cDNA restores erythroid specificity.

Lentiviral constructs containing constitutively-active NLK (CA-NLK) or Wild type NLK (WT-NLK) cDNA were generated with the NLK 3'UTR present or not included in the sequence. Human cord blood CD34+ hematopoietic stem and progenitor cells were transduced and differentiated for 12 days and CD235+ erythroid and CD11b+ Myeloid expansion calculated by multiplying viable cell counts by % CD235+ or CD11b+ respectively. Results are displayed as a percentage of cells of each lineage relative to untreated controls.

Results are displayed as a percentage of cells of each lineage relative to untreated controls.

Conclusion: Over-expression of wild type NLK is not sufficient to suppress erythroid expansion (top left). Expression of constitutively-active NLK (without an intact 3'UTR) suppresses erythroid AND myeloid expansion (top & bottom left). We hypothesize that NLK-mediated effects are limited to erythroid cells in DBA because NLK is not expressed at high enough levels because of miR-181 binding to the 3'UTR. Fitting with our hypothesis (see model on the right), transduction of CA-NLK cDNA that can't be regulated by miR-181 in myeloid cells (no 3'UTR) leads to suppression of both erythroid and non-erythroid lineages because NLK expression is high in all lineages. Importantly, expression of CA-NLK with an intact 3'UTR restores lineage specificity.

Reviewer #1 Comments: Fig 2A shows NLK is completely inactive in the absence of CD71. However, in Fig 2B, NLK activity in the absence of CD71 (lane1) is as same as that in the presence of CD71 (lane 3). This means that the results of their ivk system does not reflect real NLK activity. They should explain this discrepancy.

Response: Thank you and we appreciate the reviewer's concerns. This could reflect either artificially high signal in CD71- cells in the kinase assay, or artificially high staining of pThr-298-NLK in control CD71+ cells. In addition, increased pThr-298-NLK

phosphorylation in RPS19-insufficient cells is lower by *in vitro* kinase assays. We examined the specificity of the pThr298-NLK antibody against phosphorylated and unphosphorylated NLK. Antibody affinity to stimulated NLK was approximately 6-fold higher than purified, dephosphorylated NLK. However, the antibody still bound dephosphorylated, unstimulated NLK with approximately twice the affinity of other proteins and non-treated plates (see **Experiment 10**). Non-specific binding to non-phosphorylated NLK could also explain why control CD71+ cells (high NLK expression) demonstrate a much higher signal than CD71- populations.

We do recognize that the *in vitro* assay may not quantify increases in kinase activity with complete accuracy, but does detect significant changes in activity in small populations of cells that limit or prevent analysis using conventional approaches, such as differentiating hematopoietic progenitor populations. We contest that NLK is activated in ribosome-insufficient erythroid progenitors, but acknowledge it is difficult to state the exact increase in NLK activity within each cell or population of cells.

Summary: Non-specific binding of the anti-pThr298-NLK antibody to unphosphorylated NLK contributes to higher basal levels and reduced induction in Western analysis versus kinase assay.

Experiment 10. Anti-pThr298-NLK antibody cross reacts with non-phosphorylated NLK but to a lesser extent.

Immunopurified NLK was biotinylated and immobilized on streptavidin-coated 96-well plates. After dephosphorylation by phosphatase treatment, plates were left untreated or incubated in the presence of activated NLK, Mg²⁺ and ATP. After thorough washing, plates were incubated with anti-pThr298-NLK antibody and assessed by ELISA. For

comparison, immobilized dephosphorylated GAPDH, c-Myb and Raptor and wells with no substrate present, were assessed for anti-pThr298-NLK binding. Values are normalized to the extent of binding detected for unphosphorylated NLK.

Conclusion: Anti-pThr298-NLK antibody has approximately 6-fold greater affinity for phosphorylated NLK than unphosphorylated NLK. However, the antibody has approximately twice the affinity for unphosphorylated NLK as it does over other proteins or untreated plates. This suggests 1) cells expressing higher levels of NLK will demonstrate higher background than cells expressing low NLK levels, and 2) NLK induction will appear less with pThr298-NLK (compared to *in vitro* kinase assay) as basal binding is not restricted exclusively to active NLK.

Reviewer #1 Comments: In the ivk, the authors stated antibody against phospho-Serine/Threonine in kinase assays, in the method section. However, in legends of several Figure, they stated that anti-pSer is used. Which is correct?

Response: The initial studies were performed using the phosphoserine antibody (50ug #525280 Lot:B32644, Millipore/Sigma). Subsequent experiments were performed using a mix of p-Thr (H2) HRP (#sc5267 from Santa Cruz Biotechnology) and phosphoserine HRP (#ab9334 from Abcam).

Summary: We modified the Materials and Methods section to more accurately describe how the assay evolved over time.

Reviewer #1 Comments: In the method section, the authors stated that phospho-Serine (#525280) was used. #525280 is a kit including 4 antibodies. Which antibody was used? They should specify this in the method section.

Response: We apologize for the confusion. It is our understanding the kit of four antibodies is #525282, which is comprised of #525280, 81, 83 and 84.

Summary: The company documentation is provided below.

Documentation on phosphoserine Antibody #525280.

Response to Reviewer #1, comment 6: In the method section, the authors stated that phospho-Serine (#525280) was used. #525280 is a kit including 4 antibodies. Which antibody was used? They should specify this in the method section.

United States / English Quick Purchase | My Favorites | Contact Us

MILLIPORE SIGMA

Search product, CAS, keyword... Login Register Cart

Products | Services | Documents | Responsibility | Support | About Us | About Our Brands

Home > Life Science Research > Protein Detection and Quantification > Immunossays > Enzyme-linked Immunosorbent Assay (ELISA) > Complete ELISA Kits > PhosphoDetect™ Phosphoserine Detection Kit

525282 Sigma-Aldrich
PhosphoDetect™ Phosphoserine Detection Kit

This PhosphoDetect™ Phosphoserine Detection Kit is validated for use in immunoblotting, immunoprecipitation, ELISA.

PhosphoDetect™ Phosphoserine Detection Kit MSDS (material safety data sheet) or SDS, CoA and CoQ, dossiers, brochures and other available documents.

SDS CoA Data Sheet

View Pricing & Availability

Overview Supporting Documentation Related Product & Applications

Overview

Description
Product Information
Applications
Biological Information
Storage and Shipping Information
Supplemental Information
Pricing & Availability

Pricing & Availability

Catalog Number	Availability	Packaging	Qty/Pack	Price	Quantity	ASL ID
525282-KIT	Limited Availability	Glass bottle	1 kit	USD 661.00 Log In to See Your Pricing	<input type="text"/>	<input type="button" value="Check Availability"/> <input type="button" value="Add To Cart"/>

Description

Overview

A set of four different monoclonal antibodies specific for serine phosphorylation sites. Recognition is dependent on phosphorylation and the surrounding amino acid motif. Recognizes serine phosphorylated proteins in Arabidopsis, bacteria, chicken, human, maize, mouse, rat, tobacco, Xenopus, yeast, and zebrafish. Phosphorylation patterns in cell extracts may differ when probed with different antibodies.

Catalogue Number 525282
Brand Family Carbocem®

Description

Clone Name	Host	Isotype	Immunoconjugation	Immunoconjugation	ELISA
1C8	Mouse	IgM1	0.1-1 µg/ml	1-10 µg/10 ⁶ cells	0.05 µg/ml
4A3	Mouse	IgM1	0.1-1 µg/ml	1-10 µg/10 ⁶ cells	0.05 µg/ml
4A9	Mouse	IgM1	0.1-1 µg/ml	1-10 µg/10 ⁶ cells	0.05 µg/ml
16B4	Mouse	IgM1	0.1-1 µg/ml	1-10 µg/10 ⁶ cells	0.05 µg/ml

Application Data

1C8 4A3 4A9 16B4
Detection of phosphoserine-containing antibodies by immunoblotting. Samples: Whole cell lysate from A431 cells treated with pervanadate (all lanes). Primary antibodies: PhosphoDetect™ Anti-Phosphoserine Mouse mAb (16B4) (Cat. No. 525280) (1 µg/ml), PhosphoDetect™ Anti-Phosphoserine Mouse mAb (1C8) (Cat. No. 525281) (1 µg/ml), PhosphoDetect™ Anti-Phosphoserine Mouse mAb (4A3) (Cat. No. 525283) (1 µg/ml), and PhosphoDetect™ Anti-Phosphoserine Mouse mAb (4A9) (Cat. No. 525284) (1 µg/ml). Detection: chemiluminescence.

1C8 4A3 4A9 16B4
Detection of phosphoserine-containing antibodies by immunoblotting. Samples: Extract from rabbit muscle*. Primary antibodies: PhosphoDetect™ Anti-Phosphoserine Mouse mAb (16B4) (Cat. No. 525280) (1 µg/ml), PhosphoDetect™ Anti-Phosphoserine Mouse mAb (1C8) (Cat. No. 525281) (1 µg/ml), PhosphoDetect™ Anti-Phosphoserine Mouse mAb (4A3) (Cat. No. 525283) (1 µg/ml), and PhosphoDetect™ Anti-Phosphoserine Mouse mAb (4A9) (Cat. No. 525284) (1 µg/ml). Detection: chemiluminescence. *Positive control supplied with the antibody.

Product Information

Form Lyophilized

Formulation 25 µg each antibody lyophilized from 2x PBS, PEG, sucrose and 200 µl control phosphoproteins lyophilized 20 mM Na₂HPO₄, 0.1% SDS

Kit contents 25 µg of each of the Phosphoserine Antibodies including clones 1C8, 4A3, 4A9, and 16B4, Rabbit Muscle Positive Control, and a data sheet.

Positive control Phosphoproteins purified from rabbit muscle (included)

Preservative ≤ 0.1% sodium azide (antibodies only)

Applications

Key Applications Enzyme-Linked Immunosorbent Assay
Immunoblotting/Western Blotting
Immunoprecipitation

525282 Sigma-Aldrich
PhosphoDetect™ Phosphoserine Detection Kit

Detection of phosphoserine-containing antibodies by immunoblotting. Samples: Whole cell lysate from A431 cells treated with pervanadate (all lanes). Primary antibodies: PhosphoDetect™ Anti-Phosphoserine Mouse mAb (16B4) (Cat. No. 525280) (1 µg/ml), PhosphoDetect™ Anti-Phosphoserine Mouse mAb (1C8) (Cat. No. 525281) (1 µg/ml), PhosphoDetect™ Anti-Phosphoserine Mouse mAb (4A3) (Cat. No. 525283) (1 µg/ml), and PhosphoDetect™ Anti-Phosphoserine Mouse mAb (4A9) (Cat. No. 525284) (1 µg/ml). Detection: chemiluminescence.

Reviewer #1 Comments: The authors mentioned that phospho-Serine recognizes phospho-Threonine with similar affinity in rebuttal. That's new to me. Is there any evidence?

Response: We appreciate the concern that serine phosphorylation of NLK may reflect that p38 is present in the assay as p38 is reported to phosphorylate NLK at Ser510 (Ohnishi et al, 2010). We did detect phosphoserine on NLK when immunoprecipitating p38, but there was no significant difference between control and RPS19-insufficient progenitor cells. Phosphoserine detection increased after incubation with immunoprecipitated NLK from RPS19-insufficient cells. Given that NLK is homologous with p38, we speculate NLK may be able to phosphorylate itself at serine residues, or the antibody also detects Thr298 phosphorylation (albeit with less specificity than serine residues).

Summary: After consultation with a number of experts who have used the antibody, and Millipore technical support, we were informed there is cross-reactivity with phosphothreonine but significantly less than phosphoserine. Because of this ambiguity we did not feel we could claim phosphorylation was exclusively at serine or threonine residues. More recent assays utilize a mix of both antibodies to avoid this. We apologize for the confusion.

Reviewer #1 Comments: The authors stated that NLK siRNA did not impact related MAPKs/cdks, although they showed only that NLK siRNA did not affect their protein levels. To state this, they should show their activity but not protein levels. For example, they can use anti-phospho-MAPKs antibodies.

Response: To determine if silencing NLK impacts JNK, ERK1/2 and P38 activation we stimulated Kp53A1 myeloid leukemia cells stably expressing non-targeting or siRNA against NLK that were stimulated with SCF and Epo (stimulate JNK, ERK and P38 in these cells).

Summary: No differences were observed (see **Experiment 11 and Supplementary Fig. 2d**).

Experiment 11. siRNA against NLK does not affect activation of p38, JNK or ERK phosphorylation.

Kp53A1 cells were transfected with a non-targeting sequence (NT), siRNA against NLK (siNLK), or nontransfected (Unt = control). The sequence of the siRNA is listed above the Western blot panels. Transfected cells were cultured for 48 hours prior to lysis. Lysates were subjected to SDS-PAGE and protein expression of the listed proteins was determined by Western blot analysis. Phosphorylation of P38, JNK and ERK were similarly assessed utilizing phosphospecific antibodies against each, which is shown in the red box.

Conclusion: Expression of siRNA against NLK does not appear to affect activation of p38, JNK or ERK phosphorylation to a substantial degree.

Reviewer #1 Comments: A part of Fig 2a panels are identical with Fig 3c panels. They are redundant. I think redundant panels should be eliminated.

Response/Summary: Thank you. We have removed the redundant panels.

Reviewer #1 Comments: About anti-pThr298:

To show NLK activation in vivo, anti-pThr298 is a good tool. So, I recommended the authors to use it.

Response/Summary: We agree and have incorporated this in the revised manuscript/figures.

Reviewer #1 Comments: In rebuttal, the authors stated that they showed pT298 phosphorylation levels of intracellular NLK. However, in the revised manuscript, Fig6beh shows NLK in vitro activity. Which is correct? What in vitro mean? If this in vitro means

in vitro kinase assay, the authors did not answer Reviewer1's question.

Response: We apologize for the confusion. We did not have enough sample to perform conventional Western Blot analyses. Consequently, to respond to the reviewer's comment, we utilized capillary Western Blot analysis technology using the pThr298 antibody. Technical information regarding the technology can be found at https://www.proteinsimple.com/peggy_sue.html.

Summary: Phosphorylation of NLK at Thr298 was determined by capillary Western blot analysis.

Reviewer #1 Comments: I agree that SD inhibits NLK but I disagree SD works "through inhibiting NLK".

Response: We appreciate the reviewer's comment; however, we have no data to support another mechanism of action. When NLK is knocked down, no additional erythropoiesis effect is observed by adding SD208 (shown previously). This suggests the possibility that an NLK-associated kinase is the target of SD208. In addition, the expression of a constitutively-active NLK reduces erythroid expansion (see **Experiment 9 and Fig. 5a**).

Summary: While we cannot rule out that SD208 may have off-target effects that impact erythropoiesis, our data suggest that inhibition of NLK is a major component of restored erythropoiesis.

Reviewer #1 Comments: SD also inhibit p38 strongly. So, I feel "through inhibiting NLK" is overstatement.

Response: Thank you and we agree. SD208 has a reported IC50 for p38 or 850nM. This is a 2-fold increase compared to our observed IC50 for NLK (440nM). To examine the role of p38 in RPS19-insufficient cells we utilized two p38 specific inhibitors that do not cross react with NLK (PH797804 and SCIO-469) and treated RPS19-insufficient erythroblasts at 100nM. We observed no increase in erythropoiesis in RPS19-insufficiency however p38 was significantly inhibited (see **Experiment 12 and Supplementary Fig. 2c**).

As activated p38 can directly phosphorylate over 100 substrates (Igea and Nebrada, 2015) and is activated in erythropoiesis, we cannot rule out the possibility that NLK-associated p38 that is translocated into the nucleus has an impact in erythropoiesis that is not assessed in our in vitro models, perhaps including niche interactions or erythroid enucleation and maturation. This is not limited to p38 as any NLK-associated protein will be translocated along with activated NLK. This does not subtract from the importance of NLK activation in disease pathogenesis, instead emphasizing that the impact of NLK activation is beyond direct substrate phosphorylation.

Summary: we have no data that indicates any other kinase other than NLK is the functional target of SD208 in our DBA models.

Experiment 12. The p38 specific inhibitors do not improve the erythroid defect in RPS19-insufficient hematopoietic stem and progenitor cells.

Human cord blood CD34+ progenitors were transduced with shRNA against RPS19 (shRPS19) or luciferase (shLuc), and differentiated in erythroid media in the presence or absence of recommended concentrations of SD208, SB203580, PH797804 and SCIO-469 for 12 days. Cells were counted and analyzed for cell surface expression of CD235. Effective inhibition of p38 activity was assessed by lysis of treated controls and Western blot analysis of p38.

Conclusion: Compounds that inhibit both NLK and p38 significantly increased erythroid expansion ($p=0.0056$ and $p=0.0434$ for SD208 and SB203580, respectively). The p38 specific inhibitors did not increase erythroid expansion.

Additional comments on the authors' response to reviewer #2:

Reviewer Comment 2) Some of the paragraphs and experiments described are confusing and requires major efforts to understand the structure of the experiments or interpretation of the data. For instance, although it is obvious what is the goal of this assay, the supplemental Figure 1 and legend are confusing.

Authors' Previous Response: Thank you. We have addressed this in the revised manuscript. Several variables must be accounted for, including different progenitors within a sample as well as distinct stages of differentiation across samples. Lineage cell numbers, ratios, size, viability, differentiation, proliferation and apoptosis are all influenced during normal differentiation and impacted by ribosomal-insufficiency. While

every effort has been made to clarify the experiments in the text, an element of complexity is inherent in the experimental design.

New Reviewer's Comments: The authors have changed Figures and text significantly. The previous version of supplemental Figure 1 (Current version of supplemental Figure 2) is also improved. I feel that the problems pointed out in comment 2 have been improved.

Author's New Response/Summary: We thank the reviewer for this helpful suggestion. We apologize and have significantly revised the manuscript.

Reviewer's Comment: One of the major points of this manuscript is that NLK activity is increased in cells with DBA associated mutations. However, in many experiments, the expression level of NLK is reduced. This suggests that another protein may be responsible for the increased activity of NLK, irrespective of expression level. This is not addressed.

Authors' Previous Response: We apologize for the confusion and have now clarified that the apparent reduction in NLK expression (in the mixed population) during ribosomal-insufficiency is due to the increased ratio of low-NLK-expressing non-erythroid progenitors, rather than a loss of NLK expression in erythroid progenitors. The decrease in NLK expression in these assays is the result of different hematopoietic progenitors in the population. As this was not well explained in the initial submission, we have now clarified this by revising and reorganizing the manuscript, and have included new data. Examination of NLK expression between erythroid and non-erythroid progenitors indicates that NLK expression is significantly lower in non-erythroid progenitors. We have included new data confirming that the protein level (see Figs. 1e, 2a, S3d, 3c) correlates with previous mRNA analysis (see Fig. 1f). Therefore, NLK expression does not decrease in erythroid progenitors in DBA patients. As these assays are normalized by cell numbers, the proportion of low-NLK expressing non-erythroid progenitors will be increased in the anemic samples. Similar to other kinases that autoactivate upon dimerization, an increase in concentration should increase autoactivation, particularly in vitro. However, as we observe no increase in NLK expression in erythroid progenitors we attribute the increase in NLK activation to deregulation of upstream regulators or post-translational modification, rather than protein expression. Interestingly, attempted over-expression of recombinant NLK in erythroid progenitors did not dramatically increase total NLK expression but did increase in non-erythroid progenitors, perhaps suggesting a mechanism to limit NLK expression in these cells (Fig. 7f). To summarize, NLK is only expressed at high levels in erythroid progenitors (irrespective of ribosome status), and ribosomal-insufficiency leads to activation of that NLK.

Reviewer comments: The authors' explanation is confusing and roundabout. This reviewer concerned that, although the expression level of NLK is reduced in many experiments, NLK activity is increased. And he/she requested the authors to address

another protein which is responsible for the increased activity of NLK, irrespective of expression level.

Author's New Response/Summary: Again, we apologize and would like to re-emphasize that NLK expression is not reduced in RPS19-insufficient cells. The percentage of high NLK-expressing erythroblasts is reduced within mixed population, which is reflected when NLK expression is assessed within the mixed population. Concerns regarding another kinase, either independently, or associated with NLK being responsible for observed kinase activity (particularly p38) have been investigated and new data have been added to the manuscript.

Reviewer's Comment 3a) Actually, not only in original Fig 1b, 1c, 2c, S2b, S2c, S3a, and S3b but also revised Fig 2c, 2d, 2e, S3b, and S3c, NLK was reduced but activated. However, the authors' response is "NLK expression does not decrease in erythroid progenitors". I disagree this response.

Author's New Response/Summary: We have found no evidence that NLK expression is reduced in RPS19-insufficient cells in any lineage. A reduction in NLK expression is only observed when assessing a mixed population in which low NLK-expressing non-erythroid progenitors are proportionally more abundant. The same thing would be observed for any erythroid specific marker (e.g. CD71 or CD235).

Reviewer Comment 3b) I can agree their hypothesis that NLK was activated by deregulation of upstream regulators even NLK was reduced, if their experimental systems are reasonable and they show the upstream regulator activating NLK. But, their kinase assay system seems to be problematic and they didn't show the mechanism. Therefore, I think the authors should show that NLK proteins are really activated by ribosomal-insufficiency. For example, the authors can check whether NLK RNAi inhibits shRPS19/11-induced c-Myb phosphorylation and degradation and p-Raptor phosphorylation and almost completely eliminates p-NLK in their kinase assay system. (A part of them had been shown in the original Fig2, but they were omitted in the revised one.)

Author's New Response: As suggested, we have included the data in the manuscript that no phosphorylation of Raptor or Myb occurs in *in vitro* kinase assays if NLK expression was suppressed by siRNA. We have been unable to detect any significant contribution of p38 (or ERK, JNK, TAK1, cdk1 or cdk2) to *in vitro* or intracellular phosphorylation of c-Myb or Raptor. We have expressed kinase-insufficient NLK into RPS19-insufficient cultures and observed substrate phosphorylation and improved erythroid expansion (see **Experiment 13**). We have introduced a constitutively-active NLK into healthy controls and reduced erythroid expansion as well as resulting in c-Myb degradation. We have correlated *in vitro* activation with autophosphorylation at Thr298 and dimerization using FRET. We have used a wide selection of inhibitors both examining *in vitro* NLK activity and erythroid expansion and observed effects that correlate closely with documented and observed IC50s for NLK (see **Experiment 14 and Supplementary Fig. 4e**). As requested, we have provided an *in vitro* kinase assay

performed using SD-PAGE and western blotted for pThr298-NLK (superimposed with protein markers) with ponceau staining provide to ensure equal loading of NLK as substrate. Even if the in vitro assay system is not optimal, the central observation that NLK has greater activity in RPS19-insufficient cells has been assessed using a number of other readouts (e.g. pThr298-NLK, p-Raptor, c-Myb degradation, NLK dimerization, expression of siRNA against NLK and recombinant NLK mutants) that strongly support and correlation with the observations found with using the in vitro kinase assay.

Summary: The proposed experiments have been performed, in conjunction with the introduction of a constitutively-active NLK. Our data indicate that NLK activation is required for c-Myb ubiquitination and degradation.

Experiment 13. Kinase-deficient NLK does not significantly improve erythropoiesis.

RPS19-insufficient (shRPS19) or control human cord blood CD34+ hematopoietic stem and progenitor cells were differentiated for 12 days. RPS19 or control shRNAs, cells were transduced with siRNA against NLK alone, or with either a kinase-deficient (KD-NLKesc) or a wild type (WT-NLKesc), or siRNA-resistant NLK cDNA (with intact 3'UTR). Erythroid and myeloid expansion was calculated by multiplying viable cell counts by % CD235+ or CD11b+ respectively. Results are displayed as a percentage of cells of each lineage relative to untreated controls.

Conclusion: Silencing NLK significantly improves erythroid expansion in RPS19-insufficient cells ($p=0.02$). Expression of kinase-deficient NLK does not significantly rescue the effects of siNLK ($p=0.0733$) while expression of wild type NLK rescues the erythroid defect ($p=0.01$).

Experiment 14. Rescue of erythroid expansion of RPS19-insufficient hematopoietic stem and progenitor cells by inhibitors of NLK, but not TGF-beta receptor 1 (ALK5) or P38.

Three TGF-beta receptor inhibitors that suppress NLK activity and improve erythropoiesis in DBA models were titrated into in vitro kinase assays in the presence of immunopurified NLK or P38 or TGF-beta Receptor (ALK5), as well as ATP, Mg²⁺ and dephosphorylated c-Myb substrate (for P38 and NLK) and Smad2 substrate (for ALK5). Determining the extent of substrate phosphorylation at each drug concentration allowed us to calculate the IC50 for each kinase with each compound. Comparison between our observed values and documented values (when available) are presented. The data are represented diagrammatically (as with **Experiment 2**) with IC50 values indicated as vertical lines along a concentration gradient. Kinase activity is represented as blue and the extent of inhibition is depicted in white. Our observed values (shown in orange) can be easily compared with documented IC50 values (shown in black) for each compound against each target.

Conclusion: The TGF-beta receptor inhibitors differentially suppress NLK and p38 activity.

Reviewer Comment 3c) As this reviewer also suggested, the authors should examine the mechanisms of NLK activation. For example, the authors can check whether shRPS19/11 affect the activities of known NLK activators (TAK1 and p38 etc...). At least, they can check the effect of shRPS19/11 on NLK nuclear localization.

Author's New Response: We have begun to examine the role of TAK1 and other upstream factors in NLK activation in RPS19-insufficient cells, this is a focus of future studies to understand regulation of NLK in Diamond Blackfan Anemia.

We have attempted to examine NLK nuclear localization using fluorescence confocal microscopy. The nucleus comprises almost the entirety of these cells so we could not obtain meaningful comparisons. As an alternative, we utilized the fact that NLK dimerizes upon activation (Ohnishi et al, 2010) and examined FRET using NLK tagged with YFP-CFP FRET partners (**Experiment 15 and Fig. 3d**).

Summary: While we could not optimize fluorescence microscopy to visualize nuclear localization of NLK, we utilized NLK dimerization as a surrogate of activity. NLK dimerization corresponded with NLK kinase activity.

Experiment 15. NLK dimerization in RPS19-insufficient hematopoietic stem and progenitor cells is detected by fluorescence resonance energy transfer (FRET) upon expression of YFP-CFP NLK dimer pairs.

NLK dimerizes upon activation (Ishitani et al, 2010 MBoC) so we sought to determine if we could utilize this as a surrogate readout of NLK activity using flow cytometry. Upon dimerization of YFP- and CFP-NLK dimer pairs, excitation of YFP will cause an emission that will excite the dimerized CFP partner in a process called fluorescence resonance energy transfer (FRET). YFP- and CFP-tagged NLK was transduced into CD34+ HSPCs with shRNA against control (shLuc) and RPS19 (shRPS19) and differentiated for 7 days.

Conclusion: A 3.9-fold increase in FRET was detected in RPS19-insufficient cells compared to controls ($p=0.0124$) indicating an increase in NLK dimerization.

Reviewer Comment 3d) I also feel a part of their response is rough. They should cite references when they refer to the past findings. For example, they should cite reference showing “Similar to other kinases that autoactivate upon dimerization, an increase in concentration should increase autoactivation, particularly in vitro”.

Author’s New Response: We apologize for this oversight. A significant number of references were added to the manuscript, including papers examining NLK and p38 interactions and off-target inhibition of NLK by small molecules.

Reviewer Comment 4) Authors indicate that NLK is not deleterious in non-erythroid cells because of miR181. It is not clear why over expression of the NLK-cDNA (Figure 2) is not detrimental to non-erythroid colonies + shRPS19. Does this cDNA contain the

3'UTR region? If not, how do they reconcile this observation? If yes, it is expected that over expression of this mRNA should have some effect also in non-erythroid cells expressing miR181 + shRPS19. Or the mRNA is degraded despite it is over expressed? Please, clarify.

Author's Previous Response: This is correct. We made this observation when trying to re-introduce NLK cDNA (without 3'UTR) into progenitors expressing siRNA against NLK. We had expected to restore erythroid failure, but we observed an unexpected accompanying lineage collapse. This observation was the catalyst to probe the literature and examine the role of miR-181. For this reason, the restored erythroid failure in Fig. 1c and d does require re-introduction of NLKesc cDNA with an intact 3'UTR. We agree that gross overexpression of NLK should have effects in ribosomal-insufficient progenitors. We suspect we haven't observed that when we express our NLK (with 3'UTR) cDNA because intracellular miR-181 expression is high enough to bind to the increased NLK mRNA. MiR-181 induction is quite robust in non-erythroid lineages. It should be noted that NLK cDNA is not driven by a tremendously strong promoter, and it is hard to get a high MOI with HSPCs. We have attempted to further address the role of miR-181 in this process by including data performed in non-differentiating LCLs from DBA patients (Fig. S7). The ability of miR-181 to degrade a high proportion of NLK has been well documented in other cell types (e.g. Cichocki et al, J.Immunol 2011, Junfang et al, Hepatology 2009 and Hutchison et al, 2013).

Reviewer's comments: This is a pointed question. If the authors' hypothesis is correct, overexpression of the NLK-cDNA without 3'UTR affect shRPS19-treated non-erythroid colonies. However, this was no effect (revised Fig 1cd). So, the reviewer requested the authors to explain this discrepancy. But the authors have not explained this clearly.

On the other hand, in revised Fig 7g, overexpression of the NLK-cDNA without 3'UTR affect CD11b+ shRPS19-treated non-erythroid colonies. Only this data supports the authors' hypothesis. This situation (Fig 1cd is inconsistent and only one data is consistent) weaken their hypothesis. To strengthen the hypothesis, the authors clearly explain the discrepancy in revised Fig 1cd. And they can also confirm that overexpression of the NLK-cDNA without 3'UTR or with mutated 3'UTR affect CD41a+ shRPS19-treated non-erythroid colonies.

Author's New Response: We apologize for not explaining this clearly. To the first point raised by reviewer 2, in Fig. 1c&d the NLK expressed does have an intact 3'UTR (as described in the text). Secondly, the expression of the recombinant NLK (with 3'UTR) may not be high enough to overwhelm the mechanism of suppression (i.e. miR-181 binding to 3'UTR) in myeloid cells, otherwise there would be a reduction in non-erythroid cells as well.

While we included in the text in the revised version that NLK re-introduced in Fig 1 did contain an intact 3'UTR we accept this remains confusing. As such, we transduced constitutively active (CA) or wild type (WT) NLK with no 3'UTR into normal differentiating HSPCs. WT-NLK over-expression did not decrease erythroid or myeloid

expansion, whereas CA-NLK decreased both erythroid and myeloid expansion similarly. In contrast, when WT- and CA-NLK with intact 3'UTRs were transduced, erythroid expansion was reduced but myeloid expansion was not impacted.

Our data are consistent with the hypothesis that the 3'UTR causes NLK expression to be suppressed in non-erythroid cells, and that NLK over-expression alone is not sufficient to impact hematopoiesis, but rather NLK activation is required. An additional schematic is included to summarize the proposed regulatory mechanism of NLK expression between lineages, based on our observations (see **Fig. 5f**).

It has been reported that because NLK does not express a regulatory TXY sequence found in other MAPK family members, NLK may have constitutive kinase activity and therefore over-expression could lead to increased NLK activity without the need of upstream activators (Ishitani et al 2009 and 2010). However, we have not observed that over-expression of NLK significantly increases NLK activity in hematopoietic cells. Instead, upstream regulators (including p53) appear to be required for NLK activation. **Summary:** we have altered the text to increase clarity. In addition, we solidified our hypothesis in that expression of recombinant constitutively-active NLK without a 3'UTR reduces erythroid and myeloid expansion, active NLK with a 3'UTR only impacts the erythroid lineage.

Reviewer's comment: In summary, the authors' explanations are confusing and roundabout. It is difficult for reviewers to judge how they improved their manuscript.

Author's New Response: We apologize and have done our best to improve the clarity of our manuscript.

Reviewer #3 (Remarks to the Author)

Comment: The manuscript "Diamond Blackfan is mediated by hyperactive Nemo-like kinase" by Wilkes et al attempts to show a role for a kinase known as the Nemo-like kinase (NLK) in the pathogenesis of DBA. The work is original and potentially of importance as the kinase reported here may be a candidate for developing drugs to treat DBA.

I had a number of concerns in my initial review of this manuscript that have been adequately addressed in the revised version of the manuscript and in the author's rebuttal letter.

Having said this I think the manuscript is still in need of significant copying editing as I found slogging through the results section exceedingly laborious with many instances where I had to read and reread sections to try and discern what the authors were saying. I also found the manuscript extremely long with many experiments repeated in multiple different systems. While this repetition may be necessary to solidify the argument that the Nemo-like kinase has a role in DBA pathophysiology, the length

made me less tolerant of the authors extrapolations in the discussion suggesting that the Nemo-like kinase may be a target of glucocorticoids and leucine therapy. Thus, I think the discussion could be shortened considerably.

Response: We apologize and have significantly shortened both the introduction and discussion. We have rearranged the format of the experiments to increase clarity and flow, and modified the text to reduce redundancy while improving flow and clarity.

Specific examples of copy-editing issues are included below.

Introduction

Authors: Erythroid specificity of ribosomal insufficiency in DBA is largely due to reduced translational efficiency of genes possessing a short, complex 5'UTR that is highly upregulated, particularly GATA13.

Reviewer comment: Reduced translational efficiency, highly upregulated? Are the authors meaning reduced translational efficiency of a mRNA that is normally highly upregulated during erythropoiesis.

Response: The text has been modified to "One reason ribosomal-insufficiency specifically targets erythropoiesis is that many important transcripts that are highly upregulated in erythroid differentiation, particularly GATA1, possess short, complex 5'untranslated regions (UTRs). These transcripts require high ribosome copy numbers for efficient translation and are therefore more sensitive to ribosomal insufficiency".

Results

Authors: Each day, cells were collected and assayed for NLK activity and expression of NLK, RPS19 and RPL11. We observed a mild increase in in vitro phosphorylation of all three substrates at day 3 in control cells.

Reviewer comment: The second sentence mentions three substrates, which are not defined. Given that this follows a sentence where three proteins are mentioned, one could erroneously conclude that the three proteins in the previous sentence are the substrates. Only later in the paragraph does the reader find out that the three substrates are NLK, Myb, and raptor.

Response: Thank you for the comment. We have modified text to improve clarity

Authors: Several TGF β inhibitors in the murine RPS19-insufficient model were thus tested at 10 μ M and only SB431542 and SD208 significantly rescued c-Kit⁺ erythroid expansion, while six other TGF β inhibitors displayed no significant effect (Fig. 4b). Erythroid growth in murine RPS19-insufficiency by SB431542 and SD208 improved with EC₅₀s of 5 μ M and 0.7 μ M respectively (Supplementary Fig. 4a). All of the compounds inhibit TGF β in the RPS19-insufficient murine model, as all rescued proliferation of TGF β -treated c-Kit⁺ cells to varying extents (Fig. 4c). Collectively, this indicates suppression of the TGF β pathway is not the mechanism of action for SB431542 and SD208 during of erythroid expansion in murine RPS19-insufficiency.

Reviewer comment: There has to be a more straightforward way for the authors to make the point that only a subset of TGFbeta inhibitors rescue erythroid expansion in the Rps19-insufficient model, and these are the ones that also inhibit NLK.

Response: We have revised the text in the manuscript as recommended by the reviewer.

Authors: SD208 did not significantly impact erythroid expansion of human CB CD34+ HSPCs transduced with control shRNA (100% to approximately 105%), however SD208 increased erythroid expansion in both RPS19- and RPL11-insufficiency from 8.7% to 38.7% and 5.8% to 36.7%, respectively (Fig. 6a).

Reviewer comment: Sometimes results are presented as above (which I prefer) many other times they are simply as a percent of ribosome-competent controls. Consistency would improve readability.

Response: We have modified the results to improve consistency and readability.

Authors: The presence of SD208 with doxycycline improved erythropoiesis to 83.2%, 81.1% and 76.6% (Fig. 6d).

Reviewer comment: compared to what?

Response: Thank you. We have clarified the controls in the Results.

Discussion

Authors: In patients that qualify, stem cell transplantation can cure the disease, but is associated with a significant risk of serious, and life-threatening complications such as graft versus host disease and malignancies spawned from chemotherapy prior to transplantation.

Reviewer comment: Stem cell transplant only cures hematological manifestations of the disease.

Response: We agree with the reviewer and have corrected the oversight. The sentence has been modified to “stem cell transplantation can cure the hematological manifestations of the disease...”

Additional comments on the authors' response to previous Reviewer #2 (new Reviewer #4's concerns from previous revision):

Reviewer's Comment: Fig. 1: are the patient cells p53-deficient?

Response: This is an excellent question. The DBA patient samples we studied are not p53-deficient nor have p53 mutations (personal communication, Hanna Gazda, Harvard Medical School).

Reviewer's Comment: The response to this comment is not satisfactory. The authors have not measured p53 levels in their patient cells but instead refer to previous work suggesting that p53 levels should be raised in these patient samples since it reported to be a hallmark of DBA. The issue of whether the authors measured p53 levels in patient cells could be a minor issue if this were a tangential point in the current manuscript. However, the authors wish to make the case that hyperactivation of Nemo-like kinase is in response to elevated p53 levels resulting from ribosomal protein haploinsufficiency. Thus, it would seem that measuring p53 levels in patient cells would close the loop in their proposed model.

- Technically, measuring p53 levels would certainly appear to be within the author's capabilities as p53 levels are manipulated in their CD34+ model system in supplemental figure 3 to make the case that Nemo-like kinase activation requires p53.
- It is conceivable that the authors may not be able to measure p53 levels in the patient cells because these samples are no longer available. If this were the case, they could at least measure p53 protein levels in their CD34+ RPS19 knockdown experiments (as reported in supplemental figure 3) and address reviewer 2's comments that way. While not optimal, this would at least close the loop in their model in one of the systems they employ.

Response: Thank you for this excellent suggestion. As the reviewer pointed out, we do not have enough sample to measure p53 levels in the patient samples. We have utilized the fact that NLK dimerizes upon activation (Ohnishi et al, 2010) to circumvent issues of small sample size to examine the role of p53 in NLK activation in DBA models and patient samples. We also examined NLK activity in progenitors stimulated with the p53 activating agent Nutlin-3. These studies revealed Nutlin-3, DBA models and DBA patient samples demonstrate increased populations expressing high levels of p53. Most notably, the populations expressing high p53 that have high NLK dimerization/activity (see **Experiment 16 and Fig. 6b-d**).

Summary: In addition to previous data, our results demonstrate that NLK activation is p53-dependent.

Experiment 16. Activation of NLK is associated with higher p53 expression.

Human cord blood CD34+ hematopoietic stem and progenitor cells (HSPC) were transduced with YFP- and CFP-tagged NLK or control shRNA (shLuc) and allowed to differentiate for 3 days in the presence or absence of the MDM2 inhibitor Nutlin-3. Cells were stained for p53 and Fluorescence Resonance Energy Transfer (FRET) was measured by flow cytometry. For comparison, HSPC were transduced RPS19 shRNA (shRPS19).

Bone marrow mononuclear cells from healthy donors were transduced with YFP- and CFP-NLK and incubated alone, or with Nutlin-3 for 24 hours, prior to p53 and FRET analysis. Bone marrow mononuclear cells from three DBA patients were pooled and analyzed simultaneously.

Conclusion: NLK dimerization (representing NLK activity) correlates with high p53 expression in DBA patient bone marrow cells and RPS19-knockdown HPSCs. Cells with high levels of p53 facilitate NLK dimerization especially in Nutlin-treated cells.

Reviewer's Comment: In light of this, my suggestions are to clarify how NLK is activated in erythroid cells and the role of P53 in patient cells

I agree with the authors that providing a mechanism through which p53 activates Nemo-like kinase seems beyond the scope of the current manuscript. Addressing potential mechanisms at this point would add additional speculation to the discussion, which does not seem warranted.

Response: We are attempting to elucidate the precise mechanism through which p53 activates NLK but agree that is beyond the scope of the current study.

Reviewer #4

In this manuscript, the authors reported the contribution of NLK to erythropoiesis failure in multiple DBA human and murine models. They first observed that ribosomal-insufficiency induced chronic hyperactivation of NLK, and the subsequent phosphorylation of NLK substrates including c-Myb and Raptor. They further showed that chemical and genetic inhibition of NLK increased erythroid expansion in murine and human models of DBA as well as in bone marrow stem and progenitor cells from DBA patients. While these findings are interesting, there are many concerns that need to be addressed. Even after extensive re-writing, the logic to study NLK in DBA as well as the overall flow of the study is still not well presented and not easy to follow. For example, the key point of current manuscript is about the hyperactive NLK in DBA, however they started their studies by examining the effect of NLK expression (rather than NLK activation) on c-Myb expression. Moreover, the importance of studying miRNA181 is not clear. Specific comments are listed below:

Response: We apologize and have revised the manuscript to 1) include the logic to examine NLK; 2) improve overall flow; and 3) emphasize the importance of miR-181 in regulating NLK expression in hematopoiesis.

Major concerns:

1. Fig 1a shows that knockdown of RPS19 resulted in decreased c-Myb protein level. Interestingly, knockdown of NLK in the context of RPS19 knockdown restored c-Myb protein level. These results reveal the correlation between NLK expression and c-Myb protein level but do not necessarily support the authors' conclusion that expression of NLK facilitates c-Myb degradation. To reach this kind of conclusion, one needs to show that overexpression of NLK directly promotes c-Myb degradation rather than loss of NLK restores c-Myb protein level caused by RPS19 deficiency. Additionally, it is important to understand why knockdown of RPS19 led to decreased c-Myb protein level.

Response: Although it is well established NLK phosphorylation of c-Myb leads to ubiquitination and subsequent proteasomal degradation in other cell systems (Kanei-Ishii et al, Kanei-Ishii et al, 2004, Kurahashi et al, 2005), we agree further experimentation was necessary to confirm NLK is regulating c-Myb in the same way in ribosomal-insufficiency. Expression of a constitutively-active NLK in healthy erythroid cultures indeed reduced c-Myb expression that could be prevented by inhibition of the proteasome. We have also included data that was removed from the previous submission indicating both c-Myb in vitro phosphorylation and ubiquitination is dependent on the expression of NLK (see **Experiment 17 and Supplementary Fig 7e**).

Summary: Expression of constitutively-active NLK (not wild type NLK) resulted in c-Myb degradation in ribosome-competent cells. This could be blocked by inhibiting the proteasome.

Experiment 17. c-Myb ubiquitination in RPS19-insufficient cells.

In vitro kinase activity of immunoprecipitated NLK from human cord blood hematopoietic stem and progenitor cells transduced with RPS19 or control shRNA or in conjunction with siRNA against NLK (siNLK) was assessed as described previously. Utilizing the same technique, cell lysates from each sample were incubated with the immobilized substrates for 2 h (with lactacystin) before analysis with anti-ubiquitin antibody.

Conclusion: RPS19-insufficiency increases *in vitro* ubiquitination of c-Myb in an NLK-dependent manner.

2. Why for experiments shown in Fig 1a fetal liver CD34+ cells were used and the cells were differentiated for 5 days, while for Fig 1b, cord blood CD34+ cells and the cells were differentiated for 12 days?

Response: Fetal liver CD34+ were used when larger numbers of cells were required for Western blot and co-immunoprecipitation experiments. Having said this, defects in DBA occur during fetal development. However, as DBA usually manifests within the first year of life or very young children, cord blood was selected to perform the majority of experiments.

It is thought that in DBA, erythropoiesis is blocked early in erythropoiesis so we were interested in NLK activity at early time points. We determined that NLK activation approached maximal around day 5-7 and was early enough that a fair number of erythroid cells remained in RPS19-insufficient cultures. The more mature CD71^{hi}CD235⁺ erythroblast population does not develop until at around day 9 and this is significantly impaired in ribosomal-insufficiency. So to examine the effect of NLK manipulation on erythroid expansion cultures were examined between days 10-12.

3. What is the point to show the results under the subtitle “NLK is differentially expressed between hematopoietic lineages but is not influenced by RPS19-insufficiency”? Fig 1a already showed that RPS19 knockdown does not affect NLK expression.

Response: We agree this is redundant. As there was significant confusion that NLK expression was reduced in RPS19-insufficiency in our previous version, we wanted to

reaffirm the point that NLK is not changed within any lineage several times in the text. We apologize and have clarified this in the revised manuscript.

4. Fig 2a shows that knockdown of RPS19 led to increased phosphorylation of NLK in CD71+ cells after differentiation of CB CD34+ cells for 10 days. Based on this finding, they conclude that NLK is phosphorylated at Thr298 and hyperactivated in erythroid progenitors. It should be noted that most of cells cultured for 10 days are erythroid precursors (erythroblasts) but not erythroid progenitors (BFU-E and CFU-E). Additionally, why the CD34+ cells were differentiated for 10 days here but were differentiated for 5 days and 12 days in Fig 1?

Response: This is an excellent point. We did not differentiate between erythroid precursors and progenitors. In previously experiments NLK activity was assessed in erythroid precursors. While the CD71+ population would include CFU-E progenitors, they would be entirely excluded from a CD235+ population. We know that NLK activation increases in ribosome-insufficiency around day 5 which is when a significant proportion of cells are dividing rapidly and transitioning from progenitors to precursors. Based on this data, it is unclear if NLK is activated in progenitor populations, or just precursors. We have modified the text to make this distinction and have performed experiments to determine if NLK is activated in RPS19-insufficient stem and progenitors (see below for response to comment 7).

5. Fig 3, the figure legend is a little confusing.

Response: Thank you. We have adjusted the legend to improve clarity.

6. It is not clearly whether NLK is selectively activated in erythroid progenitors or it is also activated in HSPCs.

Response: As it is believed DBA results from a deficiency at a erythroid progenitor stage, we sought to examine NLK activity in progenitor populations from RPL11+/+ and RPL11+/flox mice, and in erythroid progenitors after CB CD34+ HSPCs were differentiated *in vitro* for 7 days. We determined that NLK activity is induced in RPS19-insufficiency in Lin-Kit+Sca-CD34-CD16/32- MEPs in mice. No significant increase was observed in more primitive or non-erythroid progenitors, i.e. HSCs, CMPs, or GMPs. In human RPS19-insufficient progenitor populations enriched for erythroid precursors, populations enriched for CFU-Es and populations enriched for MEPs and BFU-Es demonstrated elevated NLK activity relative to healthy controls (see **Experiments 18 and 19 and Figs. 6e, f**).

Summary: NLK is activated in erythroid progenitors.

Experiment 18. Human CB RPS19-insufficient hematopoietic stem and progenitor cells have increased NLK activity MEP- and BFU-E, CFU-E, and proerythroblast- and intermediate erythroblast-enriched populations.

We compared subpopulations within human cord blood RPS19-insufficient or control hematopoietic stem and progenitor cells. During erythropoiesis, modest NLK activation is observed in both RPS19-insufficient and control cells at early time-points. A clear difference in NLK activation is not apparent until after 5-7 days (data not shown). To compare erythroid differentiation at Day 7 and 9 between RPS19-insufficient and control cells, we examined CD71 and CD235 staining by flow cytometry. Day 7 was selected, as the populations remain somewhat similar in composition and NLK activity is maximal (see **Supplementary Fig. 7b**).

Samples were gated to include non-erythroid CD71⁻CD235⁻ populations, MEP- and BFU-E-enriched CD71^{low}CD235⁻ populations, CFU-E-enriched CD71^{hi}CD235⁻ populations and proerythroblast and intermediate erythroblast CD71^{hi}CD235⁺ populations. Sorted populations were lysed and immunoprecipitated NLK was subjected to *in vitro* kinase assay examining Raptor phosphorylation was used as a substrate. Activity was normalized to the non-erythroid control samples.

Conclusion: NLK activation occurs within RPS19-insufficient cell populations containing MEPs and BFU-Es (p=0.0064) and is retained throughout erythroblast precursor stages.

Experiment 19. NLK activity is elevated compared to controls in Megakaryocyte/erythroid Progenitors (MEPs), but not Hematopoietic Stem Cells (HSCs) and earliest non-committed progenitors, Common Myeloid Progenitors (CMPs) or Granulocyte and Macrophage Progenitors (GMPs) from RPL11-insufficient mice.

Lin-Kit⁺ cells from RPL11^{+/+} and RPL11^{+/flox} were sorted as described in Mayle et al, 2013. Populations are listed below:

Hematopoietic Stem Cells (HSCs) and early non-committed progenitors: Lin⁻Kit⁺Sca⁺

Common Myeloid Progenitors (CMPs): Lin⁻Kit⁺Sca⁻CD34⁺CD16/32⁻

Granulocyte Monocyte Progenitors (GMPs): Lin⁻Kit⁺Sca⁻CD34⁺CD16/32⁺

Megakaryocyte Erythroid Progenitors (MEPs): Lin⁻Kit⁺Sca⁻CD34⁻CD16/32⁻

Sorted populations were lysed and immunoprecipitated NLK was subjected to *in vitro* kinase assay examining Raptor phosphorylation as a substrate. Activity was normalized to the Lin⁻Kit⁺Sca⁺ RPL11^{+/+} samples.

Conclusion: NLK activity is induced in MEPs (p=0.0207) in an RPL11 mutant mouse model of DBA.

Minor concerns:

1. c-Myb is a master transcription factor for hematopoiesis but not specifically for erythropoiesis. The statement that c-Myb is a master TF for erythropoiesis is somewhat misleading.

Response: This has been revised in the text.

2. The authors gave a long introduction about NLK, some of them are not relevant to the current study.

Response: We have revised the introduction to be more concise.

3. NLK is widely expressed in other tissues and cell types. Targeting NLK may lead to many side effects.

Response: Thank you. We have included this in the discussion.

4. Why the authors keep emphasizing “intracellular” phosphorylation? Is this extracellular phosphorylation?

Response: This terminology was an attempt to differentiate substrate phosphorylation after *ivk (in vitro)*, with substrate phosphorylation observed from proteins lysed from cells (intracellular). However, this nomenclature has been removed. We apologize for the confusion.

5. Please clarify what do you mean by “chronic hyperactivation”?

Response: The term “a sustained activation of NLK significantly in excess of basal levels – chronic hyperactivation” has been added when this term is first introduced.

6. After stating the changes in %, there is no need to repeat fold change.

Response: Thank you. This has been revised.

7. Fig 1c, for colony assays, the authors differentiated GFP+RFP+ progenitors 12-15 days. It should be noted that human CFU-E colonies should be counted on day 7.

Response: Only BFU-E (erythroid) and CFU-GM (myeloid) colonies were counted. We agree that CFU-E colonies would be lost at such late time points. Clarification that BFU-E and CFU-GM colonies were scored will be added to the figure legend.

8. The quality of some western blots is poor and needs to be improved.

Response: Thank you. We have repeated the Western blot analyses to improve the quality of the blots to the best of our ability.

Reviewers' Comments:

Reviewer #1:

Remarks to the Author:

The authors respond almost all our comments. But explanation of new data is insufficient. In addition, the authors added novel findings, which were not shown in the previous version and are not requested by Reviewers. I feel that the authors make their paper more complicated by themselves. As reviewer, we should check whether such new hypothesis is well demonstrated or not in the revised manuscript. If the authors add new unconfirmed and low quality data every revision, review process may be endless. Although I think this study is potentially interesting, I am not able to accept if the authors provide the new data that is not well confirmed. They should simplify this paper to make us understand what is the most important finding that they want to show in this study.

In rebuttal, there is a copy & paste, that may be the authors' careless mistake. I found redundant responses to two different Reviewer's comment. "p38 association with NLK has been reported previously (Ohnishi et al,2010). Much of the experimental data ..." appears twice in Rebuttal. It is indicating that the authors did not answer "Reviewer #1 Comments: To exclude the possibility the involvement of other kinases, Reviewer1 recommend...".

Layout of Figures is not good. Please separate each small Figs (a, b, c, d,...). It is difficult to understand each Figure.

The authors consider that p38 is not essential for NLK activation, while Ohnishi et al. reported that p38 is essential for NLK activation. Therefore, the authors should discuss about this discrepancy in the main text. (For example, contribution of p38 on NLK activation is context-dependent). I think if they used pTr298 antibody, but not pSer, in all ivk, they can reduce the discussion about p38... (But I understand it is difficult and tough for the authors).

Are both NLK and p38 proteins in Fig 3f derived from RPS-19 insufficient cells? The authors should clearly mention this. It is difficult to understand what they do in this Figure. Reviewer1 is requesting the authors to check whether treatment of anti-NLK immunoprecipitates from RPS-19-insufficient cells with p38 specific inhibitor does not affect the immunoprecipitates-induced substrate phosphorylation (Myb and/or Raptor).

Although the authors showed too many data in Fig3f, they are not essential. I recommend the authors to simply show that p38 specific inhibitor does not affect the immunoprecipitates-induced Myb (and/or Raptor) phosphorylation.

In Supplementary Fig 4b and 4c, the authors stated that RPS-19-insufficiency activates NLK, but not p38. However, the NLK activity was amplified by preincubation with CIP-treated NLK, indicating that it is not pure comparison. I think these figures are unessential. Therefore, I disagree the authors' discussion based on these figures.

I think that the above p38 specific inhibitor experiment is critical and sufficient. p38 kinase assay data does not strengthen their hypothesis.

What is "induction" in Supplementary Fig 4b?

It is difficult to understand Supplementary Fig 4c and 4d. I do not understand the meaning of Supplementary Fig 4d. To compare NLK activity with p38 activity, this experiment should be done using recombinant NLK and p38 proteins (e.g. bacterially expressed one) to compare their activities (but they used immunoprecipitated NLK and p38 from unspecified cells).

But I don't recommend them to improve these Figures. I think these data are not essential in this study.

The authors stated that CA-NLK increase c-Myb degradation in Fig7c and erythroid failure in Fig 5a. What is CA-NLK? Cite the reference. I could not find any reports showing that T298E mutation constitutively activates NLK. In addition, the quality of Fig7e is very low, so I do not agree their idea. To use T298E mutant as CA-NLK, they should show the evidence (e.g. using bacterially expressed recombinant proteins). But I do not request the authors to show this. CA-NLK data are not essential for this study. I think that reviewers do not request such experiments and that careless data adding would spoil your interesting study.

The authors stated that NLK RNAi did not affect MAPKs activity (Supplementary Fig. 2d). But, in their data, p38 and ERK activities appear to be changed. To clearly show that NLK RNAi is no effect, I recommend them to quantify them.

Reviewer #3:

Remarks to the Author:

My remaining concerns have been adequately addressed.

Reviewer #4:

Remarks to the Author:

The authors performed additional experiments and addressed most of my concerns. However, the quality of some western blots is still not good. These include Fig 7e (the blot for c-myb), some blot of supplementary Fig 3c, 3d and supplementary Fig 4a. In addition, supplementary Fig 6e is difficult to read.

Reviewers' comments

We would like to thank the reviewers for their helpful and thoughtful comments. We hope that we have addressed each point. The manuscript is significantly improved as a result of the reviewer's suggestions.

Data removed in the re-submission include:

Fig.5a (replaced with data including RPS19-insufficient HSPCs).

Fig. 7e

Supplementary Fig. 4b-e

Reviewer #1 (Remarks to the Author):

Comment 1: The authors respond almost all our comments. But explanation of new data is insufficient. In addition, the authors added novel findings, which were not shown in the previous version and are not requested by Reviewers. I feel that the authors make their paper more complicate by themselves. As reviewer, we should check whether such new hypothesis is well demonstrated or not in the revised manuscript. If the authors add new unconfirmed and low quality data every revision, review process may be endless. Although I think this study is potentially interesting, I am not able to accept if the authors provide the new data that is not well confirmed. They should simplify this paper to make us understand what is the most important finding that they want to show in this study.

Response: We sincerely apologize if we failed to address any of the reviewers' concerns and if the explanation of newly included data was not clear. In response to this reviewer, we further explored the possibility of NLK-associated p38 contributing to the phenotype. As one of these experiments did reveal a previously undetected association between the two proteins, we attempted to thoroughly examine this issue. However, we did not observe a role for p38 in experiments using our model systems. We agree with the reviewer that inclusion of these "negative" data dilutes the main conclusions of our study. Consequently, we have taken the advice of the reviewer and removed much of these data (Figs. 5a, 7e and Supplementary Fig. 4b-e).

Comment 2: In rebuttal, there is a copy & paste, that may be the authors' careless mistake. I found redundant responses to two different Reviewer's comment. "p38 association with NLK has been reported previously (Ohnishi et al,2010). Much of the experimental data ..." appears twice in Rebuttal.

Response: We apologize for the oversight. An amended response is included below.

Previous comment: "To exclude the possibility the involvement of other kinases, Reviewer1 recommend treatment with kinase inhibitor. However, the authors stated "p38 and TAK1 inhibitors have off-target effects on other kinases, including NLK" in rebuttal, and they did not perform this recommended experiment. To my knowledge, there are no reports showing p38 and TAK1 inhibitors have off-target effects on NLK. So, I am not convinced their opinion"

Response: We performed the recommended experiment with 8 different p38 inhibitors (See Fig. 3f). Confirming our original statement that “p38 and TAK1 inhibitors have off-target effects on other kinases, including NLK”, a number of p38 inhibitors did impact NLK, although at different IC₅₀ values. It is for this reason we chose to perform the IC₅₀ analysis rather than treating with inhibitors at a single dose. Although we agree this adds complexity to the data, we feel this more comprehensively addresses the off-target effect of some of these drugs and the extent to which p38 is involved in the erythroid phenotype. TAK1 has not been reported to phosphorylate c-Myb or Raptor and therefore is unlikely to be an associated kinase contributing to these phosphorylation events. TAK1 is upstream of NLK activation in other systems and we are actively investigating this, but in agreement with other reviewers, including these data would be beyond the scope of this study.

Comment 3: Layout of Figures is not good. Please separate each small Figs (a, b, c, d,,). It is difficult to understand each Figure.

Response: Thank for this suggestion. We have added additional (i, ii, iii....) to the figures with multiple panels (Figs. 1-6).

Comment 4: The authors consider that p38 is not essential for NLK activation, while Ohnishi et al. reported that p38 is essential for NLK activation. Therefore, the authors should discuss about this discrepancy in the main text. (For example, contribution of p38 on NLK activation is context-dependent). I think if they used pTr298 antibody, but not pSer, in all ivk, they can reduce the discussion about p38... (But I understand it is difficult and tough for the authors).

Response: We agree and have added “As it was reported that p38 is required for NLK function in anterior development in *Xenopus*²¹, our data suggests the role for p38 in NLK activation is context-dependent. As p38 phosphorylates NLK at a specific serine residue, it is possible redundancy exists between kinases that can phosphorylate this residue under different conditions.” to the main text. We compared pSer and pThr298-NLK after ivk with both NLK and p38 (included in source data for Supplementary Fig. 4b). To summarize, we uncovered that p38 from both control and RPS19-insufficient cells could phosphorylate NLK at serine residues (but not Thr298). However, as p38 inhibition does not prevent NLK activation in these cells, either this phosphorylation is not required for activation or there are redundant kinases that phosphorylate NLK. We do agree with other reviewers that upstream activation of NLK is beyond the scope of this study so these data are not included in this manuscript.

Comment 5: Are both NLK and p38 proteins in Fig 3f derived from RPS-19 insufficient cells? The authors should clearly mention this. It is difficult to understand what they do in this Figure. Reviewer 1 is requesting the authors to check whether treatment of anti-NLK immunoprecipitates from RPS-19-insufficient cells with p38 specific inhibitor does not affect the immunoprecipitates-induced substrate phosphorylation (Myb and/or Raptor).

Response: In this experiment both NLK and p38 are immunopurified from Kp53A1 cells (K562 cells expressing a temperature sensitive p53 mutant) which is a transformed erythromyeloid leukemic line that shares many of the same properties as erythroid progenitors. NLK activation increases in these cells in response to RPS19 suppression and p53 upregulation, just as occurs

in primary erythroid progenitors. The same activation profile of NLK from these cells and erythroid progenitors is documented in Supplementary Fig. 3). We have now added “A panel of 8 small molecule p38 inhibitors were titrated into *in vitro* kinase assays in the presence of activated NLK or p38 from stimulated Kp53A1 cells” to the figure legend. A more comprehensive description of how IC₅₀ values are calculated and presented is now included in the Materials and Methods as IC₅₀ calculations were obtained using 8-9 concentrations of compound and generated using the IC₅₀ Calculator software by AAT Bioquest®. NLK, p38 or ALK5 for *in vitro* kinase assays were immunopurified from Kp53A1 cells. NLK activation was induced by incubating cells at 32 degrees for 24-48 hours, p38 by stimulating with IL-3, IL-6, SCF and Epo at the same concentrations used for differentiating erythroid progenitors. IC₅₀ values for each kinase and compound were plotted along a horizontal bar with our observed value superimposed next to documented values (if available).”

Although we did not observe differences between NLK activated between Kp53A1 cells and erythroid progenitors (see Supplementary Fig. 3), we acknowledge that some differences may exist. Since p38 inhibitors did not have a dramatic effect on erythroid expansion (Supplementary Fig. 2), additional experiments would mostly not add to results that are in the manuscript. We performed several experiments using different approaches demonstrating that p38 does not play a major role in activation of NLK in ribosome-insufficient cells.

Comment 6: Although the authors showed too many data in Fig3f, they are not essential. I recommend the authors to simply show that p38 specific inhibitor does not affect the immunoprecipitates-induced Myb (and/or Raptor) phosphorylation.

Response: We apologize for the complexity of these data. Since NLK can be an off-target of p38 inhibitors, comparing IC₅₀ values would be more sensitive than examining single doses of the drugs. In addition, some first generation p38 inhibitors potently inhibited NLK. Therefore, inclusion of first, second and third generation p38 inhibitors was necessary to further address the question. As each drug acts on the kinase directly (not the kinase substrate interaction) we observed almost identical IC₅₀ values when comparing substrates (NLK, Myb, Raptor). We have added the sentence “IC₅₀ values were obtained for each kinase against each substrate (eg. NLK phosphorylation of NLK, c-Myb and Raptor), but as IC₅₀ curves were almost identical for each substrate, only one is shown.” in the Materials and Methods section.

Comment 7: In Supplementary Fig 4b and 4c, the authors stated that RPS-19-insufficiency activates NLK, but not p38. However, the NLK activity was amplified by preincubation with CIP-treated NLK, indicating that it is not pure comparison. I think these figures are unessential. Therefore, I disagree the authors' discussion based on these figures. I think that the above p38 specific inhibitor experiment is critical and sufficient. p38 kinase assay data does not strengthen their hypothesis.

Response: We agree that this result is not critical for the conclusion of our study and has been removed.

Comment 8: What is “induction” in Supplementary Fig 4b?

Response: NLK was induced in Kp53A1 cells by incubating the cells for 24-48 hours at 32 degrees to stabilize p53. P38 was induced by addition of erythropoietin, IL-3, IL-6 and SCF. All of which are recognized to induce p38 activity in these cells.

Comment 9: It is difficult to understand Supplementary Fig 4c and 4d. I do not understand the meaning of Supplementary Fig 4d. To compare NLK activity with p38 activity, this experiment should be done using recombinant NLK and p38 proteins (e.g. bacterially expressed one) to compare their activities (but they used immunoprecipitated NLK and p38 from unspecified cells). But I don't recommend them to improve these Figures. I think these data are not essential in this study.

Response: We apologize for any confusion. Since these figures are not important for the main conclusions of the study, these figures have been removed.

Comment 10: The authors stated that CA-NLK increase c-Myb degradation in Fig7c and erythroid failure in Fig 5a. What is CA-NLK? Cite the reference. I could not find any reports showing that T298E mutation constitutively activates NLK. In addition, the quality of Fig7e is very low, so I do not agree their idea. To use T298E mutant as CA-NLK, they should show the evidence (e.g. ivk using bacterially expressed recombinant proteins). But I do not request the authors to show this. CA-NLK data are not essential for this study. I think that reviewers do not request such experiments and that careless data adding would spoil your interesting study.

Response: We agree with the reviewer regarding use of the constitutively active NLK (CA-NLK). While expression of this construct does induce NLK-mediated activities (including erythropoiesis failure and Myb degradation), we acknowledge that further characterization of the mechanism of action would be optimal prior to publication. Indeed biological activity of this construct required both the modification of the amino acid residues and addition of a NLS sequence to localize it to the nucleus. We are currently performing experiments to understand the mechanisms regulating NLK activation. We agree with the reviewer that inclusion of these data would be more appropriate to include in studies to understand the mechanism of action. For these reasons, we have removed the data with the CA-NLK.

Comment 11: The authors stated that NLK RNAi did not affect MAPKs activity (Supplementary Fig. 2d). But, in their data, p38 and ERK activities appear to be changed. To clearly show that NLK RNAi is no effect, I recommend them to quantify them.

Response: Thank you for the comment. We have quantified band intensity using ImageJ and recorded mild reductions in siNLK-treated samples. This is now reflected within the results section "Mild reductions in p38 (16%), JNK (7%) and ERK1/2 (14%) phosphorylation were observed (Supplementary Fig. 2d)".

Reviewer #3 (Remarks to the Author):

My remaining concerns have been adequately addressed.

Response: We would like to thank the reviewer for the comment.

Reviewer #4 (Remarks to the Author):

The authors performed additional experiments and addressed most of my concerns. However, the quality of some western blots is still not good. These include Fig 7e (the blot for c-myb), some blot of supplementary Fig 3c, 3d and supplementary Fig 4a. In addition, supplementary Fig 6e is difficult to read.

Response: The data provided in Supplementary Figs 3-4 are supported by the other assays throughout the manuscript. Therefore, Fig. 7e has been removed (see our response to Comment 10). Fig. 6e has been amended for clarification.

Reviewers' Comments:

Reviewer #1:

Remarks to the Author:

Overall, the authors have done a mostly adequate job of responding to the reviewers' comments.

Reviewer #4:

Remarks to the Author:

The authors have addressed my remaining concerns

Response to the Reviewer's Critiques

We would like to thank the reviewers for their helpful comments and suggestions. We have addressed each of the reviewer's concerns point by point

Reviewer #1 (Remarks to the Author):

Overall, the authors have done a mostly adequate job of responding to the reviewers' comments.

Response: We thank the reviewer for their assessment.

Reviewer #4 (Remarks to the Author):

The authors have addressed my remaining concerns

Response: We are grateful for the reviewer's input and thankful for their assessment.